# Dual targeting of tumoral cells and immune microenvironment by blocking the IL-33/IL1RL1 pathway

Denggang Fu[1,2], Hua Jiang[1,2], Alan Long [3], Ella Harris[1], Hongfen Guo[3], Maegan L. Capitano[2], John Wrangle[4], Joshua R. Faust[5], Anilkumar Gopalakrishnapillai[5], Santhosh Kumar Pasupuleti [2], Baskar Ramdas [2], Reuben Kapur [2], Sonali P. Barwe [5], Nai-Kong V. Cheung [3] & Sophie Paczesny [1,2] ✉

Leukemia stem cells (LSCs) are a small yet powerful subset of leukemic cells that possess the ability to self-renew and have a long-term tumorigenic capacity, playing a crucial role in both leukemia development and therapy resistance. These LSCs are influenced by external and internal factors within the bone marrow niche. By delving into the intricate interplay between LSCs and their immune environment, we can pave the way for innovative immunotherapies that target both the malignant stem cells and the suppressive immune microenvironment, addressing both the "seed" and the "soil" simultaneously. Through the analysis of public datasets and patient samples, we show that elevated IL1RL1 expression correlates with poor prognosis and therapy resistance in acute myeloid leukemia (AML). At the core of this process, stem cell leukemogenesis initiation and maintenance signals are driven by a stress-induced IL-33/IL1RL1 autocrine loop. This LSC-induced IL-33/IL1RL1 signaling fosters an immune regulatory microenvironment. Therefore, IL1RL1 emerges as a promising therapeutic target, with IL1RL1-specific T cell-engaging bispecific antibodies holding great potential as cutting-edge immunotherapeutics for AML.

Therapies for acute myeloid leukemia (AML) have barely changed over 30 years while treatment for other blood cancers have made remarkable leaps forward[1]. Recent advances in genomics have allowed for molecular targeted therapies (i.e., FLT3-ITD, IDH, c-KIT inhibitors) to extend survival but most patients still succumb to AML[2]. Therefore, developing more efficient, less toxic, and immune-based therapies for AML is an urgent unmet need. Leukemia stem cells (LSCs), a small subset of leukemic cells that are capable of self-renewal and thus able to have long-term leukemogenic capacity, play a pivotal role in leukemia development, relapse, and therapy refractoriness. Like normal hematopoietic stem cells (HSCs), LSCs are regulated by internal (intrinsic) and external (extrinsic) factors from the stem cell niche located in the bone marrow (BM). Mechanistic understanding of the interactions of LSCs with these two types of signaling in the BM niche could identify new targets which may accelerate the development of therapeutics.

*Interleukin-33 (IL-33)*, a member of the IL-1 cytokine family and an alarmin released upon cellular stress, is crucially involved in type 2

[1]Department of Microbiology and Immunology and Pediatrics, Medical University of South Carolina, Charleston, SC, USA. [2]Department of Pediatrics, Indiana University School of Medicine, Indianapolis, IN, USA. [3]Department of Pediatrics, Memorial Sloan Kettering Cancer Center, New York, NY, USA. [4]Department of Microbiology and Immunology and Medicine, Medical University of South Carolina, Charleston, SC, USA. [5]Nemours Children's Hospital, Lisa Dean Moseley Foundation Institute of Cancer and Blood Disorders, Wilmington, DE, USA. ✉e-mail: paczesns@musc.edu

immunity, allergy, and inflammation[3]. Different types of cells, such as epithelial, endothelial, stromal, and immune cells, produce IL-33[4]. Dysregulation of the IL-33–Il1rl1 pathway has been documented in various solid tumors and myeloid malignancies[5–9]. Previous studies showed that stromal cell-derived IL-33 contributes to the pathogenesis of myeloproliferative neoplasms[10], and that tumoral IL-33 increases TGF-β production by macrophages in squamous cell carcinoma[11]. *Il1rl1* is IL-33 unique receptor[4]. In the Cbfb-MYH11 murine model, Il1rl1 has been shown to be expressed on LSCs and contributes to their survival[7]. We and others, also, found that Il1rl1 is expressed on normal murine[12] and human[13] HSCs. In addition, IL-33 enhances proliferation of the stem cells after tissue damage[14,15]. A synergistic partnership between IL-33/Il1rl1 and Wnt pathway through Bcl-xL drives gastric cancer stemness and metastasis[6]. Therefore, understanding the underlying mechanisms of how intrinsic IL-33/Il1rl1 signaling induces LSCs to undergo self-renewal could provide a specific therapeutic strategy for AML stem cells.

Cell-extrinsic cues may originate from the immune cells in the leukemic niche. For example, AML patients have been shown to have increased regulatory T cells among other immune regulatory cells as compared to healthy donors[16]. We have shown that Il1rl1 blockade enhanced graft-versus-leukemia (GVL) activity against MLL-AF9 (KMT2A-MLLT3) AML after hematopoietic cell transplantation and this survival benefit was higher than expected with the sole graft-versus-host disease (GVHD) protection[17], suggesting a potential increase in the antitumoral activity upon Il1rl1 neutralization of the malignant niche. However, the role of the intrinsic IL-33/Il1rl1 signaling in LSCs and its interactions with the immune leukemic BM niche has not been explored.

In this study, we investigated how stem cell leukemogenesis is initiated and maintained and we hypothesized that a stress-induced intrinsic IL-33/Il1rl1 autocrine loop will initiate and promote leukemia stemness. We further hypothesized that this IL-33/Il1rl1 autocrine loop in LSCs will induce an immune tolerogenic microenvironment. Our study shows that a dual targeting of LSCs and immune cells in the leukemic niche with engineered T cell engaging anti-Il1rl1 bispecific antibodies prevent and treat AML.

## Results

### High IL1RL1 in AML cells and LSCs correlates with poor prognosis

To determine the clinical relevance of IL1RL1 in AML, we first generated a Kaplan-Meier curve using a combination of The Cancer Genome Atlas (TCGA, $n = 173$) and TARGET-AML ($n = 187$) databases[18]. Decreased survival was observed in patients with high IL1RL1 expression (Fig. 1A), as determined by comparing IL1RL1[high] and IL1RL1[low] populations, which was validated in AMLCG 1999 trial independent database[19] ($n = 417$, Fig. 1B). To identify IL1RL1 expression in leukemic cells, we utilized the same databases and found that expression of IL1RL1 is increased in leukemic cells from AML patients in the TCGA and TARGET ($n = 360$) databases as compared to GTEx healthy donors (HD) samples[18] ($n = 514$, Fig. 1C). This was further validated in the independent Fred Hutchinson Cancer Center (FHCC)[20] and AMLCG 1999 AML databases (Fig. 1D). We utilized transcriptomic datasets of molecularly defined AML subgroups from the FHCC AML dataset including IDH1[mut], IDH2[mut], NPM1[mut], CEBPA[mut], FLT3[ITD], FLT3[TKD], NRAS[mut], EVI1[POS] and found that IL1RL1 was elevated in all these groups compared to normal HD HSCs[21] (Fig. 1E). IL1RL1 expression was higher in patients with various cytogenetic abnormalities including MLL-AF9 compared to HD (Supplementary Fig. 1A). Additionally, using the cytogenetic definitions of the Microarray Innovations in Leukemia Study Group (MILE) study[22], we analyzed the recent Naef et al.'s dataset[23] which independently validated that IL1RL1 was elevated in AML regardless of the cytogenetic mutation (Supplementary Fig. 1B). We addressed wether IL1RL1 expression is lower in AML of faborable-risk [t(8;21), t(15;17), inv(16)] vs. intermediate-risk (characterized by the absence of favorable or unfavorable cytogenetic and molecular abnormalities) and if

IL1RL1 expression would stratify further AML with intermediate-risk. This was not the case as shown in Supplementary Fig. 1C. Together, these data suggest that alarmin IL1RL1 is not driven by unfavorable cytogenetic.

Next, we performed a correlation analysis of IL1RL1 expression within the infiltrated proportions of leukemic stem and progenitor cells (LSPCs) in the leukemic BM microenvironment using a combination of TCGA and TARGET-AML databases by xCell deconvolution. This algorithm allows for reliably portraying the cellular heterogeneity landscape of bulk RNA-seq data[24]. Fractions of 6 types of LSPCs were compared between IL1RL1[high] and IL1RL1[low] expression groups according to the median expression level. LSPCs which include LSCs, common myeloid progenitors (CMPs), and granulocyte-macrophage progenitors (GMPs) were all increased in patients with IL1RL1[high] expression (Fig. 1F). This was also validated in an independent AMLCG1999 database (Fig. 1G).

Since IL1RL1 is expressed in normal HSCs[12,13], we interrogated if IL1RL1 is expressed in total LSCs defined as CD34+CD38− in the Princess Margaret leukemia biobank[25] ($n = 54$) and showed that IL1RL1 expression is higher on LSCs as compared to HSCs, which was verified in an independent dataset[26–29] (Fig. 1H). Furthermore, we observed that IL-33 was up-regulated in LSCs in comparison to HSCs (Fig. 1I). We then measured the frequencies of IL1RL1+ LSCs cells in BM samples between AML complete responders (CR) vs. nonresponders (NR) (characteristics in Supplementary Table 1, gating strategy in Supplementary Fig. 2A) showing that IL1RL1+ LSCs following chemotherapy induction were increased in NR as compared to patients in CR (Supplementary Fig. 1D). Considering the heterogeneity of CD34+CD38−LSCs[25], we further compared IL1RL1 expression in the most primitive CD34+CD38−CD90+CD45RA− LSCs population comparing BM samples from NR and CR patients and found that frequencies of IL1RL1 positive primitive LSCs were higher in NR than that of CR patients (Fig. 1J). Investigation of IL1RL1 and IL-33 co-expression in the primitive LSCs showed that the frequency of IL1RL1+IL-33+ LSCs was increased in NR as compared to CR patients (Fig. 1K, Supplementary Fig. 1E). Furthermore, we found that IL1RL1 expression in human HSCs in steady state is nearly undetectable and much lower as compared to LSCs in the leukemia niche (Supplementary Fig. 1F). To determine whether IL1RL1 high expression is specific to LSCs, we analyzed bulk RNA sequencing data of AML patients[30,31]. As shown in Supplementary Fig. 1G, our results indicate that this is not the case. Since proportions of CD3+ T lymphocytes have been shown to be decreased while frequencies of CD3+FOXP3+ regulatory T cells (Tregs) were increased in AML at diagnosis[16], we also explored these populations frequencies following chemotherapy induction in the NR and CR patients and found similarly to this previous study[16] a depletion increase of total CD8+ T cells, including cytotoxic CD8+IFN-γ+ T cells, and an increase of CD4+FOXP3+ T cells in the NR vs. CR patients (Supplementary Fig. 3A−C). These data indicated that IL1RL1 and IL-33 is overexpressed in total leukemic cells, LSCs, and primitive LSCs, and their expression induced changes in the immune microenvironment landscape, and correlated with poor AML prognosis requiring further investigation of the IL-33/IL1RL1 axis in leukemogenesis and immunity.

### Il1rl1 promotes leukemogenesis initiation tested via limiting dilution assay and engraftment in secondary and tertiary transplantations

Our previous study demonstrated that Il1rl1 is a positive regulator of HSCs in murine models[12]. This finding prompted us to investigate the role of Il1rl1 in initiating MLL-AF9-derived leukemia, an aggressive MLL fusion oncogene derived leukemia[32]. To this end, using a similar strategy as described in previous reports[33–35] and similar retroviral transduction step to express MLL-AF9[17], we generated C57BL/6 WT and Il1rl1−/− LSCs by transducing HSCs-enriched hematopoietic stem/progenitor cells (LSK: Lin−c-KIT+Sca-1+) with retroviral encoding MLL-AF9, and performed limited dilution assay by transplanting 500, 200, and

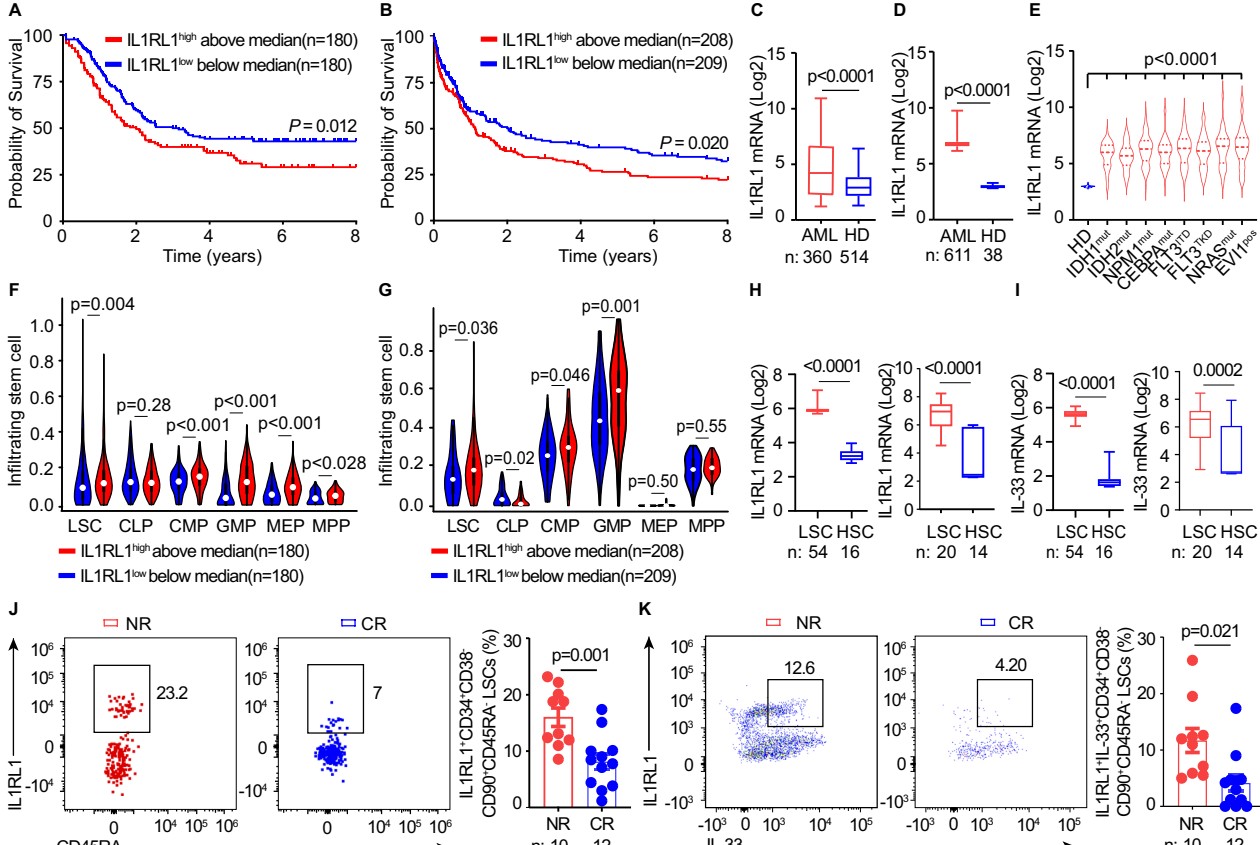

**Fig. 1 | High IL1RL1 in leukemic cells and LSCs correlate with poor prognosis and refractoriness in AML patients.** Kaplan-Meier survival of AML patients with high vs. low IL1RL1 expression (above vs. below median) in combined TCGA/TARGET datasets (**A**, $n = 360$) and AMLCG1999 (**B**, $n = 417$). Log-rank test. IL1RL1 mRNA expression in AML patients vs. healthy donors from TCGA/TARGET (AML, $n = 360$; HD, $n = 514$) (**C**), and FHCC/AMLCG1999 (AML, $n = 611$; HD, $n = 38$) (**D**). In box plots, the center line represents the mean, the box limits represent the 25th and 75th percentiles, and the whiskers extend to the minimum and maximum values of IL1RL1 expression. Unpaired two-sided t-test was used. **E** IL1RL1 expression across molecular AML subtypes (IDH1^mut, $n = 51$; IDH2^mut, $n = 55$; NPM1^mut, $n = 216$; CEBPA^mut, $n = 52$; FLT3^ITD, $n = 190$; FLT3^TKD, $n = 79$; NRAS^mut, $n = 68$; EVI1^pos, $n = 51$) and healthy donors ($n = 46$) from the FHCC AML dataset. ANOVA analysis with post-hoc Bonferroni t-test was used. Proportion of infiltrating LSC and progenitor subsets in

IL1RL1^high and IL1RL1^low AML patients from TCGA/TARGET (**F**, $n = 180$/group) and AMLCG1999 (**G**, $n \approx 208$/group) datasets. Violin plots show median, the 25th and 75th percentiles, and whisker. Wilcoxon rank-sum test. IL1RL1 (**H**) and IL-33 (**I**) mRNA expression in AML LSCs (CD34+CD38−) vs. normal HSCs (CD34+CD38+/−) from the Princess Margaret dataset (LSC, $n = 54$; HSC, $n = 16$), GSE63270 ($n = 20$), and healthy HSCs from GSE24759, GSE17054, and GSE19599 ($n = 14$). Data are presented as box and whisker plots. Unpaired two-sided t-test was used. **J** Representative flow plots and frequencies (mean ± SEM) of IL1RL1+LSCs in bone marrow aspirates ~21 days post-induction chemotherapy in AML non-responders (NR, $n = 10$) vs. complete responders (CR, $n = 12$). LSCs defined as CD34+CD38−CD90+CD45RA−. Unpaired two-sided t-test was used. **K** Representative plots and frequencies (mean ± SEM) of IL1RL1+IL-33+LSCs in NR vs. CR AML patients, LSCs as defined in (**J**) ($n = 10$–12/group). Unpaired two-sided t-test was used.

50 FACS-sorted total LSCs from WT vs Il1rl1−/− MLL-AF9 transduced cells into lethally irradiated C57BL/6 syngeneic recipient mice (Fig. 2A). We assessed the functional LSC frequency from the primary transplanted leukemic mice and found that the leukemia initiating frequency of LSCs in Il1rl1−/− cells was decreased by 15-fold as compared to WT cells ($p = 3.37e-5$, Fig. 2A). Il1rl1 deficiency in LSCs further inhibited their expansion in the BM demonstrated by decreased frequencies of leukemic cells in the serially transplanted mice which led to extended survival as compared to WT LSCs recipients visible with each dose of LSCs (Fig. 2A). Secondary transplantations from the primary recipients resulted in a progressive delay of leukemia growth and better survival in Il1rl1−/− vs WT LSCs recipients (Fig. 2B). Tertiary transplantations of Il1rl1−/− LSCs continued to show a delay in leukemia growth and an improved survival versus WT LSCs (Fig. 2C).

**Il1rl1 deficiency under inducible hematopoietic-specific Mx1 promoter shapes steady state hematopoiesis**
Next, we aimed at determining the requirement of Il1rl1 for normal hematopoietic function in vivo. To investigate this, we developed mice with loxP-flanked alleles of Il1rl1 (Il1rl1^f/f). These Il1rl1^f/f mice were

crossed with expressing Cre recombinase under inducible hematopoietic-specific Mx1 promoter to generate Il1rl1^f/f Mx1Cre mice, allowing for the deletion of il1rl1 in hematopoietic compartments. The experimental scheme of Il1rl1 deficiency in normal hematopoiesis is shown in Supplementary Fig. 4A. Il1rl1 loss in LSCs was verified after 7 doses of pIpC injection by flow cytometry (Supplementary Fig. 4B). At 6 weeks post-transplantation, there was no difference in the frequency of Il1rl1^f/f with Il1rl1^f/f Mx1Cre-derived donor cells in the peripheral blood (PB) and BM but at 12 and 20 weeks, we observed a significant decrease in hematopoietic cells chimerism in recipients of PB and BM from Il1rl1^f/f Mx1Cre mice compared to Il1rl1^f/f controls (Supplementary Fig. 4C, D). Further, analysis of the hematopoietic stem and progenitor cells (HSPCs) compartment revealed the reductions in LSK, long-term HSCs, short-term HSCs, and GMPs in Il1rl1^f/f Mx1Cre recipients compared to Il1rl1^f/f controls at 12 weeks post-transplantation (Supplementary Fig. 4E−H). However, we did not observe significant changes in the frequencies of multipotent progenitors (MPPs), CMPs, and megakaryocyte-erythroid progenitors (MEPs) (Supplementary Fig. 4I−K). These data indicates that Il1rl1 deficiency impairs steady-state hematopoiesis.

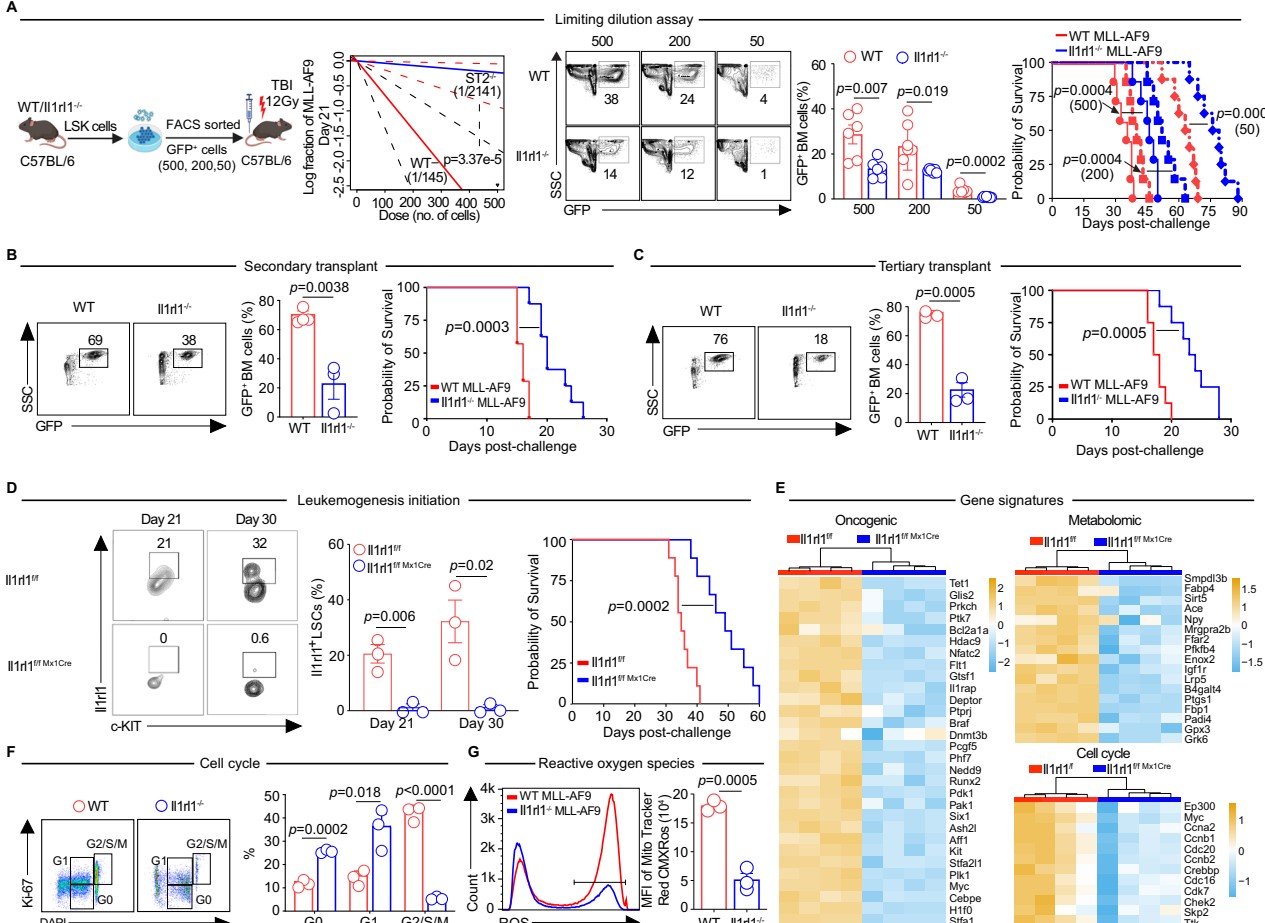

**Fig. 2 | Il1rl1 is required for leukemogenesis initiation and LSCs self-renewal.**
**A** Limiting dilution assays were performed to assess leukemogenic potential in WT and Il1rl1−/− mice using the MLL-AF9 AML model. Estimated LSC frequencies at day 21 post-transplant were calculated using ELDA from mice transplanted with 500, 200, or 50 MLL-AF9 LSCs (n = 8/group). Representative flow cytometry plots and quantification of GFP+ BM cells are shown (n = 6–8/group). Kaplan-Meier survival curves for mice transplanted with sorted WT or Il1rl1−/− LSCs (500, 200, or 50 cells) are presented. **B** Secondary transplantation with 0.5 × 10⁶ BM cells from primary leukemic mice was performed. Shown are representative flow plots and GFP+BM cell frequencies at day 15 (n = 3/group), along with survival curves of secondary recipients. **C** Tertiary transplantation was conducted using BM from secondary recipients. GFP+BM frequencies at day 15 and survival curves of tertiary recipients

are shown (n = 3/group). **D** Inducible deletion of Il1rl1 under the Mx1 promoter was evaluated post-AML challenge. Il1rl1 expression and GFP+LSC frequencies in Il1rl1f/f and Il1rl1f/f Mx1Cre mice were assessed at days 21 and 30 after pIpC induction (n = 3/group). Survival curves of transplanted mice are shown. **E** Heatmaps show differential expression of oncogenic, metabolic, and cell cycle-related genes in Il1rl1f/f and Il1rl1f/f Mx1Cre LSCs (log₂-transformed). **F** Cell cycle analysis of GFP+ BM cells in Il1rl1f/f and Il1rl1f/f Mx1Cre LSCs recipients on day 14 post-pIpC shows representative plots and distributions across G0, G1, and G2/S/M phases (n = 3/group). **G** ROS levels in GFP+ BM cells in Il1rl1f/f and Il1rl1f/f Mx1Cre LSCs recipients at day 14 post-pIpC are shown as histograms and quantified by mean fluorescence intensity (n = 3/group). Data are presented as mean ± SEM. Statistical analyses were performed using unpaired two-sided t-test (**A**–**D**, **F**, **G**) or log-rank tests for Kaplan-Meier survival.

Reduced engraftment potential of Il1rl1−/− BM cells could also be due to the altered homing capacity of donor cells. We performed homing assay and found decreased frequencies of donor-derived LSK, CXCR4+ Lineage-negative HSCs and CXCR4+LSK in the BM of recipient mice repopulated with Il1rl1−/− vs WT BM cells (Supplementary Fig. 5A–D). This might indicate that Il1rl1 deficiency results in a competitive disadvantage to HSPCs post-transplantation.

### Il1rl1 loss under inducible hematopoietic-specific Mx1 promoter shapes leukemogenesis

To further interrogate the role of Il1rl1 in MLL-AF9 induced leukemia initiation, we transduced LSK cells from Il1rl1f/f and Il1rl1f/f Mx1Cre mice with the MLL-AF9 retrovirus system as previously[17]. Il1rl1f/f and Il1rl1f/f Mx1Cre LSCs were transplanted into lethally irradiated recipients. We found leukemia burden to be lower in Il1rl1f/fMx1Cre vs Il1rl1f/f controls (Supplementary Fig. 6A). The frequencies of LSCs were also decreased over time in Il1rl1f/f Mx1Cre recipients (Supplementary Fig. 6B, gating strategy in Supplementary Fig. 2B). Of note, in Il1rl1f/f recipients, Il1rl1 expression on LSCs was elevated at day 21 and day 30 post-challenge

indicating that Il1rl1 expression was inducible along with the disease progression (Fig. 2D), and Il1rl1 expression is much higher in LSCs as compared to its expression in normal murine HSCs, which is extremely low (Supplementary Fig. 7A). Meanwhile, Il1rl1 expression was elevated in LSPCs including GMP, CMP, and MPP as compared to normal murine HSCs, but not MEP (Supplementary Fig. 7B). Loss of Il1rl1 in LSCs using the Mx1-Cre system significantly inhibited leukemic growth resulting in prolonged survival as compared to Il1rl1f/f controls (Fig. 2D). These data suggest that Il1rl1 shapes steady-state hematopoiesis as well as promotes leukemogenesis in MLL-AF9 AML.

### Il1rl1 reprograms oncogenic, cell cycle, oxidative stress, and metabolomic signatures in LSCs

Since MLL-AF9 has been associated with increased expression of oncogenic target genes[36], we examined if loss of Il1rl1 in MLL-AF9 LSCs could reprogram these transcriptional profiles. To that end, we performed RNA-sequencing of malignant BM-derived Il1rl1f/f vs Il1rl1f/f Mx1Cre LSCs and identified differentially expressed genes (DEGs) analysis (DEGs, listed in Supplementary Data 1). We first created heatmap of

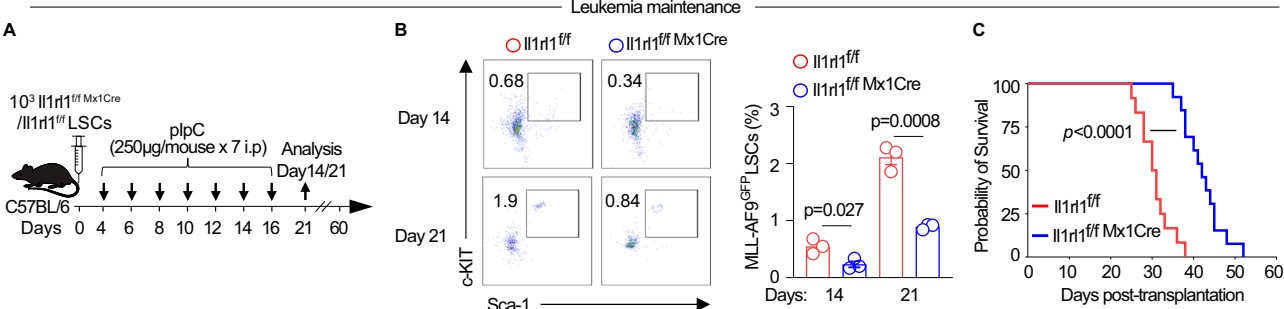

**Fig. 3 | Il1rl1 is required for leukemia maintenance. A** Experimental schema for investigating the role of Il1rl1 in murine MLL-AF9 leukemia maintenance. Il1rl1 WT or Il1rl1$^{-/-}$ LSCs were sorted and defined as MLL-AF9$^{GFP}$Lineage negative, c-Kit$^+$Sca-1$^+$. **B** Representative flow plots and the frequencies presented as mean values ± SEM of LSCs in the BM of Il1rl1$^{f/f}$ vs Il1rl1$^{f/f\ Mx1Cre}$ LSCs transferred mice on day 14 ($n = 3$) and day 21 ($n = 3$). pIpC was administered on day 4 after the leukemia was established, unpaired two-sided t-test. **C** Kaplan-Meier survival curves of mice transplanted with Il1rl1$^{f/f}$ ($n = 12$) vs Il1rl1$^{f/f\ Mx1Cre}$ ($n = 13$) LSCs with pIpC administration on day 4 after the leukemia was established. Log-rank test was used.

known oncogenic genes and observed that several of these genes such as Tet1[37], Flt1[38], Dnmt3b[39], Six1[40], IL1RAP[41], and Kit[42] were decreased in Il1rl1$^{f/f\ Mx1cre}$ LSCs vs Il1rl1$^{f/f}$ controls (Fig. 2E, Supplementary Table 2). The cell-cycle has been shown to play a crucial role in curbing DNA damage, sustaining self-renewal capabilities and enhancing mitochondrial reactive oxygen species (ROS) release in LSCs[43,44]. Here, we found reduced expression of positive regulators of cell cycle including Ccna2, Cnnb1, Cdc16 and Cdk7 in Il1rl1 deficient LSCs in contrast to control LSCs (Fig. 2E, Supplementary Table 3) indicating that Il1rl1 deficiency in LSCs may inhibit LSCs proliferation by limiting their division. This is in line with our cell cycle analysis showing that Il1rl1 deficiency in leukemic cells arrests G2/S/M cell cycle progression (Fig. 2F). We also interrogated metabolic signatures and found that several genes involved in ROS release induction such as SMPDL3b, FABP4, Ace, Igf1r, and Fbp1 were decreased in Il1rl1$^{f/f\ Mx1Cre}$ LSCs as compared to Il1rl1$^{f/f}$ control LSCs (Fig. 2E, Supplementary Table 4). All these signatures related to oncogenesis, metabolism, and the cell cycle in leukemia have been previously investigated (Supplementary Table 5). Thus, we measured the mitochondrial ROS levels in Il1rl1$^{-/-}$ MLL-AF9 leukemic cells and confirmed they were decreased as compared to Il1rl1 wild type leukemic cells (Fig. 2G). In addition, mitochondrial potential in Il1rl1$^{-/-}$ MLL-AF9 leukemic cells are diminished in contrast to WT leukemic cells (Supplementary Fig. 7C). We, additionally, performed gene functional enrichment analysis of the DEGs, which revealed that the up-regulated genes in WT LSCs were associated with cell differentiation, adhesion, oxidative stress, and the regulation of hematopoiesis. In contrast, the down-regulated genes were linked to immune system processes, inflammatory responses, cytokine production, and leukocyte migration, as depicted in Supplementary Fig. 8A, B.

## Il1rl1 is required for leukemia maintenance
To determine the role of Il1rl1 in leukemia maintenance, we established primary leukemia in immunocompetent mice by injecting 10$^3$ LSCs from Il1rl1$^{f/f\ Mx1Cre}$ or Il1rl1$^{f/f}$ MLL-AF9$^{GFP}$ and waited 4 days for the leukemia to be engrafted. After the leukemia was established, we started pIpC administration with injection every other day for a total of 7 doses (Fig. 3A). Il1rl1 deletion in Il1rl1$^{f/f\ Mx1Cre}$ LSCs was confirmed by flow cytometry staining at day 14 post-pIpC administration as compared to the leukemia-bearing mice pre-pIpC injection at day 4 (Supplementary Fig. 9). We observed that the leukemia burden was no different in Il1rl1$^{f/f\ Mx1Cre}$ or Il1rl1$^{f/f}$ MLL-AF9$^{GFP}$ LSCs transplanted mice prior to pIpC administration (Supplementary Fig. 10A). Loss of Il1rl1 in LSCs from Il1rl1$^{f/f\ Mx1Cre}$ mice decreased the total BM leukemia burden and leukemic cells proliferation vs Il1rl1$^{f/f}$ LSCs transplanted mice at day 14 and 21 post-challenge following pIpC injection (Supplementary Fig. 10A). Furthermore, frequencies of LSCs, and their proliferation were

decreased in Il1rl1$^{f/f\ Mx1Cre}$ vs Il1rl1$^{f/f}$ LSCs transplanted mice while apoptosis was increased (Fig. 3B, Supplementary Fig. 10B, C). This resulted in extended survival in Il1rl1$^{f/f\ Mx1Cre}$ LSCs recipient mice as compared with Il1rl1$^{f/f}$ controls (Fig. 3C). Cell cycle analysis showed over 80% decrease in the percentage of leukemic cells in G2/S/M phase (Supplementary Fig. 10D) and decreased in ROS levels in Il1rl1$^{f/f\ Mx1Cre}$ LSCs recipient mice vs Il1rl1$^{f/f}$ controls (Supplementary Fig. 10E). Taken together, these findings suggest that Il1rl1 is required for the sustained growth of MLL-AF9 leukemic cells, thereby contributing to their maintenance.

## HSCs initiate an IL-33/Il1rl1 autocrine loop during stress
Since Il1rl1 expression increased in LSCs over time and has previously been shown to be a positive regulator in HSCs[12], we hypothesized that IL-33 may increase in normal HSCs during stress. To test this hypothesis, we induced myeloablative stress in normal HSCs with a single dose of 5-FU and monitored expression of IL-33 using C57BL/6 IL-33$^{GFP}$ reporter mice and its receptor Il1rl1 by flow cytometry. IL-33 was increased in HSCs as well as in long-term HSCs following 5-FU induced stress versus without stress at day 3 and day 5 post-5-FU/vehicle administration (Fig. 4A, and Supplementary Fig. 11A). In parallel, there was a discernible elevation in the expression of Il1rl1, noted in both HSCs and long-term HSCs on both day 3 and day 5 of the experimental period (Fig. 4A, and Supplementary Fig. 11A, B). Of note, Il1rl1 and IL-33 double positive HSCs and LT-HSCs were also elevated during hematopoietic stress at day 3 and day 5 as compared to their counterparts without stress (Fig. 4A, and Supplementary Fig. 11A, B). These data indicated that IL-33/Il1rl1 autocrine loop was amplified in response to hematopoeitc stress.

## An IL-33/Il1rl1 autocrine loop in LSCs remodels stem cells to promote leukemia growth
To investigate if an autoloop between secreted IL-33 and its surface receptor Il1rl1 was established in LSCs, we used IL-33$^{cit/cit\ KO}$ mice previously reported[45]. Briefly, a 7 kb genome DNA fragment centered on the IL-33 start codon in exon 2 was amplified and cloned into a Bluescript SK-plasmid. Then, a fluorescent citrine cassette was inserted to the IL-33 gene after the start codon by removing 113 bp of this gene. Chimeric mice with successful germline transmission were bred with BALB/c Cre-deletor mice to excise the neomycin resistance gene. Finally, IL-33$^{cit/+}$ mice were interbred to generate C57BL/6 IL-33$^{cit/cit}$ functional knockout mice. C57BL/6 IL-33$^{cit/cit}$ knockout mice were obtained by interbreeding over 20 generations[45]. Then, we generated IL-33$^{cit/cit\ KO}$ and IL-33$^{WT}$ MLL-AF9$^{gfp}$ LSCs and transplanted 500 IL-33$^{WT}$ or IL-33$^{cit/cit\ KO}$ LSCs along with IL-33$^{cit/cit\ KO}$ BM supporting cells into lethally irradiated IL-33$^{cit/cit\ KO}$ recipients (Fig. 4B). In this design, the sole source of IL-33 comes from the LSCs in the experiment with IL-33$^{WT}$ MLL-AF9$^{gfp}$ LSCs since all other sources (BM cells and stromal cells) are deficient in IL-33. IL-33$^{cit/cit\ KO}$

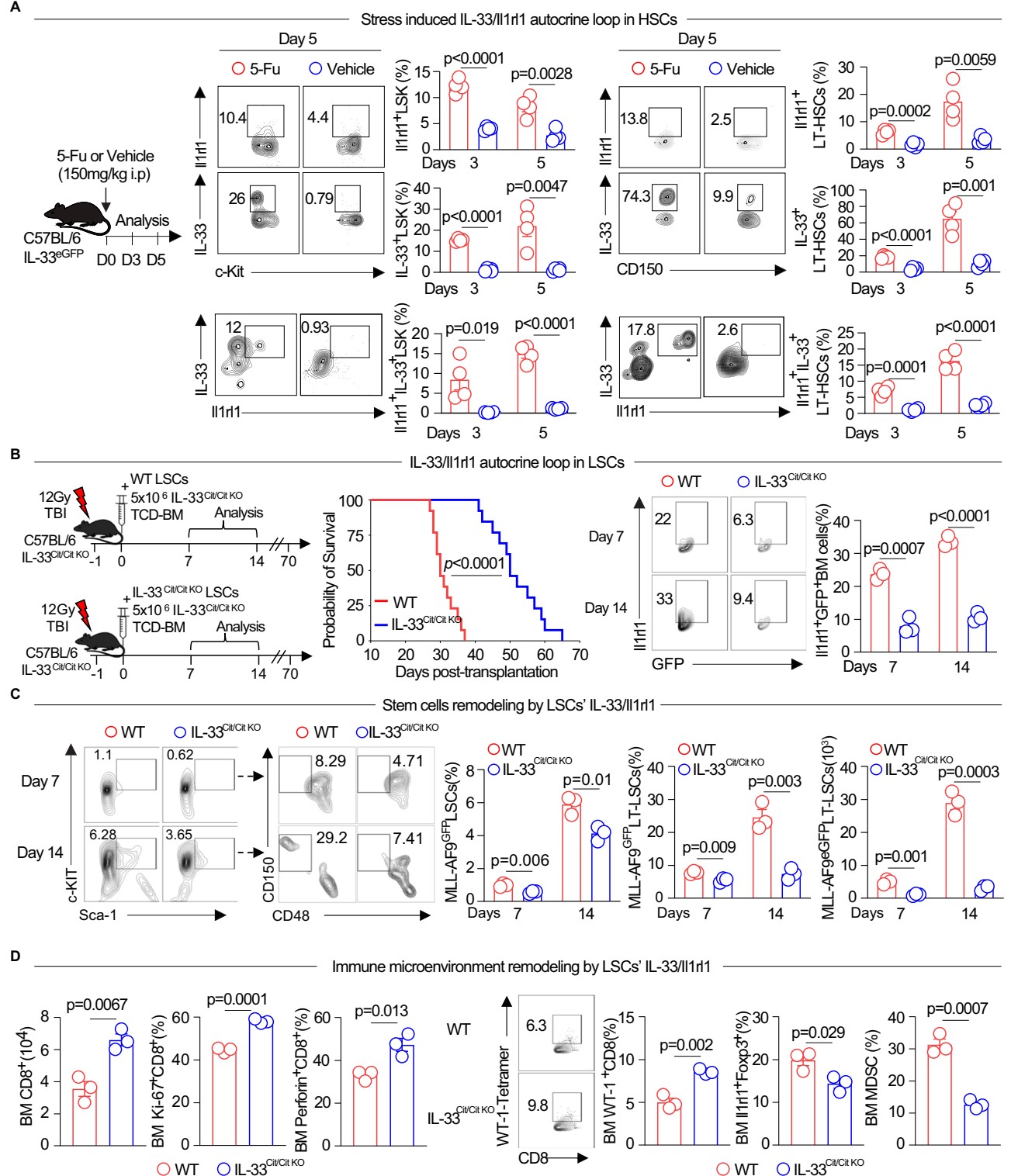

**Fig. 4 | Hematopoietic stress induces an IL-33/Il1rl1 signaling autocrine loop in HSCs and LSCs in the healthy and leukemic BM niches. A** Hematopoietic stress-induced IL-33/Il1rl1 loop in normal HSCs. Schematic of 5-FU administration in IL-33GFP reporter mice to assess Il1rl1 and IL-33 expression in LSK and long-term HSCs. Representative flow plots and quantification (mean ± SEM) of Il1rl1 and/or IL-33 expression at days 3 and 5 post-5-FU or vehicle injection (*n* = 4/group). Unpaired two-sided *t*-test was used. **B** IL-33/Il1rl1 autocrine loop in LSCs. Schematic of IL-33 deficiency in MLL-AF9GFP LSCs using IL-33Cit/Cit KO mice. Mice received 500 IL-33WT or IL-33Cit/Cit KO MLL-AF9GFP LSCs with IL-33Cit/Cit KO supporting BM cells. Kaplan-Meier survival analysis compared recipient mice (log-rank test). Flow plots and frequencies (mean ± SEM) of Il1rl1+GFP+BM cells at days 7 and 14 post-transplant (*n* = 3/

group), analyzed by unpaired two-sided *t*-test. **C** Stem cell remodeling by the LSC-derived IL-33/Il1rl1 loop. Representative plots and quantification (mean ± SEM) of total LSCs, long-term LSC frequencies, and absolute cell numbers in mice transplanted with IL-33WT vs IL-33Cit/Cit KO MLL-AF9GFP LSCs over time (*n* = 3/group). Unpaired two-sided *t*-test was used. **D** Immune microenvironment remodeling by the LSC-derived IL-33/Il1rl1 loop. Absolute numbers and frequencies (mean ± SEM) of CD8+T cells, Ki-67+CD8+T cells, perforin+CD8+T cells, WT1+CD8+T cells (with representative plots), Il1rl1+Foxp3+Tregs, and MDSCs at day 14 post-transplant in IL-33WT vs IL-33Cit/Cit KO MLL-AF9GFP LSC recipients (*n* = 3/group). Unpaired two-sided *t*-test.

LSCs transplanted recipients had better survival than their IL-33[WT] LSC counterparts (65 vs 36 days, $p < 0.0001$) (Fig. 4B). Frequencies of total leukemic cells in IL-33[cit/cit KO] LSCs recipients were significantly decreased compared to IL-33[WT] recipients at day 7 and day 14 following leukemia induction (not shown). Il1rl1 expression was increased in leukemic cells from IL-33[WT] in comparison to those from IL-33[cit/cit KO] counterparts throughout the progression of the disease (Fig. 4B). Frequencies of LSCs and long-term LSCs at days 7 and 14 were also reduced in IL-33[cit/cit KO] LSCs vs IL-33[WT] transferred recipients, as well as absolute numbers of long-term LSCs (Fig. 4C).

## LSCs IL-33/Il1rl1 autocrine loop induces tolerance of the immune microenvironment

We further investigated the impact of this IL-33/Il1rl1 autocrine loop in LSCs on the immune leukemic microenvironment using the same model of IL-33[cit/cit KO] and IL-33[WT] MLL-AF9[gfp] LSCs transplanted with IL-33[cit/cit KO] BM supporting cells into IL-33[cit/cit KO] recipients. We found that the sole effect of IL-33 deficiency in LSCs doubled CD8[+]T cells counts and significantly increased their frequencies (Fig. 4D and Supplementary Fig. 12A) as compared to IL-33[WT] controls. To determine if changes in CD8[+] T cells are a result of IL-33 deficiency, we compared the T cell composition (CD4[+], CD8[+] T cells, and CD4[+]Foxp3[+]Tregs) in IL-33[cit/cit KO] vs IL-33 WT non-leukemic mice at steady state, and did not observed differences (Supplementary Fig. 13A–C). Furthermore, CD8[+] T cells in IL-33[cit/cit KO] LSCs BM niche showed less apoptosis and exhaustion while proliferating more in contrast to IL-33[WT] group (Fig. 4D and Supplementary Fig. 12B, C). Similarly, CD8[+] T cells' expression of CD107a, a degranulation marker, granzyme B, and perforin were higher in the IL-33[cit/cit KO] LSCs group than in IL-33[WT] LSCs group (Fig. 4D and Supplementary Fig. 12C). Additionally, CD44[+]CD62L[−] effector CD8[+] T cells and CD45RA[+]CCR7[−] central memory CD8[+] T cells were increased in IL-33[cit/cit KO] vs IL-33[WT] LSCs recipients (Supplementary Fig. 12D). Since MLL-AF9 leukemic cells express the WT-1 antigen, we further investigated the antigen specificity of CD8[+] T cells towards MLL-AF9 leukemic cells using tetramers recognizing WT-1 antigen[46,47]. As expected with the development of AML, the frequency of WT-1-tetramer[+]CD8[+] T cells progressively increased in the leukemic niche. However, in the BM niche of IL-33[cit/cit KO] LSCs recipient mice, WT-1-tetramer[+]CD8[+] T cells increased significantly more compared to IL-33[WT] counterparts (Fig. 4D) suggesting more antigen specific CD8[+] T cells are available to kill the tumor in recipients for which the LSCs IL-33/Il1rl1 autocrine loop in LSCs has been deactivated. We found that frequencies of total CD4[+] T cell helper cell and natural killer cells were not different in IL-33[cit/cit KO] vs IL-33[WT] LSCs niches (Supplementary Fig. 12E, F). Although frequencies of total CD4[+]Foxp3[+]Tregs were not different in IL-33[cit/cit KO] vs IL-33[WT] LSCs recipients, tissue-specific Il1rl1[+]Tregs, myeloid-derived suppressor cells (MDSCs), macrophages, and CD11c[+] cells, all susceptible to activation by the IL-33/Il1rl1 signaling pathway[4] were decreased in IL-33[cit/cit KO] vs IL-33[WT] LSCs recipients (Fig. 4D and Supplementary Fig. 12F). In addition, cell cycle analysis was consistent with Il1rl1[−/−] vs WT MLL-AF9[gfp] that IL-33 deficiency in MLL-AF9 leukemic cells arrests G2/S/M phase in contrast to IL-33[WT] leukemic cells (Supplementary Fig. 14A). We also observed the decrease in mitochondrial ROS and mitochondrial potential in IL-33[cit/cit KO] vs IL-33[WT] leukemic cells (Supplementary Fig. 14B, C), suggesting that IL-33 deficient MLL-AF9 leukemic cells are less mitochondrially stressed and show decreased proliferation. Together, these data suggest the dual role of the specific IL-33/Il1rl1 signaling triggering in LSCs that initiates leukemogenesis through a positive autoloop and amplifies tolerance of the immune microenvironment.

## Il1rl1 deficiency in murine MLL-AF9 LSCs and human leukemic cells inhibits their growth to improve survival

To investigate whether Il1rl1 deficiency in LSCs itself, in the absence of T cell immunity, parallels the effects observed with IL-33 deficiency in

MLL-AF9 LSCs, we performed an experiment involving myeloablative irradiation followed by transfer of either WT or Il1rl1[−/−] LSCs alongside T cell-depleted BM cells as supportive cells (Supplementary Fig. 15A). Our findings showed that mice transplanted with Il1rl1[−/−] LSCs and no immune cells exhibited prolonged survival compared to those transplanted with WT LSCs (Supplementary Fig. 15B). Il1rl1[−/−] LSC recipients also showed reduced leukemia burden during disease progression compared to WT LSC recipients (Supplementary Fig. 15B). Furthermore, we utilized CRISPR/Cas9 technology to deplete IL1RL1 in human MOLM14 cells (Supplementary Fig. 16A, B) and observed reduced proliferation and increased apoptosis in IL1RL1-deficient MOLM14 cells (Supplementary Fig. 16C, D). To assess the role of IL1RL1 in AML progression in vivo, we injected NOD.Cg-Prkdc[scid] Il2rg[tm1Wjl]/SzJ (NSG) mice with IL1RL1-CRISPR/Cas9 KO or WT MOLM14 cells without adoptive T cell transfer (Supplementary Fig. 17A). Mice transplanted with IL1RL1-deficient MOLM14 cells exhibited prolonged survival and lower leukemia burden (Supplementary Fig. 17B, C) compared to WT MOLM14 recipients at day 10 and 21 post-challenge. These data suggest that while reduced T cell tolerance contributed to improved survival in immunocompetent models, IL1RL1 deficiency in LSCs itself plays a critical role in suppressing AML growth.

## Il1rl1 as a therapeutic target for myeloid leukemic cells and regulatory immune cells using T cell engager bispecific antibodies (T-BsAbs)

For translational purposes in immune competent mice, we first tested murine anti-murine T cell engager bispecific antibodies (T-BsAb) binding CD3 and Il1rl1 using the sequence of murine anti-Il1rl1 blocking antibody (BC281)[48] and a Il1rl1 mutated sequence for a non-functional bispecific antibody (BC462) in immune competent mice (Fig. 5A). These T-BsAbs were built on the IgG[L]-scFv 2 + 2 format "da vinci" platform[49,50] with improved ability to drive T cells into human tumors for effective tumor ablation when compared to other T-BsAb platforms[51]. With two anti-CD3 arms each attached to the carboxyl end of each light chain in the Il1rl1-specific IgG, T cells are activated showing the classic immune synapse[49] without additional costimulatory activation as in CAR-T, and without simultaneous CD3 crosslinking to cause T cell exhaustion or death[52]. Fc function is also silenced by eliminating non-specific FcγR binding on stromal cells and antigen-presenting cells[53] to improve T-cell trafficking and reduce C1q complement activation[54]. Both T-BsAb showed >90% purity by HPLC (Fig. 5B). For a direct clinical translation, we also built an anti-human IL1RL1 system to study human T cells against human AML, using the same IgG[L]-scFv platform built from humanized anti-IL1RL1 blocking antibody[55], humanized anti-CD3 OKT3 (BC282) and its humanized control (BC283). Fc domain silencing was achieved by inducing N297A and K322A to remove FcR and C1q affinities (Fig. 5C). HPLC analysis verified their high purity (Fig. 5D). Surface plasmon resonance microscopy (SPR) analysis revealed strong antibody affinities with slow Koff values for the mIl1rl1 (BC281) and hIL1RL1 T-BsAb (BC282) (Fig. 5E). High stability was observed at 4 °C for over 6 weeks (Supplementary Fig. 18A).

To verify the absence of potential in vivo toxicity for the murine T-BsAb in immune competent mice, we injected BC281 at 0.4, 2, 5, 10 ug doses i.p q 3 days for a total of 6 doses in normal C57BL/6 mice; all mice survived, and animals did not exhibit any in vivo toxicity including body weight loss (Fig. 5F). Since in normal mice Il1rl1[+] Tregs are frequent in the gastrointestinal (GI) tract, we monitored potential GI toxicity by following body weight, and colon tract length which were unchanged in the BC281 treated group as compared to BC462 control group (Fig. 5G). To further evaluate the on-target, off-tumor toxicity, we tested a broad panel of normal human tissues by staining human anti-IL1RL1 antibody at two different concentrations. No staining was detected in any of the tissues for the anti-IL1RL1 T-BsAb (Supplementary Table 6), indicating that it does not interact with normal human tissues. To determine the dose to be used to test in

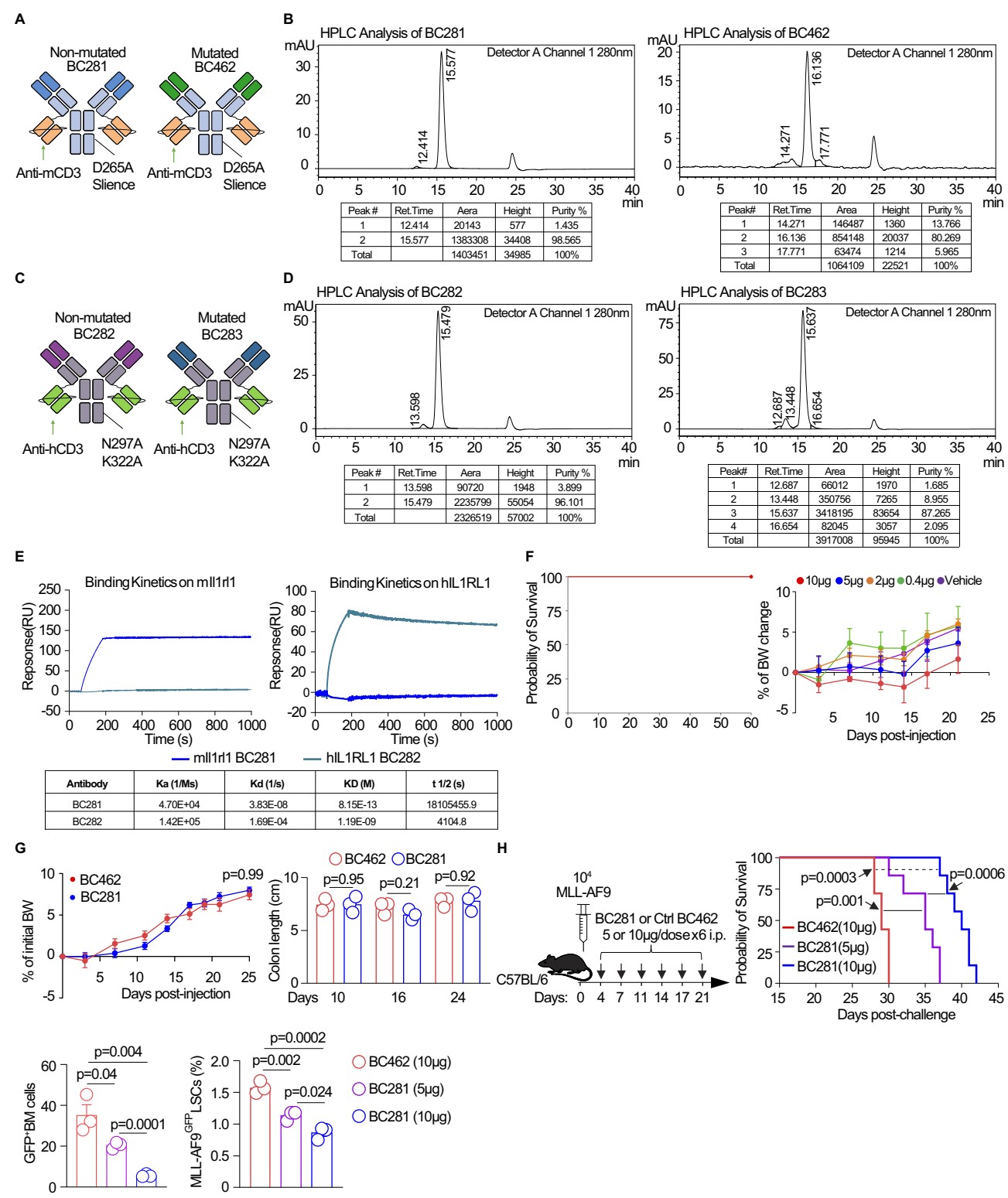

leukemia models, we performed a pharmacodynamic study for BC281 and BC282 (Supplementary Fig. 18B). Based on the maximum concentration (Cmax) and the area under the curve (AUCall), we administered a dose of 5 μg and 10 μg of BC281 vs BC462 per injection every 3 days for a total of 6 doses in the immunocompetent MLL-AF9 leukemia model (Fig. 5H). The in vivo data showed that mice treated with the two doses of BC281 had prolonged survival as compared with the BC462-treated control group, and the 10 μg dose led to a better overall survival than BC281 at 5 μg/injection with frequencies of total leukemic cells and LSCs being significantly decreased (Fig. 5H). Taken together,

these data showed that anti-Il1rl1 T-BsAb are stable, safe, and exhibit comparable pharmacokinetic to immunoglobulin G (IgG) forms in vitro, and in vivo in normal and AML mice.

### Anti-murine Il1rl1 T-BsAbs ± IL-15 superagonist reduces leukemia blasts, LSCs and reverses tumor microenvironment-mediated immune tolerance to extend survival in immunocompetent models of AML

T-BsAb that are genetically engineered to carry a second specificity for the T cell receptor through CD3 on T cells can be directed against Ilrl1[+]

**Fig. 5 | Development of functional and non-functional anti-Il1rl1 T-BsAbs.**
**A** Schematic structure of anti-murine Il1rl1 T-BsAbs (BC281) and mutated control
BC462 T-BsAbs. **B** Purity evaluated by HPLC of anti-murine Il1rl1 T-BsAb (BC281) and
control BC462. **C** Schematic structure of anti-human IL1RL1 T-BsAb (BC282) and
mutated control BC283. **D** Purity evaluated by HPLC of anti-human IL1RL1 T-BsAbs
(BC282) and control BC283. **E** Surface plasmon resonance (SPR) analysis of BC281
and BC282 binding kinetics. Interactive indexes between IL1RL1 antibodies and the
surface ligands included association (Ka), dissociation (Kd), equilibrium constant
(KD) and binding half-lives (t1/2) are shown. **F** Toxicity tests in vivo at different
dosages of anti-murine Il1rl1 T-BsAbs treatment in normal C57BL/6 mice. Kaplan-
Meier survival curves of mice injected at different dosages and body weight
changes presented as mean values ± SEM ($n = 5$/group). For survival analysis, log-
rank test was used. For the frequencies comparison ($n = 3$/per group), Unpaired

two-sided t-test was used. **G** Toxicity of anti-murine Il1rl1 T-BsAbs in the immuno-
competent MLL-AF9 leukemia model. Body weight change of mice treated with
BC281 and control BC462 ($n = 6$); colon length of mice treated with BC281 and
control BC462 on day 10, 16, 24 post-AML challenge ($n = 3$/timepoint). Data are
mean ± s.e.m. Unpaired two-sided t-test was used. **H** Experimental schema of anti-
murine Il1rl1 T-BsAbs BC281 and control BC462 treatment in immunocompetent
MLL-AF9 leukemia model; Kaplan-Meier survival curves of mice treated with
BC281(5 μg), BC281(10 μg), and control BC462 (10 μg), log-rank test was used for
survival ($n = 10$/group); Frequencies of MLL-AF9[gfp] total BM cells and LSCs in the BM
from mice treated with BC281(5 μg), BC281(10 μg), and control BC462 (10 μg) at Day
21 post-challenge, ANOVA with post-hoc Bonferroni t-test was used. Data are
mean ± s.e.m. ($n = 3$/group).

leukemic cells and Il1rl1[+] tolerogenic immune cells. Through activation
of the classic immune synapse, these CD3[+] T cells will release perforins
and granzymes that will kill Il1rl1[+] leukemic cells and Il1rl1[+] immune
cells. To first assess the direct effect of anti-Il1rl1 T-BsAb on leukemic
cells in vitro, BC281 was directly added to MLL-AF9[gfp] leukemic cells
without or with addition of T cells in a dose-dependent manner and the
killing capacity was monitored by SYTOX Blue stain. In vitro, BC281 at 6
and 16 h and at concentration up to 10 μg/ml showed a moderate
killing of MLL-AF9 cells, 10% and 25%, respectively when T cells are not
added (Supplementary Fig. 19A, B). A significant increase in killing of
leukemic cells was observed when BC281 was cocultured with CD8[+]
T cells (Fig. 6A). In vivo, we tested BC281 T-BsAb in the immuno-
competent MLL-AF9 leukemia model. Approximately 5% of MLL-AF9
leukemic cells express Il1rl1 at the time of the challenge (day 0). 10[4] of
these cells were injected at day 0 and 4 days later the first dose of
BC281 or BC462 control was given i.p every 3 days for a total of 6
injections (Fig. 6B). Mice treated with BC281 had extended survival as
compared to mice treated with BC462 (Fig. 6C). LSCs frequencies were
also decreased in the treated vs control group (Fig. 6D), accompanied
by decreased Il1rl1 expression in LSCs and total leukemic cells at day
10, 16, 24 post-treatment (Supplementary Fig. 20A, B). In addition, Il1rl1
expression in non-LSC leukemic progenitors including CMPs, MEPs,
MPPs, and GMPs was reduced in BC281 vs BC462 control treated leu-
kemic mice (Supplementary Fig. 21), suggesting that BC281 treatment
may decrease both LSCs and committed progenitors. We next asses-
sed the effect of anti-Il1rl1 T-BsAb on the immune microenvironment
particularly cytotoxic and antigen specific WT1[+]CD8[+] T cells as well as
Tregs. Analyses of BC281-treated mice compared to controls revealed
increased frequencies and counts of CD3[+]CD8[+] and WT1[+]CD8[+]T cells at
day 10, 16, and 24 (Fig. 6E, Supplementary Fig. 22A). CD3[+]CD4[+]T cells
counts did not differ (Supplementary Fig. 22B). Given the presence of
Tregs in the leukemic niche[16], we interrogated the effect of anti-Il1rl1 T-
BsAb on Il1rl1[+]Tregs. No direct cytotoxicity of BC281 towards
Il1rl1[+]Tregs was observed if CD8[+] T cells were not engaged but killing
was seen in the presence of BC281 as compared to BC462 control when
CD8[+]T cells were added in the coculture (Fig. 6F). Interestingly, fre-
quencies and absolute number of Il1rl1[+] Tregs were decreased in the
BC281-treated group vs BC462 control in the MLL-AF9 leukemia model
(Fig. 6F, Supplementary Fig. 22C). Because efficacy of T-BsAb is cor-
related to the total number of T cells and intratumoral T cells infil-
tration particularly of CD8[+] T cells[56], we investigated the BM niche
CD8[+] T cells to leukemic cells ratio over time in the MLL-AF9 leukemia
model without treatment, and found that the ratio sharply decreased
during AML progression with only 1 CD8[+] T cell out of 250 leukemic
cells at day 17 post-leukemia induction (Fig. 6G). As postulated, BC281
treatment increased the ratio CD8/AML by increasing CD8[+]T cells
counts and decreasing leukemic cells counts (Supplementary
Fig. 22D). In depth-analyses of the immune microenvironment showed
that frequencies of PD-1[+], LAG-3[+], TIM-3[+], and PD-1[+]TIM3[+] expressing
CD8[+]T cells decreased in BC281 vs BC462 control treated mice (Sup-
plementary Fig. 23A, B). Il1rl1 expression on this exhausted PD-

1[+]TIM3[+]CD8[+] T cells was elevated in the BC462 control group and
decreased after specific anti-Il1rl1 and CD3 engagement by BC281
(Supplementary Fig. 23C). Moreover, BC281 treatment increased
effector memory CD44[+]CD62L[-]CD8[+] T cells and central memory
TCF7[+]CD8[+]T cells compared to BC462 control treatment as well as
frequencies of CD27[+]CD8[+] T cells, CD28[+]CD8[+] T cells, Tbet[+]CD8[+]T cells,
GZMB-expressing CD8[+]T cells, degranulating CD107a[+]CD8[+] T cells,
perforin[+]CD8[+]T cells, and IFN-γ[+]CD8[+] T cells (Supplementary Fig. 23D).
In contrast, frequencies of IL-10[+], TGF-β[+], and CD39[+] Foxp3[+] Tregs were
decreased in BC281 treated group as compared to the BC462 control
group (Supplementary Fig. 23E). Finally, after T cell engagement of
BC281 vs BC462 control, Il1rl1 expression decreased in Foxp3[+]Tregs,
macrophages, myeloid-derived suppressor cells (MDSCs) and
CD11c[+]DCs (Supplementary Fig. 23F). To remedy the deficit of intra-
tumoral CD8[+] T cells and as a proof-of-principle study, we adoptively
transferred 10[6] polyclonal naïve CD8[+] T cells once a week in the BC281
or BC462 treated group (Supplementary Fig. 24A). We found that
weekly CD8[+] T cells transfer in addition to BC281 treatment extended
survival further than that of BC462 in combination with adoptive
transfer of CD8[+] T cells, or CD8[+]T cells adoptive transfer alone or
BC281 alone (Fig. 6H). Accordingly, frequencies of LSCs decreased in
the BC281 in combination with adoptive transfer of CD8[+] T cells group
compared to the BC462 + CD8[+] T cells group while frequencies of BM
infiltrating CD3[+]CD8[+] T cells and BM antigen-specific WT-1[+]CD8[+] T cells
increased (Fig. 6H). As above, Il1rl1[+]Tregs were reduced (Fig. 6H), and
Il1rl1 expression in leukemic cells and LSCs was reduced in BC281-
treated leukemic mice, either alone or in combination with adoptive
CD8[+] T cell transfer, compared to BC462-treated mice or those
receiving CD8[+]T cells alone or in combination with BC462 (Supple-
mentary Fig. 24B). For a cost-effective translation without cell manip-
ulation, we then combined ALT-803, which is an IL-15 superagonist that
promotes expansion of T cells[57] in vivo (Supplementary Fig. 25A–C),
with BC281 or BC462 in the MLL-AF9 leukemia model. BC281 treatment
in combination with ALT-803 extended survival and decreased leuke-
mia burden as compared to BC462 control plus ALT-803 or AL-T803
alone (Fig. 6I, Supplementary Fig. 26A). We also observed decreased
Il1rl1 expression in Tregs and the leukemic populations (Fig. 6I, Sup-
plementary Fig. 26B).

To further assess the impact of anti-murine Il1rl1 T-BsAb in an
alternative clinically relevant immunocompetent leukemia model,
we utilized the epigenetic DNMT3A/FLT3[ITD] leukemic model, which
frequently develops extramedullary leukemia in the spleen and
liver[58]. After allowing 3 weeks for engraftment, we initiated treatment
with BC462 and BC281 every 3 days for a total of 10 doses. Given the
aggressive nature of DNMT3A/FLT3[ITD] AML, we administered an
initial dose of 25 μg/mouse for the first three doses, followed by an
increased dose of 50 μg/mouse for the remaining seven doses (Sup-
plementary Fig. 27A). Following a 10-dose regimen, we analyzed the
effects of anti-murine Il1rl1 T-BsAb on the suppressive tumor
microenvironment and on the leukemic cells. Similar to findings in
the MLL-AF9 leukemia model, we observed that Il1rl1[+]Foxp3[+] Tregs

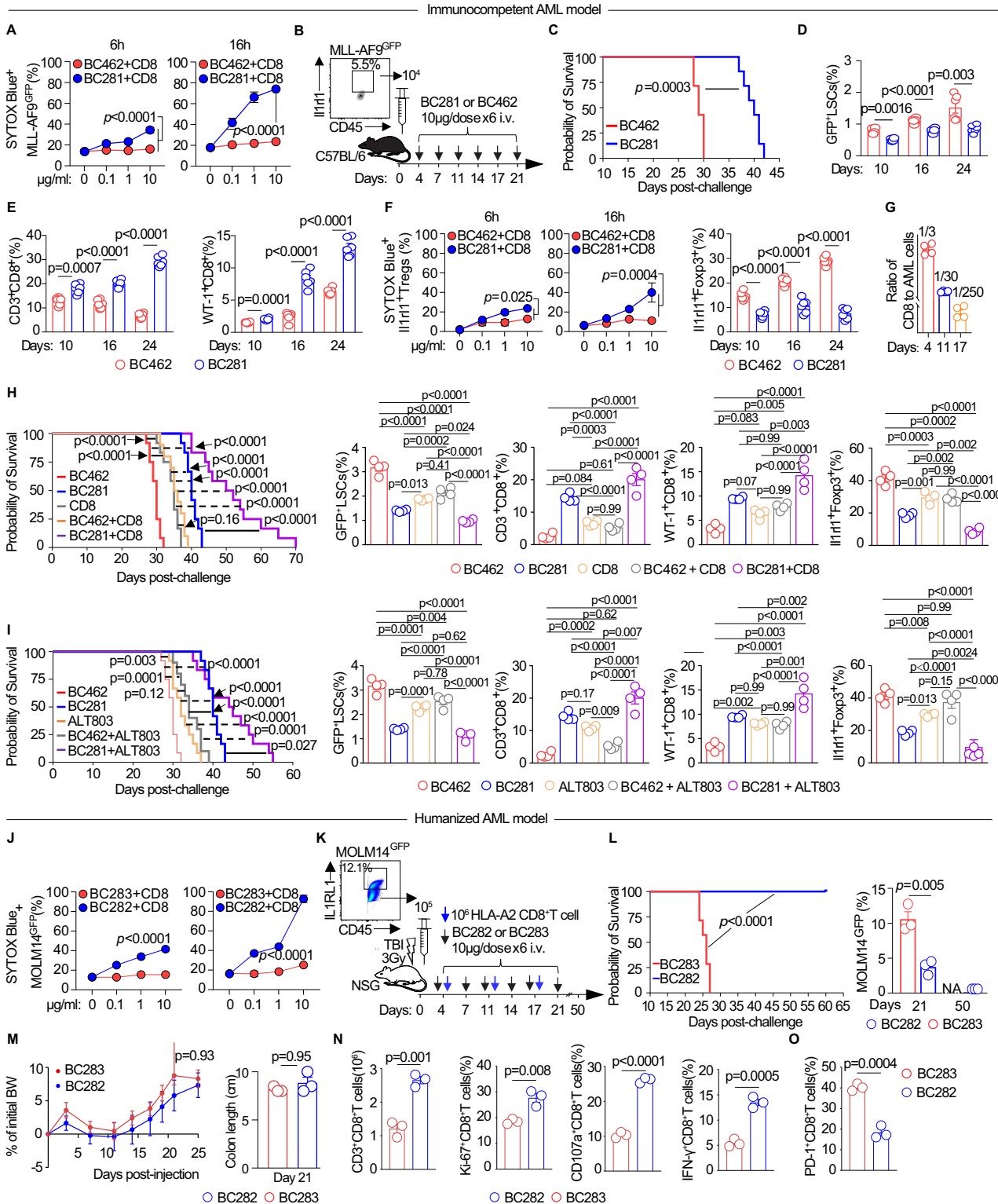

were significantly reduced in the spleen and liver of BC281-treated mice compared to those receiving BC462 (Supplementary Fig. 27B). Using confocal imaging of the spleen, we quantified the volume of Il1rl1+ cells and CD3+ T cells per field and found that in the BC281-treated group, Il1rl1+ cells were nearly eliminated, while CD3+ T cell infiltration was significantly increased compared to the BC462-treated controls (Supplementary Fig. 27C). BC281 treatment also led to a significant increase in the frequencies of CD8+ T cells, by flow cytometry, in both BM and spleen, relative to BC462-treated leukemic mice (Supplementary Fig. 27D). In parallel, frequencies of

exhausted PD-1+TIM3+CD8+ T cells were reduced in spleen and liver in the BC281-treated mice compared to BC462-treated controls (Supplementary Fig. 27E). BC281 treatment resulted in a significant reduction of percentage of AML blasts (c-Kit+CD11b+) in the BM (Supplementary Fig. 27F) as well as frequency of myeloid-derived suppressor cells (CD11b+Gr-1+) (Supplementary Fig. 27G). Taken together, these findings suggest that Il1rl1 T-BsAb exert a dual inhibition of Il1rl1+ leukemic cells and tolerogenic Il1rl1+ immune cells including Tregs leading to expansion of cytotoxic CD8+ T cells resulting in synergistic killing of AML cells as well as LSCs.

**Fig. 6 | Anti-Il1rl1 T-BsAbs eradicate leukemia and LSCs to extend survival in AML models. A** In vitro cytotoxicity of anti-Il1rl1 T-BsAbs BC281 vs. control BC462 against MLL-AF9$^{GFP}$ leukemic cells co-cultured with CD8$^+$T cells, measured by SYTOX Blue at 6 and 16 h and varying T-BsAb concentrations ($n = 3$ per concentration). Two-way ANOVA; data shown as mean ± SEM. **B** Il1rl1 expression on MLL-AF9$^{GFP}$ cells at transplantation and treatment schema in the immunocompetent AML model. **C** Kaplan-Meier survival curves of MLL-AF9$^{GFP}$ mice treated with BC281 or BC462 ($n = 7$/group). Log-rank test. Frequencies (mean ± SEM) of GFP$^+$LSCs (**D**) and CD3$^+$CD8$^+$, WT1$^+$CD8$^+$ T cells (**E**) in BM at days 10, 16, and 24 post-treatment ($n = 6$). Unpaired two-sided $t$-test. **F** In vitro cytotoxicity of anti-Il1rl1 T-BsAbs against Il1rl1$^+$Foxp3$^+$Tregs at 6 and 16 h; in vivo Tregs frequencies at days 10, 16, and 24 ($n = 6$) post-treatment. Two-way ANOVA. **G** The ratio of CD8$^+$T cell to leukemic cells in the malignant BM niche without treatment at days 4, 11, and 17 post-challenge ($n = 4$). **H, I** Kaplan-Meier survival and frequencies (mean ± SEM) of GFP$^+$LSCs, CD3$^+$CD8$^+$, WT1$^+$CD8$^+$ T cells, and Il1rl1$^+$Foxp3$^+$Tregs in mice treated with BC281, BC462, CD8$^+$T cells, ALT-803 or combinations ($n = 4$–12/group). Log-rank test for survival. One-way ANOVA for frequencies. **J** In vitro cytotoxicity of anti-human IL1RL1 T-BsAbs BC282 vs. control BC283 against MOLM14$^{GFP}$ cells co-cultured with CD8$^+$T cells at 16 h and various concentrations ($n = 3$). Two-way ANOVA. **K** IL1RL1 expression on MOLM14$^{GFP}$ cells and schema for humanized AML treatment model. **L** Kaplan-Meier survival and leukemic cell frequencies (mean ± SEM) in BC282 vs. BC283-treated MOLM14$^{GFP}$ mice at days 21 and 50 ($n = 3$–12/group). Log-rank test for survival and Unpaired two-sided $t$-test. **M** Body weight changes and colon length at day 21 post-treatment with BC282 or BC283 ($n = 3$–10). Data are mean ± s.e.m. Unpaired two-sided $t$-test for colon length and two-way ANOVA for body weight changes. **N, O** CD8$^+$T cell number, activation (Ki-67$^+$, CD107a$^+$, IFN-γ$^+$), and exhaustion (PD-1$^+$) marker frequencies in BM on day 21 post-treatment ($n = 3$/group). Data are mean ± s.e.m. Unpaired two-sided $t$-test.

## Anti-human IL1RL1 T-BsAb eradicates leukemia blasts, LSCs and increases type 1 CD8$^+$ T cells to extend survival in humanized models of AML

For clinical translational purpose, we tested the anti-human IL1RL1 T-BsAb and its control in humanized AML models. First, we assessed in vitro the direct effect of anti-IL1RL1 T-BsAb on human MOLM14 leukemic cells and found that, in line with the cytotoxicity observed with BC281 on MLL-AF9 cells, coculture of MOLM14 cells with CD8$^+$ T cells in the presence of BC282 exhibited potent killing activity against MOLM14 cells in a dose-dependent manner (Fig. 6J). We next modeled humanized leukemic mice with MOLM14$^{gfp}$ cells which expressed detectable IL1RL1 by flow cytometry in 12% of cells and treated them with weekly injection of human CD8$^+$ T cells in NSG mice. We employed a T-BsAb regimen like the one used in the immunocompetent leukemia model using BC282 and its control BC283 (Fig. 6K). To reduce GVL effect[59], we matched the CD8$^+$ T cells HLA-A2$^+$ type with MOLM14. Animals treated with BC282 exhibited improved survival 60 days post-AML compared to those treated with BC283. The frequency of MOLM14$^{gfp}$ cells was lower in the BC282 group at day 21 post-challenge, and in the surviving group treated with BC282, we found no detectable leukemia cells in the BM at day 50 post-challenge (Fig. 6L). There was no body weight loss and no toxicity in the GI tract (Fig. 6M). As in the immunocompetent murine model, the numbers of CD8$^+$ T cells as well as frequencies of proliferating CD8$^+$Ki67$^+$, CD8$^+$CD107a$^+$, and CD8$^+$IFNγ$^+$ T cells were increased in the BC282-treated mice vs BC283 control mice (Fig. 6N). The frequency of exhausted CD8$^+$ T cells was also significantly decreased in the BC282-treated mice vs BC283 control mice (Fig. 6O).

Overall, these findings indicated that anti-Il1rl1 T-BsAb using murine antibodies in an immunocompetent model and humanized antibodies in a human xenograft model effectively inhibit both leukemia growth and the tolerogenic immune microenvironment in immunocompetent and humanized pre-clinical models.

## IL1RL1 as a therapeutic target for T-BsAb in primary AML xenograft models

We further evaluated IL1RL1 expression in several previously developed patient-derived xenograft (PDX) models of pediatric AML using primary patient samples[60]. IL1RL1 expression was significantly elevated in 6 MLL oncofused pediatric-AML derived PDX that were tested as compared to 2 non-MLL oncofused prediatric-AML derived PDX (Supplementary Table 7, Fig. 7A). We first assessed the cytotoxicity of anti-IL1RL1 T-BsAb on both MLL-oncofused (NTPL-377luc, DF-5) and non-MLL (NTPL-301, NTPL-511) PDX cell lines in vitro. Co-culturing these PDX cells with CD8$^+$ T cells in the presence of BC282 induced significant cytotoxic activity against the PDX cells, compared to the BC283 control, in a dose-dependent manner. This effect was observed in both MLL- and non-MLL oncofused PDX cells (Supplementary Fig. 28A, B). Then, based on IL1RL1 fluorescence intensity and homogeneity of staining, NTPL-377 was used for the T-BsAb pre-clinical

in vivo testing. We intravenously injected $3 \times 10^5$ NTPL-377$^{luc}$ cells into NSG-B2m mice[61], and PDX engrafted mice were randomly assigned to the following treatment groups: untreated, human T cells alone, human T cells with anti-hIL1RL1 T-BsAb BC282, and T cells with control T-BsAb BC283 (Fig. 7B schema). Mice were treated using a similar regimen as in the cell line model with weekly injection of $5 \times 10^6$ activated human T cells. The bioluminescence imaging showed greater radiance in untreated, human T cell treated, and BC283 control treated engrafted mice compared to BC282 treated mice. Leukemia was not detectable within 5 weeks of BC282 administration compared with the other groups. We observed disease relapse at 6 weeks after the initial treatment and questioned if reinduction therapy with BC282 could further inhibit the relapsed leukemia. The mice were exposed to a second treatment cycle consisting of 2 weekly T cell injections and 4 biweekly T-BsAb (BC282 or BC283) injections starting at day 55 post-AML challenge (Supplementary Fig. 29A). Reinduction therapy extended survival in BC282 treated PDXs as compared to both BC283 control and solitary T cells treatment; nevertheless, it fell short of being a curative intervention (Supplementary Fig. 29B, C).

To closely mimic the clinical setting of rapid disease progression and enhance therapeutic efficacy in this PDX AML model, we utilized NSG-SGM3 mice. These mice express human IL-3, GM-CSF, and SCF, integrating the highly immunodeficient background of NSG mice with human cytokine support. This feature enables superior and stable engraftment of hematopoietic lineages and primary AML samples, as demonstrated in previous studies[62,63]. In this model, we combined anti-IL1RL1 T-BsAb with the Cheung's lab multimeric IL15/IL15Rα-Fc complex[64]. The same dose of NTPL-377$^{luc}$ cells was injected in NSG-SGM3 mice. BC282 treatment began on day 5 for a total of 7 doses and consisted of increased number of human T cells injections ($20 \times 10^6$/per dose) with IL15/IL15Rα-Fc complex (1 μg/dose) on days 5, 12, and 28 post-challenges (Fig. 7C). Bioluminescence imaging revealed higher radiance in untreated mice, as well as in those treated with BC283 control with T cells and BC282 alone vs mice treated with BC282 and T cells (Fig. 7C). A statistically significant enhancement in the median survival of mice was observed following treatment with a combination of BC282, T cells and IL15/IL15Rα-Fc complex (Fig. 7C). These data suggest that the anti-hIL1RL1 T-BsAb has potent activity against aggressive primary AML xenografts.

## Discussion

AML represents one of the most aggressive forms of blood malignancies characterized by a clonal expansion of immature myeloid cells with impaired differentiation in the BM[65]. Therapies for several blood cancers of lymphoid origin (i.e., ALL, NHL) have made remarkable progresses, while AML therapies have barely changed over the past 30 years. Although some advances in AML therapy and higher rates of complete remission have been obtained with molecular targeted therapies, many patients still experience relapse and die from AML, which remains the 6$^{th}$ most common cause of cancer death[66]. AML

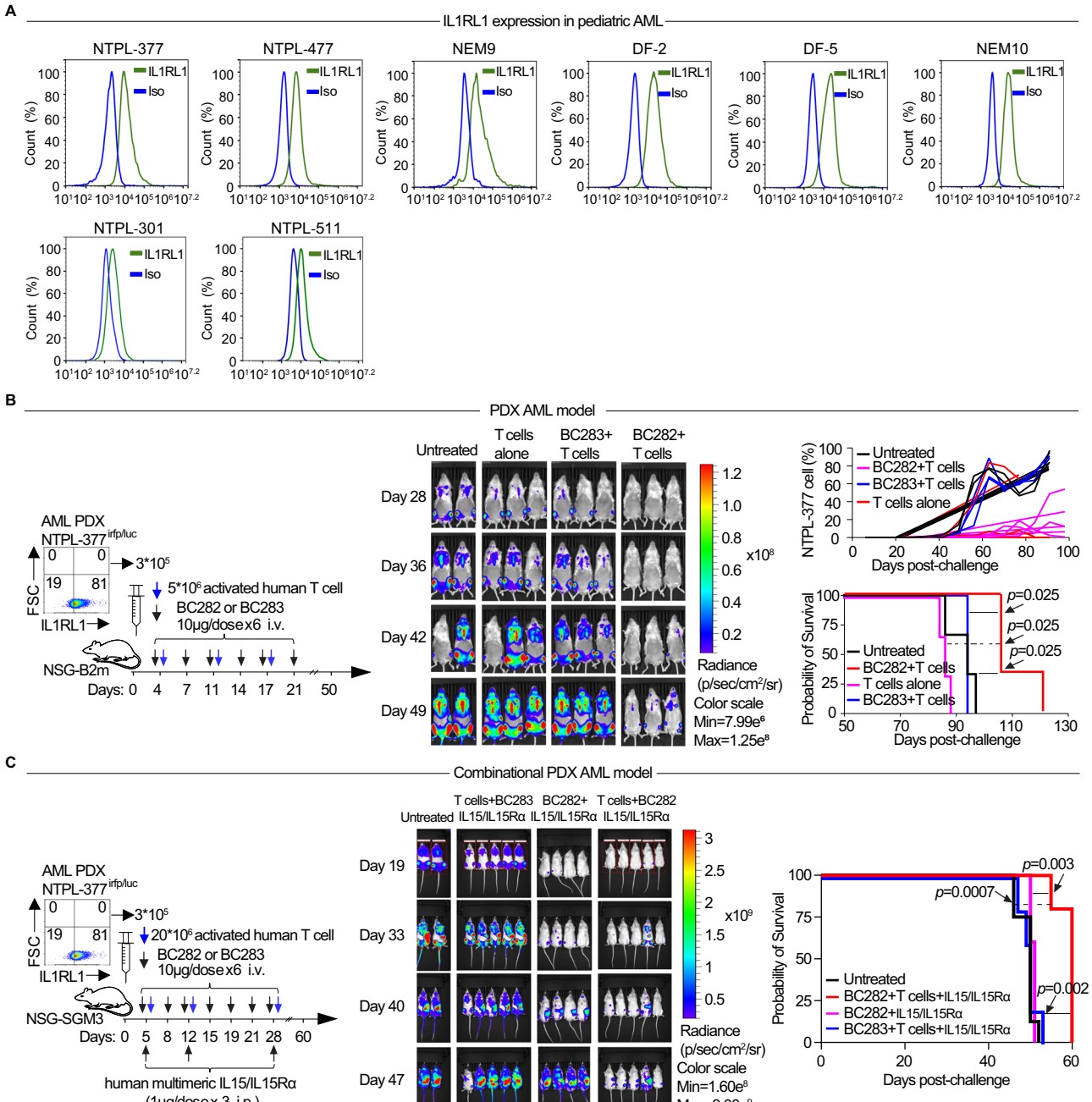

**Fig. 7 | Anti-human IL1RL1 T-BsAbs show potent activity against primary AML xenografts. A** Histograms of IL1RL1 expression vs isotype control on eight pediatric AML PDX cells, six with MLL oncofusion and 2 PDX AML cells with non-MLL oncofusion (NPTL-301 and NPTL-511) via flow cytometry. **B** Representative plot of IL1RL1 expression on AML pediatric PDX NTPL-377[irfp/luc] at the time of transplantation, and schema of anti-human IL1RL1 T-BsAb treatment in PDX AML model; Radiance imaging of AML PDX mice in untreated, adoptive transfer of human T cells, BC283 + adoptive transfer of human T cells, and BC282 + adoptive transfer of human T cells treated groups at days 28, 36, 42, and 49 post-treatment; Frequencies of NTPL-377[irfp/luc] leukemic cells over time in the 4 groups; Kaplan Meier

survival curves of NTPL-377[irfp/luc] PDX mice in the 4 groups. Log-rank test was used. **C** Anti-hIL1RL1 T-BsAb in combination with IL-15 treatment in PDX AML model. Experimental scheme of anti-hIL1RL1 T-BsAb treatment in combination with IL-15 in PDX AML model; Radiance imaging of leukemia bearing mice in untreated, T cells +BC283+hIL-15, BC82 alone+hIL-15, and T cells+BC282+hIL-15 treated groups post treatment. Data are mean ± s.e.m. (*n* = 5–8). **C** Survival analysis of NTPL-377[irfp/luc] cells bearing NSG mice untreated, and treated with T cells +BC283+hIL-15, BC282 alone+hIL-15, and T cells+BC282+hIL-15. Log-rank test was used for survival analysis (*n* = 5–8).

poses a significant challenge in its treatment primarily due to refractoriness and relapse following chemotherapy induction. Even with high remission rates, therapy resistance and relapse are often appearing and are the major obstacles to a cure. The origin of relapse has been attributed to a small subset of leukemic cells possessing long-term self-renewal capacity, identified as LSCs which reside at the apex of the hierarchy[67]. Similar to normal HSCs counterparts, LSCs are

subject to regulation by both intrinsic and extrinsic factors deriving from the stem cells themselves and the stem cell-derived niche situated in the BM[68]. This intricate interplay between intrinsic and extrinsic factors underscores the importance of understanding the microenvironment in which these cells reside. Targeting both the intrinsic and extrinsic regulatory mechanisms can provide a comprehensive approach to effectively control the behavior of LSCs, offering potential

avenues for improved therapeutic interventions in AML treatment. We and others have shown that the IL-33/Il1rl1 signaling pathway plays HSCs self-renewal and cancer-promoting roles[5–13,17]. However, its specific role in LSCs and the immune leukemia microenvironment of MLL-AF9 induced AML remains unclear. In this study, we have demonstrated that the IL-33 receptor Il1rl1 has a distinct impact on the self-renewal of LSCs. Notably, high IL1RL1 expression in AML correlated with refractoriness and poor prognosis. Loss of Il1rl1 by MLL-AF9-derived LSCs led to a reduction in their frequencies. This effect is attributed to the disruption of a stress-induced autocrine loop of IL-33/Il1rl1 signaling in LSCs. This LSCs induced-IL-33/Il1rl1 loop generated an immune regulatory microenvironment that could be blocked as well. Impediment of this IL-33/Il1rl1 signaling by T-BsAb allowed for dual targeting of LSCs and regulatory immune microenvironment cells to reduce leukemogenesis initiation and maintenance and improve survival. These findings highlight Il1rl1 as a potential therapeutic target in AML.

The observation that patients with high IL1RL1 expression exhibit lower survival rates in comparison to those with low IL1RL1 expression prompted us to investigate its expression in different molecular genetic subtypes of leukemia such as MLL-AF9 fusion. Our analysis indicated that IL1RL1 was elevated in all these subtypes in contrast to healthy BM cells suggesting a global inflammatory process rather than a specific molecular process. The alarmin IL-33 is a member of the IL-1 cytokine family that is released upon cellular stress and involved in mostly type 2 and regulatory immune responses[3]. Previously, it has been shown that another pleiotropic cytokine, IL-1, promotes the expansion of HSCs and leukemia cells[69]. Patients with AML treated with the IL-1 inhibitor anakinra for severe COVID19 pneumonia have shown promising responses of their AML[70]. However, results of large, randomized trials are not available and evidence-based treatment protocols have yet to be established. Since the discovery of the major role of IL-1 in AML, other cytokines such as IFNγ, IL-3, IL-6, IL-8, and osteopontin have been shown to support AML cell growth[71]. IL-33, released by stromal and tumoral cells, has been identified as a crucial factor in the tumorigenesis of myeloproliferative neoplasms and solid tumors, including colorectal cancers, gastric cancer, and squamous cell carcinomas[6,8,10,11]. These studies have highlighted its involvement in promoting cancer progression through mechanisms such as the IL-33-TGFβ feedforward loop in squamous cell carcinoma and the synergistic partnership between IL-33/IL1RL1 and the Wnt pathway in gastric cancer stemness and metastasis. In the context of AML as well as chronic myeloid leukemia, the IL-33/IL1RL1 axis has been involved in only two reports. Using the cbfb-MYH knock-in mice, Wang et al. identified that Il1rl1 is dynamically expressed in LSCs, and conditional knockout of Il1rl1 in Cbfb-MYH11+ leukemia under an inducible hematopoietic-specific Mx1 Cre model confirmed the role of Il1rl1 in initiating leukemogenesis in the leukemia model[7]. Il1rl1 has also been found to play a role in maintaining and expanding LSCs associated with specific genetic alterations (AML1/ETO, DEK/NUP214, and BCR/ABL1), and through the counter-regulation of Wnt and Notch pathways in LSCs, where Wnt is triggered and Notch is reduced as compared to healthy HSCs[23]. In our multiple analyses, IL1RL1 expression was not driven by unfavorable cytogenetics, and hematopoietic stress/changes in cell cycle were found to be an initial culprit of an IL1RL1/IL-33 autoloop in HSCs and LSCs. This suggests that the alarmin IL1RL1 may behave as a stress checkpoint that may prevent DNA damage and genome instability. Il1rl1 deficiency also resulted in a decrease in its homing capacity (Supplementary Fig. 5A–D). This underscores the significance of the IL-33/Il1rl1 axis including downstream pathways in driving key characteristics (maintenance, proliferation, homing, and self-renewal) of leukemic progenitors and LSCs[72–74].

In our study, examination of stem cells infiltrates in AML showed higher frequencies of LSCs in patients with elevated IL1RL1 expression. Further investigation of IL1RL1 expression confirmed its upregulation in LSCs from NR patients after chemotherapy induction. LSCs are most functionally defined by their ability to initiate and maintain AML[67]. Thus, we hypothesized that Il1rl1 is required for leukemogenesis in the aggressive retrovirally induced MLL-AF9 induced AML. The impact of Il1rl1 loss in LSCs was substantial, as evidenced by a ~15-fold decrease in stemness potential compared to their wild-type counterparts via limited dilution assays. Serial transplantations further underscored the functional requirement of Il1rl1 in initiating myeloid leukemia and revealing that Il1rl1 deficiency in LSCs significantly reduced the ability of leukemogenesis initiation, as evidenced by lower leukemic cells and LSCs infiltrates in the BM compared to wild-type leukemic cells and LSCs. Moreover, a decrease in intracellular mitochondrial ROS levels was observed in Il1rl1-deficient leukemic cells, highlighting the potential role of Il1rl1 in regulating ROS and its implications in sustaining stem cell function[75]. Loss of Il1rl1 in established leukemia also prolonged survival of Il1rl1f/f Mx1Cre mice compared with control mice, this model better mimics potential therapeutic interventions of targeting Il1rl1. To address if Il1rl1 reprograms the transcriptional profile of LSCs, RNA-sequencing of Il1rl1f/f vs Il1rl1f/f Mx1Cre was performed and showed that the loss of Il1rl1 decreased expression of oncogenic transcripts such as Six1 and IL1RAP[40,41], and cell cycle and metabolomic signatures including FABP4, Ace, and fbp1 were decreased in Il1rl1-deficient LSCs[43,44].

Previous observations of increased IL-33 and Il1rl1 expression in murine and normal HSCs suggest an involvement of IL-33/Il1rl1 signaling in hematopoietic regulation[12,13] but the mechanism had not been elucidated. To assess the impact of Il1rl1 deficiency on normal hematopoiesis, we developed a conditional knockout model of Il1rl1 in Il1rl1f/f Mx Cre HSCs. Our findings demonstrate that Il1rl1 deficiency impairs normal hematopoietic function, which is consistent with our previous report[12]. This impairment may be due to reduced engraftment efficiency of hematopoietic cells. Then, we demonstrated that IL-33 and Il1rl1 levels were increased upon hematopoiesis stress further supporting a role for alarmin cytokines in hematopoiesis modulation. To then assess the role of the IL-33/Il1rl1 axis specifically in leukemogenesis and in LSCs, experiments involving IL-33-deficient LSCs were conducted in our study. The results demonstrated that IL-33 deficiency in LSCs led to decreased LSC numbers and proliferation plus improved animal survival, indicating that IL-33/Il1rl1 signaling promoted LSCs through the formation of an intrinsic autocrine loop. We then sought out to study the effect of this LSCs intrinsic IL-33/Il1rl1 loop effect on the microenvironment immune cells, precisely the pro-tumoral subsets regulated by IL-33/Il1rl1 like Il1rl1+ Tregs, MDSCs, macrophages, and CD11c+ cells[4,8,76,77]. In our investigation, we observed that the LSCs IL-33/Il1rl1 autocrine loop fosters tolerance by the immune microenvironment, and disruption of the IL-33/Il1rl1 signaling solely in the LSCs can reverse this tolerance by increasing type 1 CD8+ T cells, especially those specific for the leukemia WT-1 antigen, while decreasing Tregs, macrophages and MDSCs. To our knowledge, this is the first evidence that IL-33/Il1rl1 autocrine loop orchestrated by LSCs has direct impacts on the surrounding immune system. Together, these data suggest that targeting the IL-33/Il1rl1 signaling pathway could be a therapeutic strategy for AML through dual inhibition of the LSCs and the inhibitory tumor microenvironment.

Several surface markers have been proposed as targets for AML because of their relative abundant expression on leukemic cells, including CD123 (IL3Rα), CD44, CD47, CD70, CD96, mesothelin, and CLECL12A/CLL-1[78–84]. However, their expression levels in normal HSCs vary widely. Here, Il1rl1 holds an advantage because its expression on HSCs in steady state is extremely low and it is only induced under hematopoitic stress and in an inflammatory tumoral microenvironment. This advantage is amplified when the goal is to target LSCs[85]. Even more important, targeting the seed (LSC) and the soil (inhibitory microenvironment) simultaneously by inhibiting the IL-33/Il1rl1 axis provides a synergetic drug and a durable remission. Indeed, Tregs is a

crucial tumor immune-evasion mechanism associated with poor prognosis in cancer, and their modulation could improve effector lymphocyte function, thereby awakening and enhancing the anti-tumoral immune responses[86–88]. By embedding themselves in non-lymphoid tissues[89], and the leukemia BM microenvironment, Il1rl1+ $T_{reg}$ cells are much harder to be selectively removed by standard anti-Treg strategies without causing broad toxicities. We previously showed that Il1rl1 expression on macrophages is associated with low CD8+ T cell cytotoxicity and inhibition of the IL-33/Il1rl1 pathway with an IL-33trap enhances antigen-specific T cell responses in vivo in colorectal cancer models[8]. Further, anti-Il1rl1 neutralizing antibody or Il1rl1 small molecule inhibitors administered after allogeneic hematopoietic cell transplantation has shown therapeutic efficacy in GVHD and GVL which was better than anticipated with the sole reduction of GVHD, suggesting a direct leukemia activity of the anti-Il1rl1 drugs[17,90]. In the current study, we developed murine and human bispecific T-BsAb targeting Il1rl1 and CD3 using a 2 + 2 [IgG(L)-scFv] platform that is Fc silenced to eliminate all FcγR and complement activation functions[53,54] and structurally optimized for increased potency without T cell exhaustion[49–52]. After confirming that these antibodies effectively eliminate Il1rl1+ leukemic cells when cocultured with CD8+ T cells, we proceeded to evaluate these T-BsAb in two syngeneic immuno-competent models and two human xenograft models, using murine specific and human specific antibodies, respectively. Our observations revealed that upon administration of anti-Il1rl1 T-BsAbs, not only LSCs were decreased but also Il1rl1+ immune cells including Tregs and exhausted CD8+ T cells accompanied by a rise in total CD8+ T cells as well as antigen specific WT1+CD8+ T cells, particularly those of the effector memory phenotype. Considering the tissue-specific nature of Il1rl1+ Treg cells, our findings support a novel approach for dual targeting within the local tumor microenvironment. Notably, we also confirmed the absence of toxicity, specifically the absence of colitis as evidenced by the absence of body weight and colon length changes (Figs. 5G and 6M). Of note, there are two spliced isoforms of IL1RL1: soluble (short IL1RL1) and membrane-bound (long IL1RL1) with opposite roles[4]. The soluble form of IL1RL1 is a decoy receptor of IL-33 and does not signal, while the membrane-bound form of IL1RL1 binds to IL-33 and signals, mostly in T helper 2 cells, regulatory T cells, innate lymphoid cell type 2, M2 polarized macrophages, and mast cells. Since IL1RL1 was initially discovered on Th2 cells[91], which is elevated in asthma and chronic obstructive pulmonary disease (COPD), several neutralizing IL1RL1 antibodies (i.e., astegolimab) have been developed. Safety of astegolimab has been established in a phase 2a placebo-controlled trial for COPD patients[92]. Similarly, astegolimab efficacy and safety in adults with severe asthma have been demonstrated in a randomized clinical trial[93]. In our binding assays, our anti-IL1RL1 antibodies did not revealed any binding across a broad panel of normal human tissues (Supplementary Table 6).

Noting the decreasing ratio of CD8+ T cells to leukemic cells during disease progression, we combined anti-Il1rl1 T-BsAb treatment with adoptive transfer of CD8+ T cells, further improving outcomes compared to anti-Il1rl1 T-BsAb treatment alone. The IL-15 superagonist ALT-803 that boosts CD8+ T cells proliferation has been used successfully in combination with checkpoint inhibitors in patients with metastatic non-small cell lung cancer[57]. Therefore, we employed this strategy in conjunction with anti-Il1rl1 T-BsAb resulting in improved survival compared to anti-Il1rl1 T-BsAb treatment alone. Together, adoptive transfer of CD8 T cells even more than ALT-803 improve AML survival because it provides, additional CD8 T cells to be engaged by BC281, external in the case of the adoptive transfer and internal in the case of ALT-803. ALT-803 is used as a cost-effective alternative to adoptive transfer of CD8 T cells that does not need cell manipulation.

In addition to the immunocompetent MLL-AF9 model, we assessed the impact of anti-murine IL1RL1 T-BsAb in another clinically relevant immunocompetent DNMT3A/FLT3$^{ITD}$ epigenetic leukemia

model[58] which showed similar results on the leukemic cells and immune cells confirming IL1RL1 dual blocking is a generalizable mechanism. Finally, we confirmed that IL1RL1 was up-regulated in eight pediatric AML PDX models regardless of MLL oncofusion and that in vivo treatment with anti-hIL1RL1 T-BsAb BC282 reduced leukemia to undetectable levels within 5 weeks as opposed to no treatment, human T cells, and T cells with control T-BsAb BC283 therapies. Although this AML PDX did progress, further survival extension after retreatment with T-BsAb BC282 suggested that the value of targeting IL33/IL1RL1 was not lost even with recurrence.

While we observed impressive survival in our models, we have seen in some instance relapse which implies there are surviving LSCs. Indeed, LSCs are a heterogeneous population and not all may be killed by the treatments. Furthermore, Fig. 2E–G suggest that LSCs are becoming more stem-like and are returning to a more quiescent state when the exogenous signal is removed and targeting IL1RL1 would potentially keep LSCs quiescent leading to later relapse. As a matter of fact, in our own PDX data, we have shown late relapse that can fortunately be treated with a reinduction therapy (Supplementary Fig. 29), as it is often the case with immune targets and T-BsAb. In view of the significant increase in survival rates, it is likely that a majority of AML cells are killed by the anti-Il1rl1 T-BsAbs. Therefore, anti-Il1rl1 T-BsAb target heterogeneous population of leukemic cells including progenitors (CMP, MEP, MPP, and GMP, Supplementary Fig. 21) and LSCs. Although T-BsAb are designed to avoid the need for chemotherapy, recent studies of combination of chemotherapy and anti-CD19 T-BsAb (Blinatumomab) have shown improved survival rates in randomized clinical trials for standard-risk B cell acute lymphoblastic leukemia in children[94]. In the context of lung cancer, a recent study showed that combination of cisplatin with anti-IL1RL1 neutralizing antibody enhances the anti-tumoral effect of cisplatin[95]. This combinatorial question will be of interest here, however, our current pre-clinical models, do not allow us to test it because standard chemotherapy regimens lead to the eradication of malignant cells in mice, preventing post-treatment analysis.

In summary, elevated IL1RL1 expression in leukemic cells and LSCs is associated with unfavorable prognosis and refractoriness in AML patients. Our findings pinpoint IL-33/IL1RL1 axis as a crucial regulator in LSCs persistence, and maintenance of the leukemic microenvironment. Mechanistically, we observed enhanced induction of the IL-33/Il1rl1 axis in LSCs under hematopoietic stress conditions, contributing to the remodeling of leukemic BM niches. This axis plays a pivotal role in initiating leukemogenesis and sustaining leukemia growth through the establishment of an augmented autocrine loop. Furthermore, this LSCs-induced IL-33/Il1rl1 positive feedback amplifies the inhibitory immune microenvironment in the leukemic niche. Using engineered T-BsAb, we tested and confirmed the therapeutic value of using T cells to target Il1rl1, to eliminate LSC and the inhibitory tumor microenvironment, using mouse specific antibodies against mouse leukemias in immunocompetent mice, and human specific antibodies against human cell line xenografts and human PDXs. The duality of Il1rl1 targeting, of leukemic stem cells (seed) and the inhibitory immune cells in the microenvironment (soil), represents a novel approach to treating one of the least curable leukemias in man.

## Methods

### Human research participants
BM aspirates were collected from 12 AML patients who achieved complete response (CR) and 10 non-responders (NR) ~21 days following induction therapy with chemotherapy. All samples and associated clinical data were obtained with informed consent under institutional review board–approved protocols at FHCC. Patient demographics are provided in Supplementary Table 1. Healthy human BM frozen cells (catalog: 2S-101D) were purchased from Lonza and the features of healthy donors were not available. AML PDX lines were generated from primary AML BM specimens collected from the Nemours Biobank or

from the Children's Oncology Group under an institutional review board approved protocol. PDX lines DF-5 (CBAM-44728) and DF-2 (CBAM-68552) were obtained from the Dana Farber Cancer Institute's PRoXe depository. The patient demographics and the sample characteristics are listed in Supplementary Table 7.

## Mice used in this study

6–8 weeks female or male C57BL/6 (B6, H-2b, CD45.2[+], Strain#: 000664) mice, C57BL/6 BoyJ (CD45.1[+], Strain #: 002014), B6(129S4)-IL-33[tm1.1Bryc]/J (Strain #:030619), DBA/2J mice (Strain #: 000671), or NOD.Cg-Prkdc[scid] Il2rg[tm1Wjl]/SzJ (NSG, Strain #: 005557) mice were purchased from The Jackson Laboratory. The NSG mice were provided and maintained by the Division of Laboratory Animal Resources of the Medical University of South Carolina. C57BL/6 *Mx1Cre* mice were generously provided by Reuben Kapur (Indiana University School of Medicine); C57BL/6 Il1rl1[−/−] (CD45.2[+]) mice were initially provided by Andrew McKenzie (University of Cambridge, Cambridge, United Kingdom). C57BL/6 BoyJ CD45.1[+]Il1rl1[−/−] mice were obtained by crossing C57BL/6 BoyJ with C57BL/6 Il1rl1[−/−] (CD45.2[+]) mice for 10 generations. C57BL/6 homozygous IL-33-Citrine knockout mice were kindly provided by William C. Gause and Aimee M. Beaulieu' lab (Center for Immunity and Inflammation (CII), New Jersey Medical School Cancer Center). C57BL/6 Il1rl1[f/f] mice were generated through the Taconic-Cyagen Academic Model Generation Alliance (Santa Clara, CA, USA) by inserting floxed LoxP sites to part of exon 6–8 using the same deletion as the published Il1rl1 full knock-out[91]. C57BL/6 Il1rl1[f/f Mx1Cre] mice were generated by crossing C57BL/6 *Mx1Cre* mice with C57BL/6 Il1rl1[f/f] mice for 4 generations. C57BL/6 Il1rl1[f/f Mx1Cre] mice were further verified by genotyping. Il1rl1 conditional knockout mice were obtained following Polyinosinic:polycytidylic acid (pIpC, catalog #: P0913, Millipore Sigma) administration with previously doses indicated[96]. Both male and female animals were included in the animal models used in this study. Animal protocols were reviewed and approved by the Institutional Animal Care and Use Committee (IACUC, No.:20-0977) at Medical University of South Carolina. All mice were housed under specific pathogen-free conditions in the animal facility at Medical University of South Carolina. The animal rooms were maintained on a 12-h light/dark cycle, with a constant ambient temperature of $22 \pm 2\,°C$ and relative humidity maintained at $50 \pm 10\%$. Mice had ad libitum access to food and water throughout the study. All animal tests were strictly conducted following the guidelines of experimental pain in conscious animals for minimizing their suffering and improving their welfare.

## AML transcriptomic datasets

TCGA-AML ($n = 173$), TARGET-AML ($n = 187$)[18] and normal GTEX RNA-sequencing datasets ($n = 514$)[18] were downloaded from the TCGA. TCGA and TARGET AML data sets were merged after removing batch effects using sva package[97]. AML microarray datasets including GSE37642 ($n = 417$)[19], GSE13159 (*MILE study dataset, n = 2096*)[22], GSE1159 (285 AML and 8 HD samples)[20], GSE6891[21] ($n = 537$), and GSE9476 (*26 AML and 38 HD*)[20]. LSCs datasets including Princess Margaret[25], GSE63270 ($n = 20$), and normal HSCs datasets including GSE17054[26] ($n = 4$) GSE42519[27] ($n = 16$), GSE19599[28] ($n = 6$), GSE24759[29] ($n = 4$) were downloaded from the Gene Expression Omnibus (GEO) database. Briefly, raw "CEL" files were downloaded and a robust multi-array averaging method using affy packages for background adjustment and quantile normalization was performed. Batch effects were removed by the combat algorithm in the sva package. These datasets were internally reviewed, and duplicate samples were excluded from the subsequent analysis.

## IL1RL1/IL-33 expression in LSCs and normal HSCs

IL1RL1 and IL-33 expression levels in LSCs from the Princess Margaret database were compared with those in HSCs from the GSE42519

dataset. These findings were validated using independent LSC datasets (GSE63270) and normal HSC datasets (GSE19599, GSE17054, and GSE24759).

For IL1RL1 expression in AML patients, comparisons were made across different molecular mutations (including IDH1, IDH2, NPM1, CEBPA, FLT3, NRAS, and EVI1) and cytogenetic abnormalities (including t(15;17), t(8;21), t(11q23)/MLL, complex, or normal karyotype) using data from GSE1159 and GSE6891. These were compared with HD from the GSE1159 and GSE9476 datasets. Although, we were not able to analyze the MILE dataset (GSE13159)[22] because it compared microarrays to standard laboratory methods, we were able to analyze the Naef's database[23] in comparison to HD from GSE42519 and confirmed IL1RL1 is elevate in AML regardless of the cytogenetic mutation.

## Survival analysis of IL1RL1 in AML

TCGA-AML ($n = 173$) and TARGET-AML ($n = 187$) were combined after the removal of batch effect, and patients were divided into IL1RL1[high] and IL1RL1[low] groups based on the median value of IL1RL1 expression. Patients' survival difference between IL1RL1[high] and IL1RL1[low] groups were compared by Kaplan-Meier curves using "survival" R package. GSE37642[19] was used for independent validation.

## Stem cell infiltration analysis

For stem cell infiltration analysis in the AML BM, patients' samples were divided into IL1RL1[high] and IL1RL1[low] groups based on the median value of IL1RL1 expression. A total of 6 different stem cell subpopulations including HSC, MPP, multipotent CMP, common lymphoid progenitor, GMP, and megakaryocyte/erythrocyte progenitor (MEP), were deconvoluted from TCGA and TARGET combined AML dataset ($n = 360$) and AMLCG 1999 trial dataset by xCell algorithm, a gene signatures-based method to infer 64 immune and stromal cell types that were described previously[24]. The fractions of these stem cell subtypes in IL1RL1[high] and IL1RL1[low] groups were compared using Wilcox test.

## Generation of MLL-AF9[GFP] leukemic cells

The retroviral plasmid that embeds the MLL-AF9[GFP] cDNA construct[98] was generously provided by Dr. Reuben Kapur and used to generate retroviral particles through transIT transfection of Phoenix Eco Cell Line (ATTC) using lipofectamine 3000 Reagent. Phoenix Eco cells were seeded in 100-mm plates ($8 \times 10^6$/plate) before transfection, and cultured overnight at 37 °C, 5% $CO_2$. After cells were 70–90% confluence, 20 µg of DNA was diluted in 1.5 mL Opti-MEM® I Reduced-Serum Medium in the presence of P3000 enhancer reagent, lipofectamine 3000 reagent was diluted in 1.5 mL opt- MEM® I Reduced-Serum Medium. Cells were transfected with a mixture of diluted DNA and P3000 transfection reagent. After 18 h of incubation, the media was replaced with 12 mL of DMEM supplemented with 10% heat inactivated, and 100 U/mL penicillin and streptomycin. After a 24-h incubation, retroviral supernatants were collected and filtered through 0.45-µm filters. Freshly prepared retroviruses were used to transduce Lineage negative cells that were magnetically enriched from BoyJ WT and Il1rl1[−/−], B6. Il1rl1[f/f] and Il1rl1[f/fMx1Cre], B6. WT and IL-33[Cit/Cit KO] BM cells using murine Lineage Cell Depletion Kit (Order no. 130-090-858, Miltenyi), and pre-stimulated for 24 h in IMDM media suppled with 10 ng/mL mIL-3, 10 ng/mL mIL-6, and 50 ng/mL mSCF (all from Peprotech). After two consecutive 24-h infections in non-tissue culture plates pre-coated with retronectin[99], cells were collected, and their infection efficiency was determined by GFP expression using flow cytometry.

## Cell culture of human leukemia cell line

Human leukemic cell line MOLM-14-eGFP was cultured in RPMI 1640 medium (Gibco, Catalog: 11875093) supplemented with 10% FBS (BI), 100 U of penicillin per ml and 100 mg/ml streptomycin (Gibco, catalog: 15140122) and grown at 37ºC with 5% CO2.

## Leukemia initiation and maintenance experiments

For primary transplant of transformed MLL-AF9 LSK cells in the leukemia initiation experiments by limiting dilution assay, 500, 200, 50 sorted Il1rl1 WT and Il1rl1$^{-/-}$ LSCs (MLL-AF9$^{GFP}$Lineage negative, c-Kit$^+$Sca-1$^+$) were injected into lethally irradiated 6–8-week-old C57BL/6 mice. In secondary, tertiary transplants, cell numbers of $0.5 \times 10^6$ BM leukemic cells from the primary or secondary leukemic mice were transplanted respectively, into lethally irradiated C57BL/6 mice.

MLL-AF9 leukemia maintenance experiment was performed by injecting $10^3$ sorted Il1rl1$^{f/f}$ and Il1rl1$^{f/f\ Mx1Cre}$ MLL-AF9 LSCs (MLL-AF9$^{GFP}$Lineage negative, c-Kit$^+$Sca-1$^+$) to 6 to 12-week-old C57BL/6 immunocompetent mice, and treating the mice with 150 mg/kg polyinosinic:polycytidylic acid (pIpC) twice a day for a total of 7 doses posttransplant. Il1rl1 deletion in LSCs were verified using flow cytometry.

## ROS production

ROS levels were detected by labeling $5 \times 10^5$ MLL-AF9$^{eGFP}$ leukemic cells for 30 min with 10 μM MitoTracker Red CMXRos (catalog: M7512, Invitrogen, Life Technologies) in PBS in incubator at 37 °C, 5% CO2. Then, labeled cells were washed twice with 1x PBS and analyzed by flow cytometry.

## Cell cycle analysis

WT and Il1rl1$^{-/-}$ MLL-AF9$^{GFP}$ leukemia cells were harvested and washed twice with 1x PBS. The washed cells were fixed by adding 70% pre-cold ethanol in a dropwise manner to the pellet with vortexing well and incubated overnight at 4 °C. Cells were washed twice with staining butter after removing ethanol. Then, pre-diluted anti-mouse Ki-67 antibody was added and incubated 30 min at room temperature. Afterward, cells were washed twice and staining with 3 nmol DAPI for 20 min. Stained cells were analyzed by flow cytometry. Triplicate experiments were conducted for each sample.

## Homing assay

A total of $1 \times 10^7$ WT or Il1rl1$^{-/-}$ BM cells (CD45.1$^+$) were injected into lethally irradiated (12 Gy) recipient mice (CD45.2$^+$). BM cells were harvested at 18 h after injection and the frequency of donor-derived cells was analyzed by flow cytometry.

## 5-FU induced hematopoiesis stress

To directly assess Il1rl1/IL-33 expression in HSCs under myeloablative stress induced by 5-Fluorouracil (5-FU). Briefly, mouse BM cells were harvested from B6. IL-33$^{eGFP}$ mice receiving one dose 5-FU (150 mg/kg, Cat.No: F6627, Sigma-Aldrich) at Day 3 and Day 5. Then, HSCs were stained with eFluor 450 mouse hematopoietic lineage cocktail (CD3, Gr-1, TER-119, CD45R, CD11b; Cat.No: 88-7772-72), PE-Cy7 mouse c-KIT (Cat.No: 25-1171-82), APC mouse Sca-1 (Cat.No: 17-5981-82), and mouse Il1rl1-PE (Clone DJ8; Catalog:101001PE) for 30 min. Data was collected on ThermoFisher Attune NxT systems and analyzed by FlowJo software.

## Generation of CRISPR/Cas9-mediated IL1RL1 genetically knockout

Stable knockout of IL1RL1 in human MOLM14 cells was generated using CRISPR/CAS9 technology. The lentiviral transduction particles for IL1RL1 knockout were produced by transient co-transfection of HEK293T cells (ATCC) with IL1RL1 sgRNA CRISPR/Cas9 All-in-one lentivector set (Cat.No: 248541110595, ABM). After 48 h incubation, the lentiviral particles were collected and stored at −80 °C until use. CRISPR scrambled sgRNA All-in-One lentiviral vector (Cat.No: K010, ABM) was used as control. MOLM14 cells were transduced with the lentivirus stocks in the presence of 8 μg ml/L polybrene (Lot: 3308409/TR-1003-G, EMD Millipore Corp.) overnight and then selected with 2 μg ml/L puromycin for at least 2 weeks to establish control MOLM14 cells or cells displaying stable IL1RL1 knockout. Flow cytometry and quantitative RT-qPCR were used to verify the deletion of IL1RL1 in MOLM14 cells.

## RNA preparation and RT-qPCR

Total RNA was extracted from control MOLM14 and IL1RL1 knockout MOLM14 cells using RNeasy Plus Mini Kit (Cat.No: 74134, QIAGEN, Germany). and 1 μg of total RNA was reverse-transcribed into cDNA with iScript$^{TM}$ cDNA synthesis kit (Cat.No: 1708891, Bio-Rad). PCR was performed with the synthesized cDNA in a final volume of 20 μL containing 4 μL 5x iScript Reaction Mix, 1 μL iScript Reverse Transcriptase, and 1 μg total RNA on a T100 Thermal Cycler (Bio-Rad). The PCR amplification was performed through incubating samples the following thermal cycler protocol: priming for 5 min at 25 °C, followed by 10 min at 46 °C for reverse transcription, and a final RT inactivation for 1 min at 95 °C.

## Quantitative RT-PCR (RT-qPCR)

Quantitative real-time PCR was conducted using SsoAdvanced Universal SYBR Green Supermix kit (Cat.No: 172-5270, Bio-Rad) in a total volume of 20 μl on a C1000 Touch Thermal Cycler CFX96 Real-Time system (Bio-Rad): 95 °C for 30 s and 40 cycles of denaturation (95 °C for 15 s) and extension (60 °C for 1 min). Experiments were performed in triplicates. The relative level of IL1RL1 gene expression in control and IL1RL1 knockout MOLM14 cells was normalized against human GAPDH. Quantification was performed using the $2^{-\Delta\Delta CT}$ method. Primers for qRT-PCR were described previously[100].

## Anti-IL1RL1 bispecific antibodies, chromatogram analysis and Surface Plasmon Resonance (SPR), and IL-15 superagonist

Anti-mouse Il1rl1 (BC281) bispecific antibody and control (BC462), anti-human IL1RL1 bispecific antibody (BC282) and control (BC283) VH and VL sequences were inserted into a mouse IgG1 Fc cassette carrying the D265 mutation for Fc silencing (U.S. patent application # 63/250,706). Each IgG was produced using the Expi293 expression system (Thermo Fisher Scientific RRID: CVCL_D615) or HEK 293T cell (RRID: CVCL_KS61) Antibodies were purified with protein an affinity column chromatography, and the stability was monitored by SEC-HPLC. Endotoxin level of <1 EU/mg was confirmed. Both chromatogram analysis and surface plasmon resonance have been detailly described in previous study. Briefly, Il1rl1 bispecific antibodies were dissolved with 10 μl mobile phase of PBS, pushed through a Superdex S200 10/300GL column (GE Healthcare) at a flow rate of 0.5 ml/min on an AKTA purifier (GE Healthcare). The percent monomer was calculated as the area of the monomeric peak divided by the total area of monomeric plus non-monomeric peaks at 280 nm. For the SPR test, recombinant murine or human IL1RL1 was immobilized to CM5 sensor chips. The binding kinetics was respectively monitored by flowing of BC281 or BC282 over the chip for association, and further monitored for their dissociation from the surface being washed for another 5 min. ALT-803, IL-15 superagonist, was provided by Dr. Wrangle through a MTA with Altor BioScience and therapeutic regimen in immunocompetent models was similar to the one published[57]. For the xenografts, the multimeric IL15/IL15Rα-Fc complex was provided by Dr. Cheung[64].

## Killing assays of leukemic cells or regulatory T cells by anti-Il1rl1 T-BsAb

CD8a-positive T cells and regulatory T cells (from Foxp3$^{eGFP}$ mice) were sorted from normal BM niche, or sometimes from the same mice spleens, and co-cultured with background matched MLL-AF9$^{GFP}$ leukemic cells at indicated ratio for 6–16 h in 96 round bottom wells with the presence of different concentrations of anti-Il1rl1 T-BsAb BC281 or control BC462. Human peripheral blood mononuclear cells were enriched from healthy donor PB via density gradient centrifugation by using Ficoll-PaqueTM PREMIUM (GE Healthcare #17-5442-02). CD8$^+$ T cells were further purified via positive selection using

immunomagnetic beads (Miltenyi #130-045-201) according to the manufacturer's protocol and passed through the LS magnetic cell-sorting column. Killing assays were then performed as previously described[101]. Briefly, co-cultured cells were taken out from the incubator and washed with PBS for twice and stained with viability dye (1:1000; SYTOX-Blue, Invitrogen) 30 min before flow analysis. Data were analyzed by Flowjo (10.0.1), and lysed cells were defined as GFP and SYTOX-Blue double positive ones.

## Stability and Pharmacokinetics analysis of Anti-Il1rl1/IL1RL1 T-BsAb

Il1rl1/IL1RL1 BsAb antibodies were dissolved with endotoxin-free PBS and filtered through 0.22 μm filterers. In advance of the animal test, a LAL chromogenic endotoxin quantification kit (Thermo Scientific, # 88282) was used to validate the endotoxin levels.

For stability assessment of murine and human Il1rl1/IL1RL1 T-BsAb (BC281 and BC282), these T-BsAb were stored at two temperatures (4 °C and 37 °C), with purity measured weekly using high-performance liquid chromatography (HPLC), as detailed previously[49].

To assess their pharmacokinetics in vivo, a single dose of 10 μg/mouse of BC281 was intravenously injected into normal 6- to 10-week-old male and female C57BL/6 mice. Similarly, one dose of 10 μg of BC282, with or without $1 \times 10^6$ activated T cells, was intravenously injected into normal 6- to 8-week-old male and female NOD.Cg-Prkdc$^{scid}$ Il2rg$^{tm1Wjl}$/SzJ (NSG) mice. Serum samples were collected at various time points post-injection, and the serum concentrations of BC281, BC282, and BC282 with T cells were determined using enzyme-linked immunosorbent assay (ELISA), following established protocols[102].

## In vivo therapeutic study of anti-Il1rl1/IL1RL1 T-BsAb in murine and humanized leukemia models

In the murine leukemia model, $10^4$ MLL-AF9$^{gfp}$ leukemic cells per mouse were intravenously injected into 6–8-week-old male and female C57BL/6 immunocompetent mice. After a 4-day incubation period, mice received 10 μg of anti-Il1rl1 BsAb BC281 or control BC462 intravenously twice a week for 6 doses. BC281 was also tested with adoptive transfer of murine CD8$^+$ T cells or ALT-803 (IL-15 superagonist, provided by Dr. Wrangle under an MTA with Altor BioScience). Murine CD8$^+$ T cells were purified using CD8a$^+$ T Cell Isolation Kit (catalog: 130-104-075, Miltenyi Biotec) according to the manufacturer's instructions. Then CD8$^+$T cells ($1 \times 10^6$ per mouse) were adoptively transferred weekly by intravenous injection, or 4 μg of ALT-803 was administered subcutaneously to the leukemic mice.

In the Dnmt3a/Flt3$^{ITD}$ epigenetically induced immunocompetent myeloid leukemia model, $1.5 \times 10^6$ leukemia cells were intravenously injected and allowed 3 weeks for engraftment before initiating treatment with BC462 and BC281. Treatment was administered every 3 days for a total of 10 doses. Due to the aggressive nature of DF AML, the initial dose was set at 25 μg/mouse for the first three doses, followed by an increased dose of 50 μg/mouse for the remaining seven doses.

In the humanized leukemia treatment model, $10^5$ MOLM14$^{gfp}$ cells were injected into 6–8-week-old male and female NSG mice. Post a 4-day incubation period, BC282 and control BC283 BsAbs were injected using a similar treatment strategy as in the immunocompetent models. Concurrently, $1 \times 10^6$ purified human CD8$^+$ T cells (Human CD8$^+$T Cell Isolation Kit, catalog:130-096-495, Miltenyi Biotec) were injected weekly for a total of 3 weeks.

## Noncompetitive transplants and chimerism assessment

Noncompetitive transplants involved injecting $2 \times 10^6$ BM cells from 6–8-week Il1rl1$^{f/f\ Mx1cre}$ or Il1rl1$^{f/f}$ mice into lethally irradiated B6SJL congenic CD45.1 recipients. In the recipients, chimerism was checked from 4 to 20 weeks via PB or BM aspirates. For chimerism, either PB or BM cells was extracted and subjected to red blood cell lysis.

Hematopoietic stem and progenitors are defined based upon the expression of surface markers: LT-HSC (Lin⁻Sca1⁺Kit⁺CD48⁻CD150⁺), ST-HSC (Lin⁻Sca1⁺Kit⁺CD48⁻CD150⁻), MPP (Lin⁻Sca1⁺Kit⁺CD48⁺CD150⁻), CMP (Lin⁻Sca1⁻Kit⁺CD16/32⁻CD34⁺), GMP (Lin⁻Sca1⁻Kit⁺CD16/32⁺CD34⁻), and MEP (Lin⁻Sca1⁻Kit⁺CD16/32⁻CD34⁻)[33].

## Flow cytometry staining and cell sorting

All antibodies used for intracellular, or surface staining used in this study were purchased from eBioscience (San Diego, CA) or BD Bioscience (San Jose, CA) or Biolegend (San Diego, CA) unless otherwise stated. Mouse Il1rl1-PE (Clone DJ8; Catalog:101001PE) and human IL1RL1-FITC (Clone B4E6; Catalog:101002F) antibodies were purchased from MD Biosciences Bioproducts (Oakdale, MN). For cell surface multiple antibodies staining, single cell suspension for each sample was blocked with anti-mouse CD16/CD32 mAb (BD Biosciences, Catalog: 553141) for 10-20 min at 4 °C to prevent non-specific binding. Afterward, the cells were incubated with recommended dilutions of antibodies against cell surface markers at 4 °C for 30 min. Then the fixable viability dye was added at 1:1000 to distinguish live cells from dead cells. Cells were washed and stored in PBS/2% FBS at 4 °C in the dark until analyzed by flow cytometry.

For intracellular and cytokines staining, cells were fixed after surface staining with Foxp3/Transcription Factor Staining Buffer Set and the Fixation/Permeabilization Kit (Cat.No: 00-5523-00, Invitrogen) for 30 min at room temperature, followed by permeabilization with 1x permeabilization buffer. For the cytokine staining, single cell samples were generally stimulated with PMA (50 ng/ml; Sigma-Aldrich), ionomycin (1 μg/ml; Sigma-Aldrich), and in the presence of GolgiStop monensin (1 μg/ml) for 5–6 h at 37 °C. Cells were resuspended in 1X permeabilization buffer following twice washing with 1X permeabilization buffer.

For BM transplantations of sorted LSCs defined by Lineage⁻c-KIT⁺Sca-1⁺, mouse leukemia cells isolated from the BM of leukemia-bearing mice were stained with eFluor 450 mouse hematopoietic lineage cocktail (CD3, Gr-1, TER-119, CD45R, CD11b; Cat.No: 88-7772-72), PE-Cy7 mouse c-KIT (Cat.No: 25-1171-82), and APC mouse Sca-1 (Cat.No: 17-5981-82) for 30 min, and then sorted using BD FACSAria (BD Biosciences).

Both the surface and the intracellular staining samples were analyzed via flow cytometry using BD LSRII and BD LSR Fortessa (X-20) or ThermoFisher Attune NxT systems. All flow cytometric analyses were performed using FlowJo 10.7.0 software (TreeStar). The gating strategies used to analyze BM samples of AML patients and leukemic mice are presented in Supplementary Fig. 2. To determine the exact count of a cell population, we start by tallying up all the living cells in the tissue we're studying, like BM or spleen, for each mouse. Then, the absolute number of a specific population (i.e., CD3$^+$CD8$^+$T cells) is calculated by multiplying the total number of viable cells per tissue (BM or spleen) by the population frequency (%) on parents' cells. Each absolute number is per mouse.

## RNA-sequencing and analysis

BM cells from mice bearing Il1rl1$^{f/f}$ MLL-AF9$^{gfp}$ and Il1rl1$^{f/f\ Mx1Cre}$ MLL-AF9$^{gfp}$ were harvested following 7 doses of pIpC administration by I.P at Day 21. LSCs were enriched using mouse Lineage Cell Depletion Kit (Cat.No: 130-090-858, Miltenyi), and then stained by eFluor 450 mouse hematopoietic lineage cocktail (CD3, Gr-1, TER-119, CD45R, CD11b; Cat.No: 88-7772-72), PE-Cy7 mouse c-KIT (Cat.No: 25-1171-82), APC mouse Sca-1 (Cat.No: 17-5981-82), and mouse Il1rl1-PE (Clone DJ8; Catalog:101001PE) for 30 min. Both types of LSCs ($1 \times 10^5$) were sorted using BD FACSAria (BD Biosciences). LSCs were immediately pelleted by centrifugation at 4 °C and resuspended in Homogenization Buffer containing thioglycerol provided in the purification kit. RNA was extracted with a Promega Maxwell RSC 16 using the Maxwell RSC simplyRNA Cells kit (Cat# AS1390) according to the manufacturer's

instructions. RNA concentration was measured using a NanoDrop 8000. RNA quality was assessed using an Agilent 4200 TapeStation and RIN$^e$ values ranged from 7 to 9. Total RNA (75 ng) was used the construction of libraries with the New England Biolabs NEBNext® Poly(A) mRNA Magnetic Isolation Module (Cat# 7490L) and Ultra II Directional RNA Library Prep Kit for Illumina (Cat# 7760L) according to the manufacturer's instructions. Dual-indexed libraries were pooled and sequenced at VANTAGE (Vanderbilt University Medical Center) on an Illumina NovaSeq 6000 (S4 flow cell) to a depth of ~25 million paired-end 150 bp reads per library.

Sequencing data was analyzed using Partek® Flow® software. Reads were aligned to mouse genome assembly mm10 by STAR and quantified to annotation model (Partek E/M) using mm10. Differential expression analysis comparing Il1rl1$^{f/f}$ to Il1rl1$^{f/f\ Mx1Cre}$ MLL-AF9$^{gfp}$ LSCs were performed by DESeq2 (Moderated estimation of fold change and dispersion for RNA-seq data with DESeq2). DEGs were defined as adjusted p-value (FDR step up) <0.05 and absolute fold change >1. KEGG pathway analysis was performed to explore the biological functions of DEGs, including both upregulated and downregulated genes, with an adjusted $p$ value threshold of <0.05 for significance. All RNA-sequencing data from this study were deposited in NCBI Gene Expression Omnibus under accession number (GSE253167).

### Confocal microscopy for quantification of Il1rl1$^+$ cells and CD3$^+$ T cells in AML mice

To quantify Il1rl1$^+$cells and CD3$^+$ T cells in the spleens of Dnmt3a$^{+/-}$/FLT3$^{ITD/+}$ AML mice treated with either control T-BsAb (BC462) or anti-Il1rl1 T-BsAb (BC281), we performed confocal microscopy using the following protocol: Sample Preparation: spleens were harvested from DF AML mice treated with BC462 or BC281, and spleen cells were isolated. The cells were then seeded into ibidi μ-slide 8-well™ chambers (Ibidi, Munich, Germany) for subsequent imaging. To preserve cellular structure and protein localization, cells were fixed with 4% paraformaldehyde at room temperature for 15 min. Following fixation, cells were blocked with 5% UltraCruz® blocking reagent (Santa Cruz Biotechnology, sc-516214) for 30 min at room temperature to minimize non-specific antibody binding. Antibody Incubation: blocked cells were incubated overnight at 4 °C with the following fluorescently conjugated antibodies to identify the target populations: Il1rl1 (PE) and CD3 (FITC). Washing and Nuclear Staining: after antibody incubation, cells were washed three times with phosphate-buffered saline (PBS) for 5 min each to remove unbound antibodies. The nuclear stain DAPI (4′,6-diamidino-2-phenylindole) was then applied for 1 min to visualize cell nuclei, followed by a final PBS wash. Confocal Imaging: fluorescent images were acquired using a Leica SP8 resonant scanning confocal/multiphoton microscope at the Indiana University Microscope Core Facility. Imaging was performed at multiple z-planes to capture high-resolution cellular interactions. A minimum of 4–5 different fields per condition were imaged and analyzed to ensure robust data collection and minimize bias. Colocalization Analysis: Colocalization of Il1rl1$^+$cells and CD3$^+$T cells was quantitatively assessed using Imaris (Elmo) software. This analysis provided insights into the spatial distribution of Il1rl1$^+$cells and CD3$^+$ T cells in response to anti-Il1rl1 T-BsAb treatment.

### AML PDX models

Male and female 6- and 8-week-old NOD.Cg-B2m$^{tm1Unc}$ Prkdc$^{scid}$ Il2rg$^{tm1Wjl}$/SzJ (NSG-B2m) and NOD.Cg-Prkdc$^{scid}$ Il2rg$^{tm1Wjl}$ TgCMV-IL3, CSF2, KITLG 1Eav/MloySzJ (NSG-SGM3) mice were purchased from the Jackson Laboratory and maintained in pathogen-free conditions. For AML PDX xenograft model, $3 \times 10^5$ pediatric AML NTPL-377$^{luc}$ cells in 200 μl of sterile PBS were injected into NSG-B2m mice via the tail vein. Mice were randomly assigned to the following treatment groups ($n = 6$–8/group): (1) untreated, (2) human activated T cells, (3) Anti-hIL1RL1 bispecific antibody (BC282) with human activated T cells, and (4) BC283 control bispecific antibody with human activated T cells.

10 μg of BC282 or BC283 intraperitoneally injected 4 days later and subsequently every 3 days for a total of 6 injections with weekly injections of $5 \times 10^6$ human activated T cells. For reinduction therapy, we used the similar treatment regimen with increased human activated T cells ($20 \times 10^6$/dose/mouse).

To amplify the T cell immune response, we implemented a treatment approach involving the combination of BC282 vs BC283 (10 μg/dose/mouse) with IL-15 (1 μg/dose/mouse), following a similar therapeutic protocol in NSG-SGM3 mice. Human activated T cells ($20 \times 10^6$/dose/mouse) were injected on day 5, 12, and 28 post-challenge. Engraftment of xenograft was assessed 2–14 weeks after injection through submandibular bleeding. The frequency of AML PDX cells (iRFP$^+$) in the PB was detected by flow cytometry. During the study, the animals were monitored for potential signs of toxicity, including but not limited to body weight loss exceeding 20%, fur loss, diarrhea, and hindlimb paralysis associated with the disease.

### Quantification and statistical analysis

Quantification methods and samples sizes are described in the figure legends. Unless otherwise noted, statistical analysis was performed using GraphPad Prism 10. The statistical tests applied in each experiment are described in the figure legends. Data are presented as mean ± standard error of the mean (s.e.m.). No data exclusion or outlier analysis was performed during the data acquisition and analysis. Briefly, phenotypic and functional data were compared using unpaired $t$ test for comparison between two groups and analysis of variance (ANOVA) for comparison of three or more groups. To account for the type I error inflation due to multiple comparisons, we applied the Bonferroni correction. A log-rank test was used for survival analysis. All tests were two-sided at the significance level $P \leq 0.05$.

### Reporting summary

Further information on research design is available in the Nature Portfolio Reporting Summary linked to this article.

## Data availability

The data generated in this study has been deposited in NCBI's Gene Expression Omnibus under GEO Series accessions GSE253167. The following previously published datasets were used and appropriately cited: TCGA and TARGET-AML cohorts, GTEx, GSE37642, GSE13159, GSE1159, GSE6891, GSE9476, GSE42519, GSE19599, GSE63270, and GSE24759. Source data are provided with this paper.

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

## Acknowledgements

We thank A. McKenzie from the Medical Research Council Laboratory of Molecular Biology, Cambridge, UK for providing the C57BL/6 IL1RL1$^{-/-}$ mice. We thank Wade Clapp from Indiana University School of Medicine for helping with the generation of the C57BL/6 Il1rl1f/f mice. We thank Reuben Kapur kindly provide MLL-AF9 retrovirus plasmid and C57BL/6 Mx1Cre mice. We thank Derek Stirewalt and Era Pogosova-Agagjanyan for providing patients BM samples from the Fred Hutch/UW Hematopoietic Diseases Repository. We thank Skyler Brand for technical assistance in AML PDX study. This work was supported by a National Cancer Institute (NCI) cancer moonshot U01CA232491 grant (to S.P. and N.-K.V.C.). S.P. is partly supported by the Sally Abney Rose Endowed Chair in Cancer Stem Cell Biology, and NCI-designated Hollings Cancer Biology and Immunology Program (P30 CA138313). N.-K.V.C. is partly supported by the Enid A. Haupt Endowed Chair and Robert Steel Foundation. The Flow Cytometry core at Hollings Cancer Center, Medical University of South Carolina is partially funded by a National Cancer Institute grant (P30 CA138313). AML PDX study was supported by the DoD EO1 W81XWH2210981 (S.P.B.).

## Author contributions

D.F., H.J., A.L., S.K.P., B.R., R.K., S.P.B., N.-K.V.C. and S.P. designed research, analyzed data, and wrote the manuscript. D.F., H.J., E.H., J.R.F., A.G. performed analysis on leukemia patients' samples and in vivo models and interpreted the data. M.L.C. provided essential materials and provided intellectual input in the design of the leukemia initiation and maintenance experiments. J.W. provided the murine IL-15 super-agonist (ALT803), dosage, and experimental design. A.L., H.G. and N.-K.V.C. designed and produced the T-BsAbs including biochemical characterization, affinity evaluation, and PK/PD analyses. S.K.P., B.R. and R.K. designed and performed all experiments related to the epigenetic model. S.B., J.R.F. and A.G. designed and performed all experiments related to AML PDX models. N.-K.V.C. and S.P. conceptualized the project and supervised all aspects of this study.

## Competing interests

N.-K.V.C. reports receiving commercial research grants from Y-mabs Therapeutics and Abpro-Labs Inc.; holding ownership interest/equity in Y-Mabs Therapeutics Inc., holding ownership interest/equity in Abpro-Labs, and owning stock options in Eureka Therapeutics. N.-K.V.C. is the inventor and owner of issued patents licensed by Memorial Sloan Kettering Cancer Center (MSKCC) to Ymabs Therapeutics, Biotec Pharmacon, and Abpro-labs. Both MSKCC and N.-K.V.C. have financial interest in Y-mabs. N.-K.V.C. is an advisory board member for Abpro-Labs and Eureka Therapeutics. Otherwise, the authors declare that they have no competing interests.
