## [Transparent Peer Review file · Nature Communications]

Dual targeting of tumoral cells and immune microenvironment by blocking the IL-33/IL1RL1 pathway

Corresponding Author: Professor Sophie Paczesny

Version 0:

Reviewer comments:

Reviewer #1

(Remarks to the Author)

Fu et al. describe the expression of ST2 in leukemic stem cells, utilize murine models of AML to investigate the role of ST2 in leukemic initiation and maintenance, and assess the anti-tumor efficacy of anti-ST2-CD3 bispecific molecules in nonclinical AML and mastocytosis models. Gene expression analysis of multiple available databases revealed increased expression of ST2 across AML subtypes and that high ST2 expression was associated with poorer overall survival. The authors utilized knockout and inducible cre models to describe the impact of ST2 loss in leukemia, demonstrating that loss of ST2 was associated with reduced disease burden and changes within the immune microenvironment. Generation of murine and human anti-ST2-CD3 bispecific molecules allowed for the treatment of nonclinical AML and mastocytosis models to demonstrate that T cell directed killing of tumor cells is feasible approach for the treatment of these diseases.

This is a very thorough and descriptive manuscript with multiple different approaches being taken to address the role of ST2 in leukemia and mastocytosis.

Major Comments:

1. The authors indicate that ST2 mRNA is increased across a number of molecularly defined AML subgroups relative to healthy donors. According to Naef et al. *Science Translational Medicine* 2023;16:eadd7705, ST2 expression was elevated primarily in t(8;21), t(9;22), del(9q), or DEK/NUP214 subgroups. Was the analysis criteria used for Figure 1E applied to the TCGA or MILE datasets and identified ST2 as being elevated in IDH1mut, IDH2mut, NPM1mut, CEBPAmut, FLT3-ITD, FLT3-TKD, NRASmut and EVIpos? Please include reference to Naef et al. analysis in the results and explain that the data presented here either supports and further expands or is it in contrast to already published data (and why there are differences between the analyses).
2. The authors describe an autocrine loop induced by the LSCs however, co-expression of IL-33 and ST2 is not investigated. What is the co-expression of IL-33 and ST2 on the LCSs for the data included in Figure 4? For the autocrine loop involvement in immune tolerance data, expression of ST2 and IL-33 in the immune cells should be included as part of the analysis. What is the IL-33 and ST2 expression on the immune cells included as part of Figure 4D and S7?
3. Given the described expression of ST2 on immune cell populations (including T cells) and data from Figure 4D and S7 showing impacts of IL-33 knockout on MDSCs, Tregs, macrophages, dendritic cells, the effects of BC281 treatment on these populations in the tolerability study as well as the immunocompetent disease models needs to be assessed and discussed.
4. ST2 expression on the tumor cells should be indicated for the post-challenge samples for each of the models where it can be assessed, especially for the models where anti-ST2-CD3 treatment was used. Does BC281/BC282 treatment eliminate the ST2+ cells?
5. For all inoculation models, the authors need to clarify if only ST2+ cells were selected and inoculated or if bulk cells were inoculated. The schematics show percent ST2+ cells in the population but the text and figure legends are unclear as to whether those are first selected or not. It is important to understand if 100% of the starting tumor cells were ST2+ or if just a small population was.
6. The weekly adoptive T cell experiment (Figure 6H) and ALT803 combination studies (Figure 6I) need to be compared to the appropriate controls. The overall survival and percent of CD3+CD8+ T cells looks similar between Figure 6C/E (no weekly adoptive T cell transfer) and Figure 6H (weekly T cell transfer). The combination of bispecific + ALT803 (Figure 6I) needs to include the BC462 alone and BC281 alone controls in the survival curve and graphs for Figure 6I. When compared to the bispecific alone, is there an expansion of the T cells and improvement in survival with ALT803? The survival curves

for Figure 6C and 6I look very similar as do the percent T cell populations in 6E vs. 6I. Lines 545-546 in the discussion currently imply an improved survival when comparing anti-ST2 + IL-15 to anti-ST2 but the data in Figure 6 does not make this comparison as it is plotted. If the data are not significant, modify the statements in the discussion to reflect the data. The authors should also address if the remaining leukemic cells post-challenge are ST2- and whether continuing to inoculate T cells or trying to expand them with IL-15 would be expected to have an increased efficacy and why.

7. For the humanized NTPL-377 model, the authors need to comment on/clarify why NSG-SGM3 mice and 20×10^6 activated T cells were used for the experiment in Figure S12 in the results section when the comparator study (Figure 7) utilized NSG-B2m mice and 5×10^6 activated T cells. Also comment on why the overall survival was shorter for the animals in Figure S12 than Figure 7. BC282 + T cells and BC283 + T cell controls need to be included in the study to accurately assess whether the addition of IL-15 is improving survival compared to bispecific treatment alone.

8. Clarify whether the mice in Figure S13 were a continuation of study from Figure 7 or if this is a new study. If it is a new study, why were NSG mice used when NSG-B2m or NSG-SGM3 mice were used for Figure 7 and Figure S12, respectively? Survival curves need to be plotted to demonstrate the "notable survival extension in BC282 treated PDXs as contrasted with both BC283 control and solitary t cells treatment" as stated in the text (lines 403-404).

9. The authors need to clarify the expression of ST2 on normal HSCs. Line 509 of the discussion: "Here, ST2 holds a distinct advantage because it is not constitutively expressed on normal HSCs" seems to contrast with the introduction (lines 81-82) that state "We and others, also, found that ST2 is expressed on normal murine and human HSCs, respectively." This does present a potential therapeutic liability and wouldn't allow for the targeting minimal residual LSC disease as implied by the authors.

10. The authors should discuss potential on-target off-tumor toxicity that could be relevant to targeting ST2. In addition to the immune cell subsets, according to ProteinAtlas there is positivity in most normal tissues, including myocytes, endothelial cells, respiratory epithelial cells, etc.

Minor Comments:

1. Figure 1H, there are $n=54$ LSC and $n=16$ HSC samples shown in the graph, however $n=192$ is referenced as being in the biobank on line 127. How were the 54 LSC samples chosen out of the 192 samples available?

2. Please update the labeling for Figure 2F and 2G. They are labeled as WT and ST2^{-/-} instead of ST2^{fl/fl} and ST2^{fl/fl} Mx1cre as in the figure legend. Update text on line 189 to indicate ST2^{fl/fl} and ST2^{fl/fl} Mx1cre MLL-AF9 leukemic cells.

3. Please comment on the differences observed in cell cycle stages between the initiation model (Figure 2F) and the maintenance model (Figure S5D). Much higher levels of G0 are seen in maintenance model and difference between the ST2^{fl/fl} and ST2^{fl/fl} Mx1cre for G1 and G2/S/M are larger in the initiation model compared to the maintenance model.

4. For ROS data, Figure 2F is shown as %ROS while Figure S5E is shown as MFI. Replot data so that it is represented consistently for the initiation and maintenance models. Is the data consistent between the two models or are there differences in ROS? If there are differences, please comment on why these differences might be observed.

5. Figure 4A, please update the figure legend to indicate what day the representative flow plots are for and show the same day for both LSKs and LT-HSCs. Is the labeling correct on the bar graphs for the LSKs or was the ST2 and IL-33 switched? The flow plots show 10.4 and 4.4% ST2⁺ for the 5-FU and vehicle, respectively but the bar graph shows ~25 and 10%. Figure S6 needs to be updated to show the same day for both populations (day not shown in Figure 4A) and update the legend to reflect the correct day it currently shows day 3 for LSK and day 5 for LT-HSC.

6. Figure 5H should be placed prior to panel 5G so that it follows the initial toxicity studies. Panel 5I should be moved to Figure 6 with the humanized AML model. If it is not the Molm14 model, then clarify which humanized model this is in the figure legend and methods.

7. For Figure 6E and F, what are the total numbers of CD8⁺, CD4⁺ and ST2⁺FoxP3⁺ T cells?

8. For Figure 6G, given that there is a reduction in LSCs and increase in CD8 ratio following BC281 treatment, please show ratio of CD8 to AML cells for both treatment groups. Currently, the text and figure legend don't indicate which treatment group is being shown.

9. Panel J is not labeled in Figure 6 but is referenced in the text (line 333).

10. Check line 327, should this be "leukemia burden decreased in the BC281+CD8⁺ T cells group...". Currently it is written as BC462+CD8⁺ T cells group.

11. For the AML PDX model, indicate why NSG-B2m mice were used and why T cells were activated prior to inoculation since NSG mice and HLA matched CD8 cells were used for the other humanized model.

12. Were PD-1, LAG-3 and TIM-3 expression assessed on T cells from MLL-AF9 model? If so, were they consistent with the P815 model? Is ST2 increased on exhausted T cells?

Reviewer #2

(Remarks to the Author)

In this study the authors have shown that high IL1RL1 gene expression correlates to survival. They show that IL-33 and its receptor ST2 contribute to leukemia growth in MLL/AF9 models. They have developed and tested a bi-specific T-cell engaging antibody which targets ST2 and drastically increases survival of mouse models. Overall this is an interesting and important study, the data with the antibody on the cell line in vivo is particularly striking, but there are some issues with interpretation or missing controls that need to be addressed prior to publication. The paper would also benefit from thorough proof-reading and the figures relabelling to help the reader (e.g. there are 5 panels in Figure 2A which should be labelled separately).

Major:

1. The paper discusses LSCs extensively but the data shown does not confirm LSC specificity, in fact several pieces of data contradict the concept that this is an LSC phenomenon. Figures 1-2 show data for ST2 expression in LSCs as inferred by conventional cell surface markers, but do not show the counterpart expression in non-LSCs. Is expression higher in LSCs or is it equally in all AML cells? The fact ST2 KO AML cells still show tertiary engraftment (Fig 2C) with 100% death by day 30 suggest some LSCs are at least still present though leukemogenesis is clearly impaired. Similarly, in Fig 7 the authors note relapse which implies surviving LSCs. The reviewer appreciates that LSCs are a heterogeneous population so not all may be killed by the treatments but the dramatic increase in survival rates suggests the majority of AML cells are being hit. This is not a bad thing but needs to be analysed and recognised.

2. To what extent is this an MLL/AF9 specific phenomenon? The mouse model, human cell line and human PDX model all carry MLL/AF9, figure 1E endeavours to show upregulation of ST2 in other AML backgrounds but does not show any karyotypic aberrations, including MLL/AF9 to compare. Six PDXs are shown but the other one used is MLL/AF9, at the very least in vitro assays should be done with the others although they all carry MLL translocations too.

As regards points 1 and 2, in two recent LSC gene expression datasets IL1RL1 looks to be reasonably LSC-specific in RUNX1/ETO AML (Kellaway 2024 Nat Comm) but not in PML/RARA APML (Jin 2024 Nat Comm).

3. The results in Figs 2E-G could suggest that the LSCs are becoming more stem-like and returning to a more quiescent state when the exogenous signal is removed. Targeting ST2 would potentially therefore keep LSCs quiescent when targeted leading to later relapse which has therapeutic implication. Are the IL33 KO mice LSCs also quiescent? Are they pushed back into cycle if IL33 is added back in?

4. What is the relevance of the section on mastocytoma in an AML paper? This does not add to the paper and is confusing dumped in the middle.

Minor:

1. Is there any validation available for the approach used in Figure 1F? The results seem very different between the two datasets and de-convolution of bulk RNA-seq is recognised for often performing poorly.
2. Figure 1J – please show all datapoints rather than a box plot for this data
3. Line 139 header “ST2 is required for leukemogenesis initiation”/line 170 conclusion. This is overstated as 100% of mice still died. This is particularly confusing in 2D where the data implies no LSCs are present yet all mice died of leukemia. Please explain these results better.
4. Gene expression data provided in Table S2 should be provided in full, all genes to confirm that gene expression decrease is not global.
5. The models need a brief introduction for non-experts – ST2 Mx1 CRE, IL33 cit/cit, humanized model of leukemia (line 294) in particular
6. What are the % of CD8 cells etc. in the IL33 KO mice compared to the WT in the absence of AML? Are the immune changes just a feature of the KO mouse or a direct result of AML interaction?
7. Are flow cytometry etc. results from mouse experiments from bone marrow or peripheral blood? The graphs say BM, the methods only state PB. As this paper studies microenvironment and AML cells were injected intravenously it is important that results are from BM.

Reviewer #3

(Remarks to the Author)

The manuscript entitled "Dual targeting of tumoral cells and immune microenvironment by blocking the IL-33/ST2 pathway" by Fu and colleagues discusses the role of ST2 and IL-33 axis in LSC maintenance and initiation and further implicates that they also have a potential impact on the immune microenvironment by inducing T cell tolerance. In the end, all the described observations led to the development of a therapeutic intervention to block the ST2/IL-33 pathway.

The paper has a well-organized presentation. However, it would be helpful to address several concerns so that the reader can reach the same conclusions that the authors made.

(1) All the datasets presented in figure 1 show a high expression of ST2 in AML, and the survival looks worse in those patients with high ST2. However, it might be helpful to break down the curves using other risk-stratifications used for AML. In other words, would favorable-risk AML have the least ST2, and would intermediate-risk AML be stratified better with ST expression?

(2) For the validation of the expression of ST2 in LSCs in Figure 1J, it would be necessary to (a) show expression before the therapy of patients who responded well and patients who did not respond. (b) show an additional marker to distinguish LSCs from HSCs since there are reports by the same authors that ST2 is important for HSCs. (c) what is the immune landscape on such samples, specifically on T cells? As part of the paper is to highlight the impact on the immune microenvironment.

Of note: Please add the numbers and ticks to the flow cytometry plots. Indicate the number of total events collected. Make a note on top of the gate that indicates what populations are shown: e.g., Figure 1J CD34+CD38-CD90+CD45RA- or LSC. Please show plots as scatter plots to see where each of the samples is.

(3) In all experiments performed to demonstrate the role of ST2 in leukemia initiation and maintenance, the animals died 10 to 20 days later; what is the cause of death? If it is leukemia, what happens in a secondary transplant?

(4) A secondary transplant would be informative for Figure 3 to demonstrate a role in LSC maintenance. Can the authors also show the level of leukemia burden when the plpC treatment started and the expression of ST2 before and during the treatment?

(5) Figure 4, what happens to the animals without ST2 in LSCs and no T cells? How much of the longer survival is driven by T cells being less tolerant?

(6) Do the authors have other hallmarks/signatures that show tolerance of T cells?

(7) Based on their paper cited (ref 12), one would not expect an evident phenotype of toxicity on mice just treated with the antibodies. Can the authors show functional assays for HSCs? Can the authors show what happens in the context of chemotherapy treatment that induces the expression of IL-33? This is thinking in the potential clinical application where patients will be previously treated or simultaneously treated with chemotherapy.

(8) Last figure with the PDX mice, why would the animals only treated with T cells die sooner? Is this a result of such a small "n" ?

Of note: Please clearly state that multimeric IL15/IL15R α was used on the figures.

Reviewer #4

(Remarks to the Author)

This manuscript builds on a growing body of literature proposing a role for the cytokine IL-33 and its receptor IL1RL1 (formerly ST2) in LSCs. This study expands on previous work showing that IL-33 synthesized by leukemia cells not only acts on LSCs through an autocrine loop but also influences the immune cells in the bone marrow microenvironment. Finally, the authors develop mouse and human bi-specific antibodies against IL1RL1 and show that they increase CD8+ recruitment, decrease leukemic burden, and increase survival. Collectively, this work provides compelling evidence supporting the development of IL1RL1-targeted therapies for the treatment of AML.

Comments:

1. According to the HUGO gene nomenclature committee website, IL1RL1 is now the official name for both the gene and the protein that was formerly called ST2.
2. In Figure 2E, the authors provide heat maps of genes deregulated in IL1RL1 deficient cells that are associated with oncogenesis, metabolomics, and cell cycle. However, it is unclear how these lists were compiled and if it was done in an unbiased way, such as gene set enrichment analysis or comparison to established KEGG gene lists.
3. According to the Materials and Methods section, the stain used in Fig. 2F was MitoTrackerCMXRox. This stain labels active mitochondria, not cytoplasmic reactive oxygen species, as stated in the text.
4. There is minimal description or characterization of the novel conditional IL-33 knockin mouse developed. It is unclear whether this allele could produce a truncated protein or whether the antibody used in Figure 4 would be able to recognize a protein lacking exons 6-8. In addition, there is no discussion as to whether this conditional allele induces defects in the CD8+ cells in non-leukemic mice.
5. In Figure 6I, the authors show that the combination of BC281 and ALT803 increases survival compared to the combination of the mutated antibody (BC462) and ALT803. However, they did not include any mice treated with BC281 alone in this experiment. Compared to the data shown in Figure 6C, the effect of ALT803 seems minimal, and may not be statistically significant. Consequently, it is difficult to conclude that ALT803 "improved survival compared to anti-ST2 T-BsAb treatment alone" as stated in line 546.

Version 1:

Reviewer comments:

Reviewer #1

(Remarks to the Author)

Fu et al. have sufficiently addressed all major comments concerning the manuscript.

The additional analyses and utilization of additional murine models more completely assesses the role of IL1R1/IL-33 in leukemia initiation, maintenance and utilizing IL1R1 as a target for bispecific antibody treatment.

There are additional minor comments to address in the manuscript:

1. For the primary AML samples analyzed for Figure 1K and J, what is the source of the samples? Please add information into the materials and methods and figure legend.
2. Figure 1 legend minor corrections: A) number of samples for TCGA dataset is missing from text; H) the legend indicates n=54 for the AML LSCs but the main text on line 136 indicates n=193 from the Princess Margaret leukemia biobank, please correct the discrepancy.
3. Lines 1215-1218 should be included with Figure 2A legend, not Figure 2B. For Figure 2B and 2C indicate the number of cells that were used for the data shown for the secondary and tertiary transplants.
4. Supplemental Figure 1F legend (line 83) indicates n=2 for the LSC (CD34+CD38+) while the non-LSC leukemic cells n=363. Is it correct that there are only 2 samples for the LSCs?
5. Supplementary Figure 4B: Are the labels for the representative flow plots switched? Please review and confirm if they are correct.
6. Supplemental Figure 12B and 12C legend: Based on the figure layout, PD-1+CD8+T cells should be included in the legend for 12B not 12C.
7. Supplemental Figure 16B: The bar graph is missing the y-axis label.
8. Supplemental Figure 22D: Please check and confirm the labeling is correct. The ratio of CD8 to AML cells is plotted very similar to Figure 6G, but the labeling of S22D suggests that the CD8:AML ratio at day 24 is 1:116 for BC281 treatment vs. 1:5 for BC462, which is the opposite of what is described in the text and the reduction in tumor burden and increased CD8 T cells would indicate.
9. Supplemental Figure 29: Please include the quantified luciferase radiance data for key time points that were imaged during the experiment (i.e. during dosing, end of dosing period, pre-reinduction and during reinduction) and survival curves in the figure. This is a nice treatment/re-treatment model and showing more of the quantified data would strengthen the data

Reviewer #2

(Remarks to the Author)

The authors have addressed all of my previous queries either by showing new data or appropriate changes to the text and the findings described by the manuscript are now sufficiently well evidenced for publication.

Reviewer #3

(Remarks to the Author)

The authors have addressed most of the concerns of all the reviewers. Overall, the manuscript has improved; the authors added significant information and added important figures.

Some minor comments to be addressed:

1. "Hispanic" is not a race; please change the subheading of the table and indicate race/ethnicity.
2. Add in the table of patients the percent of IL1RL1+LSCs and IL1RL1+IL-33+ LSCs
3. If the authors included FMOs for IL1RL1 and/or IL-33 in their flow cytometry panels, please include them in supplemental data.
4. If available, please include a table with the characteristics of the healthy donors (i.e., sex, age)
5. How do the patients split in the survival curves if the parameter is subdivided into quartiles?
6. Can the authors clarify when the sample shown was taken (when separating people as NR and CR)-figure 1 and supplemental Fig 1?
7. It will be helpful to use the same nomenclature in the table of patients and in the figures, e.i. Non-responder/Refractory.

Reviewer #4

(Remarks to the Author)

The others have appropriately addressed my concerns.

Point-by-point responses to reviewers' remarks (indicated by blue font).

Reviewer # 1:

This is a very thorough and descriptive manuscript with multiple different approaches being taken to address the role of ST2 in leukemia and mastocytosis.

We appreciate the reviewer's positive comments and thank the reviewer for his/her constructive criticisms and suggestions to improve the manuscript.

Major Comments:

1. The authors indicate that ST2 mRNA is increased across a number of molecularly defined AML subgroups relative to healthy donors. According to Naef et al. *Science Translational Medicine* 2023;16:eadd7705, ST2 expression was elevated primarily in t(8;21), t(9;22), del(9q), or DEK/NUP214 subgroups. Was the analysis criteria used for Figure 1E applied to the TCGA or MILE datasets and identified ST2 as being elevated in IDH1mut, IDH2mut, NPM1mut, CEBPAmut, FLT3-ITD, FLT3-TKD, NRASmut and EVIpos? Please include reference to Naef et al. analysis in the results and explain that the data presented here either supports and further expands or is it in contrast to already published data (and why there are differences between the analyses).

Response: We used a combination of GSE1159 (*Stirewalt DL et al. Genes Chromosomes Cancer* 2008. PMID: 17910043) and GSE6891 (*Verhaak RG et al. Haematologica* 2009. PMID: 18838472). This has been clarified in the methods and legends. We did not use the Microarray Innovations in Leukaemia (MILE) dataset as it was a comparison of microarrays to standard laboratory methods. Based on the reviewer's suggestion, we have now also analyzed the Naef et al.'s dataset which is presented in Supplementary Fig.1B and described in line 118-121. We confirmed, in this database that IL1RL1 is upregulated regardless of the cytogenetic abnormalities.

Note that based on Reviewer 4' s comment, we will use, in the revised manuscript, the new nomenclature for ST2 which is IL1RL1 (for both the gene and the protein) in human and Il1rl1 in mice.

2. The authors describe an autocrine loop induced by the LSCs however, co-expression of IL-33 and ST2 is not investigated. What is the co-expression of IL-33 and ST2 on the LSCs for the data included in Figure 4? For the autocrine loop involvement in immune tolerance data, expression of ST2 and IL-33 in the immune cells should be included as part of the analysis. What is the IL-33 and ST2 expression on the immune cells included as part of Figure 4D and S7?

Response: We now show in Fig.1K (patients' samples) the co-expression of IL1RL1 (previously ST2) and IL-33 in non-responders' and complete responders' samples which is described in line 146-148.

We also now show in new Fig. 4A and Supplementary Fig. 11A-B, for the 5-Fu induced stress model using IL-33^{eGFP} reporter mice, the co-expression of Il1r1 (previously ST2) and IL-33 which is described in line 278-281.

However, it is to note that in Figure 4B and S7, we cannot analyze the expression of IL-33 and co-expression of IL-33 and Il1r1 since mice are transferred with IL-33 KO LSCs devoid of IL-33.

3. Given the described expression of ST2 on immune cell populations (including T cells) and data from Figure 4D and S7 showing impacts of IL-33 knockout on MDSCs, Tregs, macrophages, dendritic cells, the effects of BC281 treatment on these populations in the tolerability study as well as the immunocompetent disease models needs to be assessed and discussed.

Response: We thank the reviewer for his/her comment and agree.

For tolerability: please also see response to your comment 10. Anti-IL1RL1 neutralizing antibodies are currently in clinical trials for asthma and chronic obstructive pulmonary disease (COPD) and have been shown to be safe and well tolerated. In our leukemia models, since we used the anti-Il1r1 T-BsAbs as a checkpoint inhibitor, we were mostly concerned by the in the gastrointestinal (GI) tract's toxicity which we have shown was not affected by BC281 (shown in Fig. 5G and 6M), lines 384-386, and lines 509.

For the immunocompetent models: please see new supplementary Fig. 22 and 23 that show in-depth analysis of the immune microenvironment changes in response to anti-Il1r1 T-BsAbs (BC281) vs control BC462 treatment. Indeed, Il1r1 expression, on MDSCs, Tregs, macrophages, dendritic cells, was decreased in the anti-Il1r1 T-BsAbs (BC281) vs control BC462 treated mice. These results are presented lines 425-453. We also tested the epigenetic DNMT3A/FLT3^{ITD} leukemic model (new supplementary Fig. 27) and showed similar effects of BC281 treatment. It has also been presented lines 471-493.

4. ST2 expression on the tumor cells should be indicated for the post-challenge samples for each of the models where it can be assessed, especially for the models where anti-ST2-CD3 treatment was used. Does BC281/BC282 treatment eliminate the ST2+ cells?

Response: Il1r1 expression on the leukemia cells and LSCs after treatment is now shown, and we found that Il1r1 expression on leukemic cells and LSCs in the anti-Il1r1 BC281 treated mice are much reduced as compared to the control BC462 treated mice as shown in new Supplementary Fig. 20, Supplementary Fig. 21, Supplementary Fig. 24B, and Supplementary Fig. 26B. Il1r1 expression on these tumor cells is less than 10% *in vivo* after treatment although not completely abrogated. These results are described lines 419-423, 461-464, and 469-470.

5. For all inoculation models, the authors need to clarify if only ST2+ cells were selected and inoculated or if bulk cells were inoculated. The schematics show percent ST2+ cells in the population but the text and figure legends are unclear as to whether those are first selected or not. It is important to understand if 100% of the starting tumor cells were ST2+ or if just a small population was.

Response: We apologize for the lack of clarity on these models. In the inoculation models, we used unsorted bulk cells where Il1r1+ cells represented only a subset population of the total cells transferred. We used the approach described in previous studies (*Chen, S., et al. Nat Commun. PMID: 31827082; Joydeep Ghosh., et al. J Clin Invest. 2016. PMID: 27294524; Park SM, et al. Cell Stem Cell. 2019. PMID: 30472158*). This has now been clarified in lines 166-171.

6. The weekly adoptive T cell experiment (Figure 6H) and ALT803 combination studies (Figure 6I) need to be compared to the appropriate controls. The overall survival and percent of CD3+CD8+ T cells looks similar between Figure 6C/E (no weekly adoptive T cell transfer) and Figure 6H (weekly T cell transfer). The combination of bispecific + ALT803 (Figure 6I) needs to include the BC462 alone and BC281 alone controls in the survival curve and graphs for Figure 6I. When compared to the bispecific alone, is there an expansion of the T cells and improvement in survival with ALT803? The survival curves for Figure 6C and 6I look very similar as do the percent T cell populations in 6E vs. 6I. Lines 545-546 in the discussion currently imply an improved survival when comparing anti-ST2 + IL-15 to anti-ST2 but the data in Figure 6 does not make this comparison as it is plotted. If the data are not significant, modify the statements in the discussion to reflect the data. The authors should also address if the remaining leukemic cells post-challenge are ST2- and whether continuing to inoculate T cells or trying to expand them with IL-15 would be expected to have an increased efficacy and why.

Response: All the controls have now been added in Figure 6H and 6I, and the manuscript has been edited accordingly. We also added Supplementary Fig. 25 showing the T cells expansion with ALT-803 injection in non-leukemic mice and described in lines 455-470. Il1r1 expression on leukemic cells and LSCs in these experiments has also been added in Supplementary Fig. 24B and described in lines 461-464 and Supplementary Fig. 26 which was depicted in lines 469-470.

Together, adoptive transfer of CD8+ T cells even more than ALT-803 improve AML survival because it provides, additional CD8+ T cells to be engaged by BC281, external in the case of the adoptive transfer and internal in the case of ALT-803. ALT-803 is used as a cost-effective alternative to adoptive transfer of CD8+ T cells that does not need cell manipulation. This is discussed in lines 713-717.

7. For the humanized NTPL-377 model, the authors need to comment on/clarify why NSG-SGM3 mice and 20×10^6 activated T cells were used for the experiment in Figure S12 in the results section when the comparator study (Figure 7) utilized NSG-B2m mice and 5×10^6 activated T cells. Also comment on why the overall survival was shorter for the animals in Figure S12 than Figure 7. BC282 + T cells and BC283 + T cell controls need to be included in the study to accurately assess whether the addition of IL-15 is improving survival compared to bispecific treatment alone.

Response:

In the PDX model with 0.3×10^6 NTPL-377 cells (Supplementary Fig. 12, now Figure 7C), we utilized NSG-SGM3 mice and administered 20×10^6 activated T cells per dose to create an aggressive PDX leukemia model that closely mimics clinical settings with rapid disease progression. The use of NSG-SGM3 mice was chosen due to these mice expressing the human cytokines IL3, GM-CSF and SCF, allowing for superior and stable engraftment of hematopoietic

lineages and primary AML samples (Janke LJ, et al., *Vet Pathol* 2021. *PubMed*: 33208054; Coughlan AM, et al. *Stem Cells Dev* 2016. *PubMed*: 26879149), thereby providing a suitable platform for evaluating the efficacy of T-BsAbs in a highly immunocompetent niche. In contrast, in Figure 7B, we employed NSG-B2m mice (King MA. et al., *Clin Exp Immunol* 2009. *PubMed*: 19659776) and 5×10^6 activated T cells per dose to ensure slower disease progression and an extended treatment window, allowing for a more precise assessment of the therapeutic effects of BC282 on leukemic mice survival.

We acknowledge the importance of including control BC283 + T cells in the study, so we repeated the anti-IL1rl1 T-BsAb treatment in combination with IL-15 in NTPL-377 PDX model. The shorter survival of animals in Figure S12 (now Figure 7C) compared to those in Figure 7 is attributed to the more aggressive disease progression in the NSG-SGM3 model. The higher T cell doses (20×10^6 per mouse) and the injection human IL-15 (1ug per dose per mouse) in NSG-SGM3 mice potentiate T cell immune response to enhance anti-leukemia immunity, and we did observe an improved survival in BC282-treated mice as compared to untreated mice and control BC283-T cells, and BC283 alone-treated mice. In contrast, the NSG-B2m model supports a more controlled disease progression, allowing for a longer survival window in Figure 7B.

All controls have been added with specifically:

For Figure 7B, the groups are 1) untreated, 2) human T cells (effector cells) alone, 3) control T-BsAbs BC283 with T cells, and 4) anti-hIL1RL1 T-BsAb BC282 with T cells.

For Figure S12 (now Figure 7C), in a different PDX model, the groups are 1) untreated, 2) control T-BsAbs BC283 with human T cells in combination with IL15/IL15R α , 3) anti-hIL1RL1 T-BsAb BC282 in combination with IL15/IL15R α without T cells (as a reminder NSG mice do not have any endogenous T cell to engage), and 4) anti-hIL1RL1 T-BsAb BC282 in combination with IL15/IL15R α with T cells.

Figure labels in Figure 7B and Figure 7C have been updated using BC282 and BC283 similarly to the other legends in the manuscript.

These updated have been described in the manuscript in lines 533-572.

8. Clarify whether the mice in Figure S13 were a continuation of study from Figure 7 or if this is a new study. If it is a new study, why were NSG mice used when NSG-B2m or NSG-SGM3 mice were used for Figure 7 and Figure S12, respectively? Survival curves need be plotted to demonstrate the “notable survival extension in BC282 treated PDXs as contrasted with both BC283 control and solitary t cells treatment” as stated in the text (lines 403-404).

Response: Supplementary Fig. 29 (formerly Fig. S13) continues the study presented in Figure 7B. We apologize for the typo regarding the mouse strain in the original Figure S13. As this figure is a continuation of the study in Figure 7B, the mouse strain remains unchanged. The text in the manuscript in the lines 537-543 has been updated accordingly.

9. The authors need to clarify the expression of ST2 on normal HSCs. Line 509 of the discussion: “Here, ST2 holds a distinct advantage because it is not constitutively expressed on normal HSCs” seems to contrast with the introduction (lines 81-82) that state “We and others,

also, found that ST2 is expressed on normal murine and human HSCs, respectively.” This does present a potential therapeutic liability and wouldn’t allow for the targeting minimal residual LSC disease as implied by the authors.

Response: We agree with the reviewer that these statements are confusing. As shown in new Supplementary Fig.1E in lines 148-150 and Supplementary Fig. 7 in lines 214-215, IL1RL1 expression on murine and human HSCs in steady state is expressed at low levels by flow cytometry as compared to LSCs. The statement “because it is not constitutively expressed on normal HSCs” has been removed.

In addition, see response to Reviewer#2 major comment #2, to determine whether IL1RL1 high expression is specific to LSCs, we analyzed bulk RNA sequencing data of AML patients (*Kellaway et al., Nat Commun 2024*, PMID: 38355578; *Jin et al., Nat Commun 2024*, PMID: 38365836). As shown in Supplementary Fig. 1F, our results indicate that this is not the case. It is updated lines 150-152.

10. The authors should discuss potential on-target off-tumor toxicity that could be relevant to targeting ST2. In addition to the immune cell subsets, according to ProteinAtlas there is positivity in most normal tissues, including myocytes, endothelial cells, respiratory epithelial cells, etc.

Response: The Protein atlas shows any IL1RL1 independent of its isoform. There are two spliced isoforms of IL1RL1: soluble (short IL1RL1), and membrane-bound (long IL1RL1) with opposite roles (reviewed in *Griesenauer et al. Frontier Immunology 2017*, PMID: 28484466). The soluble form of IL1RL1 is a decoy receptor of IL-33 and does not signal while the membrane-bound form of IL1RL1 binds to IL-33 and signals, mostly in T helper 2 cells, regulatory T cells, innate lymphoid cell type 2, M2 polarized macrophages, and mast cells. Since IL1RL1 was initially discovered on Th2 cells (*Townsend et al. J Exp Med, 2000*, PMID: 10727469), which is elevated in asthma and chronic obstructive pulmonary disease (COPD), several neutralizing IL1RL1 antibodies (i.e. astegolimab) have been developed. Safety of astegolimab has been established in a phase 2a placebo-controlled trial for COPD patients (*Yousuf et al. Lancet Respir Med, 2022*, PMID: 35339234). Similarly, astegolimab efficacy and safety in adults with severe asthma has been demonstrated in a randomized clinical trial (*Kelsen et al., J Allergy Clin Immunol, 2021*, PMID: 33872652). In our binding assays, our anti-IL1RL1 antibodies did not revealed any binding across a broad panel of normal human tissues (Supplementary Table 7 shown below).

Supplementary Table 7. Human Anti-IL1RL1 antibody doesn't not react with normal human tissues.

Human Tissues	Anti-IL1RL1 Ab	
	1ug/ml	2ug/ml
ILEUM	0	0
SKELETAL MUSCLE	0	0
CEREBELLUM	0	0

FRONTAL LOBE	0	0
PONS	0	0
STOMACH	0	0
SPINAL CORD	0	0
PANCREAS	0	0
LIVER	0	0
LUNG	0	0
SIGMOID COLON	0	0
SPLEEN	0	0
THYROID	0	0
KIDNEY	0	0

Additionally, since we and other have found large amount of IL1RL1⁺Tregs in the gastrointestinal (GI) tract and because we are using high doses of anti-IL1RL1 T-BsAb, we monitored, in the immunocompetent mice, potential GI toxicity by following body weight, and colon tract length which were unchanged in the BC281 treated group as compared to BC462 control group (Fig. 5G).

Minor Comments:

1. Figure 1H, there are n=54 LSC and n=16 HSC samples shown in the graph, however n=192 is referenced as being in the biobank on line 127. How were the 54 LSC samples chosen out of the 192 samples available?

Response: 54 patients were selected for AML LSCs defined as CD34⁺CD38⁻. This has been clarified in Fig. 1H legend.

2. Please update the labeling for Figure 2F and 2G. They are labeled as WT and ST2^{-/-} instead of ST2^{fl/fl} and ST2^{fl/fl} Mx1cre as in the figure legend. Update text on line 189 to indicate ST2^{fl/fl} and ST2^{fl/fl} Mx1cre MLL-AF9 leukemic cells.

Response: We thank the reviewer for noticing this. We have fixed it accordingly.

3. Please comment on the differences observed in cell cycle stages between the initiation model (Figure 2F) and the maintenance model (Figure S5D). Much higher levels of G0 are seen in maintenance model and difference between the ST2^{fl/fl} and ST2^{fl/fl} Mx1cre for G1 and G2/S/M are larger in the initiation model compared to the maintenance model.

Response: We appreciate the reviewer's thorough observation. We are sorry for this oversight, and we have updated the cell cycle data in new Supplementary Fig. 10D (previous Figure S5).

4. For ROS data, Figure 2F is shown as %ROS while Figure S5E is shown as MFI. Replot data so that it is represented consistently for the initiation and maintenance models. Is the data consistent between the two models or are there differences in ROS? If there are differences, please comment on why these differences might be observed.

Response: Thanks very much for the reviewer' detail suggestion. We have updated Fig. 2F.

5. Figure 4A, please update the figure legend to indicate what day the representative flow plots are for and show the same day for both LSKs and LT-HSCs. Is the labeling correct on the bar graphs for the LSKs or was the ST2 and IL-33 switched? The flow plots show 10.4 and 4.4% ST2+ for the 5-FU and vehicle, respectively but the bar graph shows ~25 and 10%. Figure S6 needs to be updated to show the same day for both populations (day not shown in Figure 4A) and update the legend to reflect the correct day it currently shows day 3 for LSK and day 5 for LT-HSC.

Response: We are sorry for the confusion in the labeling. We have now updated Figure 4A and Supplementary Fig. 11, as well as figure legends.

6. Figure 5H should be placed prior to panel 5G so that it follows the initial toxicity studies. Panel 5I should be moved to Figure 6 with the humanized AML model. If it is not the Molm14 model, then clarify which humanized model this is in the figure legend and methods.

Response: We agreed with the reviewer's comments. Figure 5H has now been placed prior to panel 5G. Panel 5I has been moved to Figure 6M. Figure legends for all these figures are also updated.

7. For Figure 6E and F, what are the total numbers of CD8+, CD4+ and ST2+FoxP3+ T cells?

Response: The total numbers of CD8+, CD4+ and Il1r1+FoxP3+ T cells corresponding to Fig. 6E and 6F are provided in Supplementary Fig. 22A-C.

8. For Figure 6G, given that there is a reduction in LSCs and increase in CD8 ratio following BC281 treatment, please show ratio of CD8 to AML cells for both treatment groups. Currently, the text and figure legend don't indicate which treatment group is being shown.

Response: Fig. 6G shows the ratio of CD8 to AML cells in the MLL-AF9 leukemic model at different timepoints post-AML challenge but without treatment intervention. This has been clarified in the manuscript and figure legend. In addition, the ratio of CD8 to AML cells for BC281 vs BC462 treatment groups are shown in Supplementary Fig. 22D.

9. Panel J is not labeled in Figure 6 but is referenced in the text (line 333).

Response: Panel J in Figure 6 has been labeled, as well as in the figure legend.

10. Check line 327, should this be "leukemia burden decreased in the BC281+CD8+ T cells group...". Currently it is written as BC462+CD8+ T cells group.

Response: Thanks for noticing, this has now been fixed in new line 458-459.

11. For the AML PDX model, indicate why NSG-B2m mice were used and why T cells were activated prior to inoculation since NSG mice and HLA matched CD8 cells were used for the other humanized model.

Response: Please also see response to major comment #7.

In the PDX model with 0.3×10^6 NTPL-377 cells (Figure S12, now Figure 7C), we utilized NSG-SGM3 mice and administered 20×10^6 activated T cells per dose to create an aggressive PDX leukemia model that closely mimics clinical settings with rapid disease progression. The use of NSG-SGM3 mice was chosen due to these mice expressing human IL3, GM-CSF and SCF combine the features of the highly immunodeficient NSG mouse with cytokines, allowing superior stable engraftment of hematopoietic lineages and primary AML samples (*Janke LJ, et al., Vet Pathol 2021. PubMed: 33208054; Coughlan AM, et al. Stem Cells Dev 2016. PubMed: 26879149*), thereby providing a suitable platform for evaluating the efficacy of T-BsAbs in a highly immunocompetent niche. In contrast, in Figure 7B, we employed NSG-B2m mice (*King MA, et al. Clin Exp Immunol 2009. PubMed: 19659776*) and 5×10^6 activated T cells per dose to ensure slower disease progression and an extended treatment window, allowing for a more precise assessment of the therapeutic effects of BC282 on leukemic mice survival.

12. Were PD-1, LAG-3 and TIM-3 expression assessed on T cells from MLL-AF9 model? If so, were they consistent with the P815 model? Is ST2 increased on exhausted T cells?

Response: As requested by reviewer 2 in major comment #4, we removed the data on the mastocytoma model. Instead, we analyzed in the MLL-AF9 model the exhaustion markers on T cells and observed that the frequencies of PD-1⁺, LAG3⁺, TIM3⁺ and PD-1⁺TIM3⁺CD8⁺ T cells are decreased in the BC281 vs control BC462 treated mice. In addition, Il1rl1 expression in PD-1⁺TIM3⁺CD8⁺ T cells is also down-regulated in BC281 treated group in contrast to BC462 treated group. These data are now shown in Supplementary Fig. 23A-C.

To further assess the impact of anti-murine Il1rl1 T-BsAb in an alternative clinically relevant immunocompetent leukemia model, we utilized the epigenetic DNMT3A/FLT3ITD leukemic model, which frequently develops extramedullary leukemia in the spleen and liver. Similar to findings in the MLL-AF9 leukemia model, we observed that BC281 treatment led to a significant decrease of exhausted PD-1⁺TIM3⁺CD8⁺ T cells frequencies (Supplementary Fig. 27E).

Reviewer #2:

In this study the authors have shown that high IL1RL1 gene expression correlates to survival. They show that IL-33 and its receptor ST2 contribute to leukemia growth in MLL/AF9 models. They have developed and tested a bi-specific T-cell engaging antibody which targets ST2 and drastically increases survival of mouse models. Overall this is an interesting and important study, the data with the antibody on the cell line in vivo is particularly striking, but there are some issues with interpretation or missing controls that need to be addressed prior to publication. The paper would also benefit from thorough proof-reading and the figures relabelling to help the reader (e.g. there are 5 panels in Figure 2A which should be labelled separately).

Major:

1. The paper discusses LSCs extensively, but the data shown does not confirm LSC specificity, in fact several pieces of data contradict the concept that this is an LSC phenomenon. Figures 1-2 show data for ST2 expression in LSCs as inferred by conventional cell surface markers, but do not show the counterpart expression in non-LSCs. Is expression higher in LSCs or is it equally in all AML cells? The fact ST2 KO AML cells still show tertiary engraftment (Fig 2C) with 100% death by day 30 suggest some LSCs are at least still present though leukemogenesis is clearly impaired. Similarly, in Fig 7 the authors note relapse which implies surviving LSCs. The reviewer appreciates that LSCs are a heterogeneous population so not all may be killed by the treatments but the dramatic increase in survival rates suggests the majority of AML cells are being hit. This is not a bad thing but needs to be analysed and recognised.

Response: We agree with the reviewer. We have modified the text in the results and discussion to mention we target a heterogeneous population of leukemic cells including progenitors and LSCs. Indeed, we have analyzed by flow cytometry Il1r1 expression in non-LSC leukemic progenitors including CMP, MEP, MPP, and GMP in the Il1r1 leukemogenesis initiation model and found that Il1r1 was upregulated in these leukemic progenitors as compared to HSCs (Supplementary Fig. 7B). We also analyzed Il1r1 expression in these subpopulations in anti-Il1r1 T-BsAbs BC281 vs control BC462 treated leukemic mice. Il1r1 expression was reduced in these subsets post BC281 treated as compared to those mice treated with BC462 (Supplementary Fig. 21). This suggests that BC281 treatment may decrease both LSCs and leukemic progenitors, which was updated in lines 216-217, and 421-423.

2. To what extent is this an MLL/AF9 specific phenomenon? The mouse model, human cell line and human PDX model all carry MLL/AF9, figure 1E endeavours to show upregulation of ST2 in other AML backgrounds but does not show any karyotypic aberrations, including MLL/AF9 to compare.

Response: Thank you for your thoughtful comments.

- (i) We have now compared IL1RL1 expression in AML patients with different karyotypic aberrations including MLL-AF9 shown in Supplementary Fig.1A-B and found that Il1r1 is higher in these AML patients with different cytogenetic abnormalities as compared to healthy donors regardless of the type of cytogenetic abnormality (lines 120-124).
- (ii) We have also added 2 PDX lines that were not MLL/AF9 (see response below)
- (iii) We have also added the epigenetic DNMT3A/FLT3ITD leukemic model and showed similar findings as the MLL-AF9 model (new supplementary Fig. 27)

Together, our results suggest IL1RL1 overexpression in AML is not MLL-AF9 specific and is not cytogenetically driven.

Six PDXs are shown but the other one used is MLL/AF9, at the very least in vitro assays should be done with the others although they all carry MLL translocations too.

Response: We appreciate the reviewer's insightful suggestion. We have expanded our analysis to include two additional PDX cell lines without MLL oncogenesis mutations (NTPL-301 and

NTPL-511) and assessed IL1RL1 expression in these models. As shown in Figure 7A, IL1RL1 expression was also elevated in these two non-MLL PDX cell lines. Furthermore, to evaluate the cytotoxic activity of BC282, we conducted an *in vitro* killing assay using two MLL oncofusion PDX cell lines (NTPL-377 and DF-5) alongside the two non-MLL PDX cell lines (NTPL-301 and NTPL-511). Our results demonstrated that BC282, when co-cultured with these PDX models in the presence of human CD8⁺ T cells, exhibited potent cytotoxicity across both MLL and non-MLL PDX cell lines as compared to BC283 control. These findings suggest that BC282-mediated cytotoxicity is not restricted to MLL oncofusion-positive models. The corresponding data are provided in Supplementary Figure 28 (lines 527-528).

As regards points 1 and 2, in two recent LSC gene expression datasets IL1RL1 looks to be reasonably LSC-specific in RUNX1/ETO AML (Kellaway 2024 Nat Comm) but not in PML/RARA APML (Jin 2024 Nat Comm).

Response: Thank you for the suggestion. We re-analyzed IL1RL1 expression in LSCs (defined as CD34⁺CD38⁻) and non-LSCs from AML patients using the two datasets (GSE226603 and GSE172057) suggested by the reviewer, as reported in Kellaway et al., Nat Commun 2024 and Jin et al., Nat Commun 2024. As shown in Supplementary Fig. 1F, our analysis revealed no significant difference in IL1RL1 expression between LSCs and non-LSCs. This has been updated lines 150-152.

3. The results in Figs 2E-G could suggest that the LSCs are becoming more stem-like and returning to a more quiescent state when the exogenous signal is removed. Targeting ST2 would potentially therefore keep LSCs quiescent when targeted leading to later relapse which has therapeutic implication. Are the IL33 KO mice LSCs also quiescent? Are they pushed back into cycle if IL33 is added back in?

Response: We have expanded cell cycle, mitochondrial ROS, and mitochondrial potential analysis in IL-33/Il1r1 signaling in the LSC autocrine loop model in Supplementary Fig.14A-C. We found the same trend as shown in Figure 2E-G, text lines 330-335. It is therefore possible that targeting ST2 (Il1r1) would potentially keep LSCs quiescent and lead to late relapse. As a matter of fact, in our own PDX data, we have shown late relapse that can be treated with a reinduction therapy (Supplementary Fig.29) as it is often the case with immune targets and T-BsAb.

4. What is the relevance of the section on mastocytoma in an AML paper? This does not add to the paper and is confusing dumped in the middle.

Response: We removed the data on mastocytoma but added the DNMT3A/FLT3^{ITD} leukemic model to have two immunocompetent models.

Minor:

1. Is there any validation available for the approach used in Figure 1F? The results seem very different between the two datasets and de-convolution of bulk RNA-seq is recognised for often performing poorly.

Response: Thanks for the reviewer's insightful comment; we agree that deconvolution of bulk RNA-seq might be performed poorly. However, here, IL1RL1 was found higher in LSCs and

progenitors in two datasets (Fig. 1F), and further validated in an independent AMLCG1999 database in Figure 1G. We have also analyzed by flow cytometry (protein level) in our murine leukemia models, Il1r1 expression in LSCs and non-LSC leukemic progenitors including CMP, MEP, MPP, and GMP (Supplementary Fig. 7A-B).

2. Figure 1J – please show all datapoints rather than a box plot for this data

Response: Figure 1J now show all data points.

3. Line 139 header “ST2 is required for leukemogenesis initiation”/line 170 conclusion. This is overstated as 100% of mice still died. This is particularly confusing in 2D where the data implies no LSCs are present, yet all mice died of leukemia. Please explain these results better.

Response: We appreciate the reviewer’s critical assessment and agree that the original statement may have overstated the requirement of ST2 in leukemogenesis initiation. To address this, we have revised the section title to “Il1r1 promotes leukemogenesis initiation” in new line 162. We have also modified our paragraph conclusion (lines 219-220).

4. Gene expression data provided in Table S2 should be provided in full, all genes to confirm that gene expression decrease is not global.

Response: Differential expressed gene list was provided in Supplementary Table 2 in full.

5. The models need a brief introduction for non-experts – ST2 Mx1 CRE, IL33 cit/cit, humanized model of leukemia (line 294) in particular.

Response: We are sorry for confusion. We have updated the text with a brief introduction for a) Il1r1^{Mx1Cre} lines 184-189, b) IL-33^{cit/cit KO} lines 285-291, and c) humanized model of leukemia line 501-504.

6. What are the % of CD8 cells etc. in the IL33 KO mice compared to the WT in the absence of AML? Are the immune changes just a feature of the KO mouse or a direct result of AML interaction?

Response: Frequencies of CD8, CD4, Tregs in IL-33^{cit/cit KO} vs WT normal mice have been added in Supplementary Fig.13, and we did not observe differences.

7. Are flow cytometry etc. results from mouse experiments from bone marrow or peripheral blood? The graphs say BM, the methods only state PB. As this paper studies microenvironment and AML cells were injected intravenously it is important that results are from BM.

Response: We agree with the reviewer that results from BM are the most important and all the results shown in the manuscript are from BM. On occasion peripheral blood was collected for tumor monitoring and it is now specified when it is the case.

Reviewer #3:

The manuscript entitled "Dual targeting of tumoral cells and immune microenvironment by blocking the IL-33/ST2 pathway" by Fu and colleagues discusses the role of ST2 and IL-33 axis in LSC maintenance and initiation and further implicates that they also have a potential impact on the immune microenvironment by inducing T cell tolerance. In the end, all the described observations led to the development of a therapeutic intervention to block the ST2/IL-33 pathway.

The paper has a well-organized presentation. However, it would be helpful to address several concerns so that the reader can reach the same conclusions that the authors made.

(1) All the datasets presented in figure 1 show a high expression of ST2 in AML, and the survival looks worse in those patients with high ST2. However, it might be helpful to break down the curves using other risk-stratifications used for AML. In other words, would favorable-risk AML have the least ST2, and would intermediate-risk AML be stratified better with ST2 expression? **Response: This is an important translational question. We have now analyzed:**

- (i) IL1RL1 expression in the favorable risk AML [t(8;21), t(15;17), inv(16)] vs. intermediate-risk AML (characterized by the absence of favorable or unfavorable cytogenetic and molecular abnormalities) and did not show difference (add these data as Supplementary Fig. 1C)
- (ii) Focusing on the intermediate-risk group and using datasets that have information available (TCGA, n=90), no difference in survival was observed between IL1RL1^{low} and IL1RL1^{high} groups (Supplementary Fig. 1C).

Together, these data suggest that the alarmin IL1RL1 is not driven by unfavorable cytogenetic.

(2) For the validation of the expression of ST2 in LSCs in Figure 1J, it would be necessary to (a) show expression before the therapy of patients who responded well and patients who did not respond. (b) show an additional marker to distinguish LSCs from HSCs since there are reports by the same authors that ST2 is important for HSCs. (c) what is the immune landscape on such samples, specifically on T cells? As part of the paper is to highlight the impact on the immune microenvironment.

Response: We appreciate the reviewer's suggestions although some requests are not feasible.

a) show expression before the therapy of patients who responded well and patients who did not respond.

We do not have access to the responders' and non-responders' samples prior to therapy. However, we have access to 8 AML patients' samples at diagnosis that are subsequently used for PDX model development (see Supplementary Table 8). As shown in Fig. 7A, all these cells

before expansion express IL1RL1. Of note, we now also present data in Figure 1K of the co-expression of IL1RL1 and IL-33 non-responders (NR) and complete responders (CR).

(b) show an additional marker to distinguish LSCs from HSCs since there are reports by the same authors that ST2 is important for HSCs.

Response: Il1rl1 (previously ST2) was detected at an extremely low level on murine HSCs in our previous report. Unfortunately, there is no universally accepted surface marker that definitively differentiates LSCs from HSCs. While various surface markers have been proposed, their expression overlaps between LSCs and HSCs, making it challenging to establish a clear distinction based solely on surface phenotype. Leukemia-associated molecular mutations, such as DNMT3A may serve as a more reliable identifier of LSCs but cannot be combined in a flow panel.

c) what is the immune landscape on such samples, specifically on T cells? As part of the paper is to highlight the impact on the immune microenvironment.

Response: Proportions of CD8⁺ cytotoxic T lymphocytes have been shown to be decreased while frequencies of CD3⁺FOXP3⁺ regulatory T cells (Tregs) were increased in AML at diagnosis as compared to HD in previous study (*van Galen, P. et al., Cell, 2019, PMID: 30827681*). We explored these populations frequencies following chemotherapy induction in the NR and CR patients and found similarly to this previous study a depletion of CD8⁺ T cells, including cytotoxic CD8⁺IFN- γ ⁺ T cells and an increase in CD4⁺FOXP3⁺ T cells in the NR vs. CR patients (Supplementary Fig. 3A-C), lines 152-158.

In the immunocompetent MLL-AF9 model, treatment with anti-Il1rl1 BC281 vs BC462 control decreased frequencies and absolute number of Il1rl1⁺ Tregs while increased frequencies and counts of CD3⁺CD8⁺ and antigen specific WT1⁺CD8⁺T cells (Fig. 6, Supplementary Fig.22).

In the immunocompetent epigenetic DNMT3A/FLT3^{ITD} leukemic model treatment with BC281 vs BC462 resulted in similar changes as in the MLL-AF9 model (Supplementary Fig. 27B).

Of note: Please add the numbers and ticks to the flow cytometry plots. Indicate the number of total events collected. Make a note on top of the gate that indicates what populations are shown: e.g., Figure 1J CD34+CD38-CD90+CD45RA—or LSC. Please show plots as scatter plots to see where each of the samples is.

Response: We thank the reviewer for his/her suggestion. Flow cytometry plots with the numbers and ticks were show in Fig.1J and 1K. The statistical plots are shown in scatter plots.

(3) In all experiments performed to demonstrate the role of ST2 in leukemia initiation and maintenance, the animals died 10 to 20 days later; what is the cause of death? If it is leukemia, what happens in a secondary transplant?

Response: The cause of death in all instances is leukemia as verified by leukemia burden (gfp leukemic blast > 30% in the BM). For the secondary transplants, 0.5x10⁶ unsorted BM cells from the primary AML mice were collected at Day 25 post-challenge (~60% leukemic cells) and transplanted in secondary irradiated recipients. For the tertiary transplants, 0.5x10⁶ total unsorted BM cells from the secondary recipients collected at Day 25 (> 70-80% leukemic cells) were transplanted in tertiary irradiated recipients. The higher tumor burden at day 25 in

secondary transplant recipients resulted in the mice dying faster. Details of these experiments are presented in lines 860-871.

(4) A secondary transplant would be informative for Figure 3 to demonstrate a role in LSC maintenance. Can the authors also show the level of leukemia burden when the plpC treatment started and the expression of ST2 before and during the treatment?

Response: Leukemia burden, represented by the frequency of MLL-AF9⁺ cells at the start of plpC treatment on day 4 and during treatment on day 14, is shown in Supplementary Figure 9A. We are sorry but we do not have secondary transplant for this experiment.

(5) Figure 4, what happens to the animals without ST2 in LSCs and no T cells? How much of the longer is survival driven by T cells being less tolerant?

Response: We sincerely appreciate the reviewer's insightful suggestion. To address this, we performed transplantation experiments following myeloablative irradiation, using WT vs. Il1rl1^{-/-} LSCs along with T cell-depleted bone marrow cells as supporting cells. Our results indicate that Il1rl1 deficiency in LSCs, independent of immune cells, inhibits AML progression and improves survival, as shown in Supplementary Figure 15.

Additionally, we generated IL1RL1^{-/-} and scramble MOLM14 cells using CRISPR/Cas9 (Supplementary Figure 16) and transplanted these cells into NSG mice without adoptive T cell transfer. The findings demonstrate that IL1RL1 deficiency in leukemic cells extends survival solely through tumor-intrinsic mechanisms, independent of CD8⁺ T cell-mediated immunity (Supplementary Figure 17). These results suggest that while reduced T cell tolerance may contribute to improved survival, IL1RL1 deficiency in LSCs itself plays a critical role in suppressing AML progression.

(6) Do the authors have other hallmarks/signatures that show tolerance of T cells?

Response: We have found that anti-Il1rl1 T-BsAb treatment decreases suppressive Tregs and CD8⁺T cells exhaustion while increasing CD8⁺T cells stemness and effector function (See Supplementary Fig. 23). Specifically, frequencies of PD-1⁺CD8⁺, Lag3⁺CD8⁺, TIM3⁺CD8⁺ and TIM3⁺PD-1⁺CD8⁺ T cells in the BM of leukemic mice on day 10 post BC281 treatment were decreased as compared to control BC462 treatment while frequencies of CD44⁺CD62L⁻CD8⁺, TCF7⁺CD8⁺, CD27⁺CD8⁺, CD28⁺CD8⁺, Tbet⁺CD8⁺T cells, GZMB-expressing CD8⁺T cells, degranulating CD107a⁺CD8⁺ T cells, perforin⁺CD8⁺T cells, and IFN-γ⁺CD8⁺ T cells were increased. Furthermore, IL-10⁺Foxp3⁺, TGF-β⁺Foxp3⁺, and CD39⁺Foxp3⁺ Tregs in the BM of BC281 treated mice were decreased as compared to control BC462 treated mice. Finally, after T cell engagement of BC281 vs BC462 control, Il1rl1 expression decreased in Foxp3⁺Tregs, macrophages, myeloid-derived suppressor cells (MDSCs) and CD11c⁺DCs (Supplementary Fig. 23F). Please see updates in the text, lines 440-453.

(7) Based on their paper cited (ref 12), one would not expect an evident phenotype of toxicity on mice just treated with the antibodies. Can the authors show functional assays for HSCs?

Response: we agree with this comment for HSCs. We have now conducted *in vivo* HSC functional assays in normal hematopoiesis using Il1rl1^{fl/fl} Mx1Cre vs Il1rl1^{fl/fl} normal mice model as

shown in Supplementary Fig. 4. We found that Il1r1 deficiency impairs normal hematopoiesis, and this has been updated in lines 181-205.

Can the authors show what happens in the context of chemotherapy treatment that induces the expression of IL-33? This is thinking in the potential clinical application where patients will be previously treated or simultaneously treated with chemotherapy.

Response: We appreciate the reviewer's point of view. While we were revising the current study, Yang et al (*Yang, YE., et al. Cell Death Dis, 2024, PMID: 38778059*), reported in the context of lung cancer that cisplatin-induces IL-33 signals, which promotes M2 macrophages to reduce the anti-tumor effect of cisplatin. IL-33 stimulates membrane-bound Il1r1 (previously ST2) upregulation by activating NF- κ B and Rab37 facilitates the transport of membrane-bound Il1r1 to the plasma membrane in M2 macrophages. This IL-33/NF- κ B/Il1r1/Rab37 axis drives positive-feedback loops that amplify Il1r1 expression and facilitate its membrane trafficking in M2 macrophages. Therefore, the combination of cisplatin and neutralizing antibodies against IL33 or Il1r1 has been shown to alleviate cisplatin resistance in lung cancer models.

Although T-BsAbs are designed to avoid the need for chemotherapy, recent studies of combination of chemotherapy and anti-CD19 T-BsAb (Blinatumomab) have shown improved survival rates in randomized clinical trials for standard-risk B cell acute lymphoblastic leukemia in children and this question will be of interest to us, our current pre-clinical models, utilized in our study, do not allow us to test the question posed by the reviewer because standard chemotherapy regimens lead to the eradication of malignant cells in mice, preventing post-treatment analysis.

This is now discussed lines 738-745.

(8) Last figure with the PDX mice, why would the animals only treated with T cells die sooner? Is this a result of such a small "n" ?

Response: We observed that mice treated solely with T cells exhibited a shorter survival time compared to those treated with BC283 control plus T cells, as reflected in the survival curve. However, statistical analysis revealed no significant difference between these two groups. This may be attributable to the small sample size (n) used in the experiment, which could limit the power of the study to detect subtle differences, another possibility, however, is that BC283-engaged human xenogeneic T cells become less polyclonal leading to less xenogeneic graft-versus-host-disease (GVHD).

Of note: Please clearly state that multimeric IL15/IL15R α was used on the figures.

Response: We appreciate the reviewer's insightful suggestion. We utilized the multimeric IL-15/IL-15R α complex provided by our collaborators in Dr. Cheung's lab (*Xu, H., et al. Oncoimmunology, 2021, PMID: 33763293*), and this has been updated in Figure 7.

Reviewer #4:

Comments:

1. According to the HUGO gene nomenclature committee website, IL1RL1 is now the official name for both the gene and the protein that was formerly called ST2.

Response: The reviewer is correct. As the reviewer can guess, this project was started before the change in HUGO nomenclature, and we had continued to use the old nomenclature. That said, we agree also that it was a confusing nomenclature and we have therefore changed, throughout the manuscript, ST2 protein for IL1RL1 in human and Il1rl1 in mice.

2. In Figure 2E, the authors provide heat maps of genes deregulated in IL1RL1 deficient cells that are associated with oncogenesis, metabolomics, and cell cycle. However, it is unclear how these lists were compiled and if it was done in an unbiased way, such as gene set enrichment analysis or comparison to established KEGG gene lists.

Response: In Figure 2E, we, first, selected the differentially expressed genes in the heatmaps comparing WT and Il1rl1^{-/-} MLL-AF9 LSCs based on literature search. We, then, used pheatmap R package to scale these genes expression difference by heatmap. In addition, we conducted unbiased KEGG pathway analysis described in lines 1065-1067, and found that up-regulated genes in WT LSCs were involved in cell differentiation, adhesion, oxidative stress, and regulation of hemopoiesis, while down-regulated genes are related with immune system process, inflammatory response, cytokine production and leukocyte migration, as shown in Supplementary Fig.8A-B.

3. According to the Materials and Methods section, the stain used in Fig. 2F was MitoTrackerCMXRos. This stain labels active mitochondria, not cytoplasmic reactive oxygen species, as stated in the text.

Response: We are sorry for this oversight and edited accordingly. In addition, we have added the MitoTracker deep red data in Supplementary Fig. 7C.

4. There is minimal description or characterization of the novel conditional IL-33 knockin mouse developed. It is unclear whether this allele could produce a truncated protein or whether the antibody used in Figure 4 would be able to recognize a protein lacking exons 6-8.

Response: A brief description of the conditional IL-33 knockin mouse was added in lines 292-299 and the reference that describes how it was generated was referenced.

In addition, there is no discussion as to whether this conditional allele induces defects in the CD8⁺ cells in non-leukemic mice.

Response: Frequencies of CD8, CD4, Tregs in IL-33^{cit/cit} KO vs WT normal mice have been added in Supplementary Fig.13, and we did not observe difference at steady state (non-leukemic).

5. In Figure 6I, the authors show that the combination of BC281 and ALT803 increases survival compared to the combination of the mutated antibody (BC462) and ALT803. However, they did not include any mice treated with BC281 alone in this experiment. Compared to the data shown in Figure 6C, the effect of ALT803 seems minimal, and may not be statistically significant.

Consequently, it is difficult to conclude that ALT803 "improved survival compared to anti-ST2 T-BsAb treatment alone" as stated in line 546.

Response: We originally wanted to simplify this experiment but agree with the reviewers that adding all the controls will help with thorough comparison. All the controls have now been added in Figure 6H and 6I, and the text in the manuscript has been edited accordingly. We also added Supplementary Fig. 25 showing T cells expansion with ALT-803 injection in non-leukemic mice.

Reviewer #1 (Remarks to the Author):

Fu et al. have sufficiently addressed all major comments concerning the manuscript. The additional analyses and utilization of additional murine models more completely assesses the role of IL1R1/IL-33 in leukemia initiation, maintenance and utilizing IL1R1 as a target for bispecific antibody treatment.

There are additional minor comments to address in the manuscript:

1. For the primary AML samples analyzed for Figure 1K and J, what is the source of the samples? Please add information into the materials and methods and figure legend.

Response: We apologize for the lack of clarity in our initial description. We have now updated the Materials and Methods section to clearly specify the source of the primary AML samples used in Figures 1J and 1K. Additionally, the Figure 1 legend has been revised to include this information.

2. Figure 1 legend minor corrections: A) number of samples for TCGA dataset is missing from text; H) the legend indicates n=54 for the AML LSCs but the main text on line 136 indicates n=193 from the Princess Margaret leukemia biobank, please correct the discrepancy.

Response: We thank the reviewer for carefully identifying these discrepancies. The sample size for the TCGA dataset has now been added to the Figure 1A legend. Additionally, the inconsistency between the Figure 1H legend and the main text regarding the AML LSC sample size has been corrected. Both the Figure 1A and 1H legends have been updated accordingly.

3. Lines 1215-1218 should be included with Figure 2A legend, not Figure 2B. For Figure 2B and 2C indicate the number of cells that were used for the data shown for the secondary and tertiary transplants.

Response: Thanks for pointing out the unclear description. We have moved lines 1215-1218 to Figure 2A legend. The number of cells were clearly indicated in Figure 2B and 2C legends.

4. Supplemental Figure 1F legend (line 83) indicates n=2 for the LSC (CD34+CD38+) while the non-LSC leukemic cells n=363. Is it correct that there are only 2 samples for the LSCs?

Response: We re-analyzed IL1RL1 expression in LSCs (defined as CD34⁺CD38⁻) and non-LSCs from AML patients using the two datasets (GSE226603 and GSE172057), as reported in Kellaway et al., Nat Commun 2024 and Jin et al., Nat Commun 2024. Unfortunately, in this dataset, there are only 2 samples for the LSCs. This is shown in Supplementary Figure 1F.

5. Supplementary Figure 4B: Are the labels for the representative flow plots switched? Please review and confirm if they are correct.

Response: Thank you for pointing out the mislabeling in Supplementary Figure 4B. We have reviewed and corrected the labels accordingly.

6. Supplemental Figure 12B and 12C legend: Based on the figure layout, PD-1⁺CD8⁺T cells should be included in the legend for 12B not 12C.

Response: Thank you for your careful review. The PD-1⁺CD8⁺T cells have now been correctly moved to the legend for Supplementary Figure 12C, and the manuscript has been updated accordingly.

7. Supplemental Figure 16B: The bar graph is missing the y-axis label.

Response: Thank you for pointing out the missing label. We have updated the y-axis label in Supplementary Figure 16B.

8. Supplemental Figure 22D: Please check and confirm the labeling is correct. The ratio of CD8 to AML cells is plotted very similar to Figure 6G, but the labeling of S22D suggests that the CD8:AML ratio at day 24 is 1:116 for BC281 treatment vs. 1:5 for BC462, which is the opposite of what is described in the text and the reduction in tumor burden and increased CD8 T cells would indicate.

Response: We apologize for the mislabeling. The reviewer is correct, and we have corrected the labeling accordingly. The updated version is now reflected in Supplementary Figure 22D.

9. Supplemental Figure 29: Please include the quantified luciferase radiance data for key time points that were imaged during the experiment (i.e. during dosing, end of dosing period, pre-reinduction and during reinduction) and survival curves in the figure. This is a nice treatment/re-treatment model and showing more of the quantified data would strengthen the data

Response: We appreciate the reviewer's insightful suggestion. We have now included the quantified luciferase radiance data for key time points—pre-reinduction, reinduction and post-reinduction—in Supplementary Figure 29C. In addition, we have updated the manuscript and the corresponding supplementary figure legend to reflect these changes.

Reviewer #2 (Remarks to the Author):

The authors have addressed all of my previous queries either by showing new data or appropriate changes to the text and the findings described by the manuscript are now sufficiently well evidenced for publication.

Response: Thank you for your helpful comments and suggestions. We're glad to hear that the revisions and additional data have addressed your concerns. We appreciate your positive

feedback and support for the manuscript.

Reviewer #3 (Remarks to the Author):

The authors have addressed most of the concerns of all the reviewers. Overall, the manuscript has improved; the authors added significant information and added important figures.

Some minor comments to be addressed:

1. "Hispanic" is not a race; please change the subheading of the table and indicate race/ethnicity.

Response: We appreciate the reviewer's suggestion. As per the recommendation, we have updated the subheading of Supplementary Table 1 to "Race/Ethnicity."

2. Add in the table of patients the percent of IL1RL1+LSCs and IL1RL1+IL-33+ LSCs

Response: Thank you for the reviewer's suggestion. We have now included the percent of IL1RL1+LSCs and IL1RL1+IL-33+LSCs in Supplementary Table 1.

3. If the authors included FMOs for IL1RL1 and/or IL-33 in their flow cytometry panels, please include them in supplemental data.

Response: Thank you for the reviewer's suggestion. We have now included fluorescence minus one (FMO) control for IL1RL1 and IL-33 in complete responders and non-responders of AML patients, as shown in Supplementary Figure 1E.

4. If available, please include a table with the characteristics of the healthy donors (i.e., sex, age)

Response: Thank you for the reviewer's suggestion. The healthy human bone marrow frozen cells (Catalog: 2S-101D) were obtained from Lonza. However, detailed donor characteristics (e.g., sex, age) are not available from the supplier.

5. How do the patients split in the survival curves if the parameter is subdivided into quartiles?

Response: We appreciate the reviewer's thoughtful comment. In response, we stratified patients from the combined TARGET-AML and TCGA-AML datasets based on IL1RL1 expression, dividing them into four groups according to the quartiles of IL1RL1 expression. As shown in the Figure below, patients with IL1RL1 expression above the 75th percentile exhibited significantly reduced overall survival compared to all 3 other quartile groups ($p = 0.007$).

6. Can the authors clarify when the sample shown was taken (when separating people as NR and CR)-figure 1 and supplemental Fig 1?

Response: Bone marrow (BM) aspirates were collected from 12 AML patients who achieved complete response (CR) and 10 non-responders approximately 21 days following induction chemotherapy. We have clarified this in the Materials and Methods section under the subheading Human research participants.

7. It will be helpful to use the same nomenclature in the table of patients and in the figures, e.i. Non-responder/Refractory.

Response: Thank you for the reviewer's suggestion. We have updated the terminology throughout the manuscript and figure legends to consistently use "non-responder (NR)" and "complete responder (CR)."

Reviewer #4 (Remarks to the Author):

The others have appropriately addressed my concerns.

Response: Thank you for your review and feedback. We're pleased that the revisions have addressed your concerns, and we appreciate your time and support.